# Mitochondrial metabolism sustains *DNMT3A*-R882-mutant clonal haematopoiesis

Malgorzata Gozdecka[1,2✉], Monika Dudek[1,2,19], Sean Wen[2,3,19], Muxin Gu[1,2], Richard J. Stopforth[4], Justyna Rak[1,2], Aristi Damaskou[1,2], Guinevere L. Grice[4], Matthew A. McLoughlin[1,2], Laura Bond[1,2], Rachael Wilson[1,2], George Giotopoulos[1,2], Vijaya Mahalingam Shanmugiah[1,2], Rula Bany Bakar[4], Eliza Yankova[1,2,5], Jonathan L. Cooper[1,2], Nisha Narayan[1,2], Sarah J. Horton[1,2], Ryan Asby[1,2], Dean C. Pask[1,2], Annalisa Mupo[6], Graham Duddy[7], Ludovica Marando[1,2], Theodoros Georgomanolis[8], Paul Carter[9], Amirtha Priya Ramesh[1,2], William G. Dunn[1,2], Clea Barcena[1,2,10], Paolo Gallipoli[11], Kosuke Yusa[12], Slavé Petrovski[3], Penny Wright[13], Pedro M. Quiros[1,2,10], Christian Frezza[8,14], James A. Nathan[4], Arthur Kaser[4,15], Siddhartha Kar[16], Konstantinos Tzelepis[1,2,5], Jonathan Mitchell[3], Margarete A. Fabre[2,3,17], Brian J. P. Huntly[1,2,17,20✉] & George S. Vassiliou[1,2,17,18,20✉]

Somatic *DNMT3A*-R882 codon mutations drive the most common form of clonal haematopoiesis (CH) and are associated with increased acute myeloid leukaemia (AML) risk[1,2]. Preventing expansion of *DNMT3A*-R882-mutant haematopoietic stem/progenitor cells (HSPCs) may therefore avert progression to AML. To identify *DNMT3A*-R882-mutant-specific vulnerabilities, we conducted a genome-wide CRISPR screen on primary mouse *Dnmt3a*^R882H/+ HSPCs. Among the 640 vulnerability genes identified, many were involved in mitochondrial metabolism, and metabolic flux analysis confirmed enhanced oxidative phosphorylation use in *Dnmt3a*^R882H/+ versus *Dnmt3a*^+/+ (WT) HSPCs. We selected citrate/malate transporter *Slc25a1* and complex I component *Ndufb11*, for which pharmacological inhibitors are available, for downstream studies. In vivo administration of SLC25A1 inhibitor CTPI2 and complex I inhibitors IACS-010759 and metformin suppressed post-transplantation clonal expansion of *Dnmt3a*^R882H/+, but not WT, long-term haematopoietic stem cells. The effect of metformin was recapitulated using a primary human *DNMT3A*-R882 CH sample. Notably, analysis of 412,234 UK Biobank participants showed that individuals taking metformin had a markedly lower prevalence of *DNMT3A*-R882-mutant CH, after controlling for potential confounders including glycated haemoglobin, diabetes and body mass index. Collectively, our data propose modulation of mitochondrial metabolism as a therapeutic strategy for prevention of *DNMT3A*-R882-mutant AML.

Clonal haematopoiesis (CH), the clonal expansion of haematopoietic stem cells (HSC) and their progeny driven by somatic driver mutations, is associated with increased risk of progression to acute myeloid leukaemia (AML) and other myeloid neoplasms[3–6]. More than half of all CH cases are driven by mutations in *DNMT3A*, the gene for de novo DNA methyltransferase 3A[3,4,7,8]. *DNMT3A* mutations occur throughout the gene but most commonly affect codon R882 in the protein's methyltransferase domain. These alter DNMT3A enzymatic activity

and seem to have a dominant-negative effect and/or lead to defective homodimerization[9,10], resulting in DNA hypomethylation. Studies of murine models showed that *Dnmt3a* mutations impart a differentiation block and increased self-renewal on HSC[11–13], drive HSC clonal expansion in the presence of inflammation[14,15], promote AML development and impart resistance to anthracyclines[16,17]. *DNMT3A*-mutant CH is associated with an increased risk of progression to AML, an association that is driven primarily by *DNMT3A*-R882 hotspot mutations[1,18,19]. Proposed

[1]Cambridge Stem Cell Institute, University of Cambridge, Cambridge, UK. [2]Department of Haematology, University of Cambridge, Cambridge, UK. [3]Centre for Genomics Research, Discovery Sciences, BioPharmaceuticals Research and Development, AstraZeneca, Cambridge, UK. [4]Cambridge Institute of Therapeutic Immunology and Infectious Disease (CITIID), Jeffrey Cheah Biomedical Centre, University of Cambridge, Cambridge, UK. [5]Milner Therapeutics Institute, Jeffrey Cheah Biomedical Centre, University of Cambridge, Cambridge, UK. [6]Altos Labs, Granta Park, Cambridge, UK. [7]The Francis Crick Institute, London, UK. [8]Cluster of Excellence Cellular Stress Responses in Aging-associated Diseases (CECAD), University Hospital Cologne, Cologne, Germany. [9]Section of Cardiovascular Medicine, The Victor Phillip Dahdalleh Heart and Lung Research Institute, The University of Cambridge, Papworth Road, Cambridge Biomedical Campus, Cambridge, UK. [10]Departamento de Bioquímica y Biología Molecular, Instituto Universitario de Oncología (IUOPA), Universidad de Oviedo, Oviedo, Spain. [11]Centre for Haemato-Oncology, Barts Cancer Institute, Queen Mary University of London, London, UK. [12]Stem Cell Genetics, Institute for Life and Medical Sciences, Kyoto University, Kyoto, Japan. [13]Department of Anatomic Pathology, Canterbury Health Laboratories, Christchurch, New Zealand. [14]Institute of Genetics, Faculty of Mathematics and Natural Sciences, Faculty of Medicine, University of Cologne, Cologne, Germany. [15]Division of Gastroenterology and Hepatology, Department of Medicine, University of Cambridge, Addenbrooke's Hospital, Cambridge, UK. [16]Early Cancer Institute, Department of Oncology, University of Cambridge, Cambridge, UK. [17]Department of Haematology, Cambridge University Hospitals NHS Foundation Trust, Cambridge, UK. [18]Wellcome Sanger Institute, Wellcome Genome Campus, Hinxton, Cambridge, UK. [19]These authors contributed equally: Monika Dudek, Sean Wen. [20]These authors jointly supervised this work: Brian J. P. Huntly, George S. Vassiliou. ✉e-mail: mg717@cam.ac.uk; bjph2@cam.ac.uk; gsv20@cam.ac.uk

pharmacological interventions for *DNMT3A*-mutant AML include hypomethylating agents[20], induction of apoptosis and/or necroptosis[21], or targeting genes and/or pathways activated in *DNMT3A*-R882-mutant cells, such as DOT1L[22] and mTOR[23].

Recent advances in our ability to predict the risk of progression to AML and related cancers[1,2] have spurred an interest in therapeutic approaches to avert/delay progression of CH to these cancers. This approach could be applied to *DNMT3A*-R882-mutant CH, given its significant risk of progression to AML. Previous studies reported azacytidine[20] and oridonin[21] as potential strategies for targeting *DNMT3A*-R882 CH. However, their effectiveness has not been formally tested, and there is an unmet need for safe and well-tolerated treatments to prevent progression of *DNMT3A*-R882-mutant CH.

Here, we demonstrate that *DNMT3A*-R882-mutant cells rely on oxidative phosphorylation (OXPHOS) and that their clonal proliferative advantage is vulnerable to inhibition by pharmacological agents targeting mitochondrial genes shown by a genome-wide CRISPR screen in mouse *Dnmt3a*-R882H hematopoietic stem and progenitor cells (HSPCs). We validate these vulnerabilities in our preclinical model of CH using several modulators of mitochondrial metabolism, including CTPI2, IACS-010759 and metformin. Importantly, analysis of 412,234 UK Biobank (UKB) participants showed that individuals taking metformin ($n = 11,190$) exhibited markedly lower prevalence of *DNMT3A*-R882-mutant CH (odds ratio (OR) = 0.49 (95% confidence interval (CI), 0.32–0.74), $P = 0.00081$). Notably, non-R882 *DNMT3A*-CH and CH driven by other driver genes did not exhibit this relationship, emphasizing the striking specificity of metformin for *DNMT3A*-R882 CH. Given the established safety record of metformin, this work proposes its investigation in clinical studies of prevention of *DNMT3A*-R882 CH progression to AML.

## *Dnmt3a*-R882H enhances HSPC self-renewal

To study the effect of *DNMT3A*-R882 hotspot mutations on haematopoiesis and leukaemogenesis, we developed a conditional mouse model by floxing the native murine *Dnmt3a* exon 23 (containing codon R878, equivalent to human R882) and inserting human R882H-mutant exon 23 immediately downstream (*Dnmt3a*^flox-R882H^; Fig. 1a). *Dnmt3a*^flox-R882H^ mice were crossed into the *Mx1-Cre* background, enabling efficient excision of native exon 23 and thereafter expression of *Dnmt3a*-R882H, following polyinosinic-polycytidylic acid (pIpC) administration (hereafter *Dnmt3a*^R882H/+^; Fig. 1b and Extended Data Fig. 1a,b). *Dnmt3a*^R882H/+^ showed no major haematopoietic phenotypes compared to control mice (*Dnmt3a*^+/+^ or WT), including equal numbers of long-term HSC (LT-HSC) at six weeks post-pIpC (Extended Data Fig. 1c–i), mirroring previous reports[16]. With age, *Dnmt3a*^R882H/+^ mice developed modest increases in the proportions of peripheral blood (PB) myeloid and T cells (Extended Data Fig. 1j) and demonstrated reduced frequencies of megakaryocyte-erythroid progenitors (MEPs) in bone marrow (BM) (Extended Data Fig. 1k). Frequencies of BM LT-HSC, short-term HSC (ST-HSC) and other progenitor compartments and PB parameters, including blood leukocyte counts (WBC), haemoglobin (HGB) and platelet counts (PLT) were unchanged (Extended Data Fig. 1k–s). However, *Dnmt3a*^R882H/+^ BM cells displayed a striking self-renewal phenotype in vitro (Fig. 1c) and enhanced BM competitive repopulation in vivo upon transplantation into lethally irradiated mice, in line with other reports[17,20]. A competitive advantage was observed in both PB (Fig. 1d,e and Extended Data Fig. 2a,b) and BM progenitor compartments, including LT-HSC (Fig. 1f–g and Extended Data Fig. 2c).

On ageing to a humane endpoint, *Dnmt3a*^R882H/+^ mice demonstrated decreased survival versus WT (median 487 versus 618 days; $P < 0.0029$; Fig. 1h), in line with other reports[23]. At necropsy, 40% of *Dnmt3a*^R882H/+^ mice had splenomegaly (Extended Data Fig. 2d); however, mean WBC was not elevated (Extended Data Fig. 2e). Histological examination of BM, spleen, liver, heart and lungs showed a higher incidence of myeloproliferative neoplasm (MPN)/AML in *Dnmt3a*^R882H/+^ mice (37%, 13 of 35

Fig. 1i,j), reflecting the higher risk of progression to AML of *DNMT3A*-R882 compared to other *DNMT3A* mutations[1]. Exome sequencing of six *Dnmt3a*^R882H/+^ MPN/AMLs showed several somatic mutations in each MPN/AML, including in genes known to co-occur with *DNMT3A* mutations in human AML, such as *Tet1*, *Setd2* and *Cux1* (Supplementary Table 1).

## Genetic vulnerabilities of *Dnmt3a*^R882H/+^ HSPCs

To understand the genetic vulnerabilities of *Dnmt3a*^R882H/+^ HSPC, we exploited the fact that these cells display increased serial replating in cytokine-enriched media, a surrogate for enhanced self-renewal potential (Fig. 1c). To interrogate the basis of this, we crossed *Dnmt3a*^flox-R882H/+^, *Mx1-Cre* and *Rosa26-Cas9* (hereafter *Cas9*)[24] mice and induced mutation with pIpC. We isolated *Dnmt3a*^R882H/+^; *Cas9*; *Mx1-Cre* BM HSPCs and cultured these in cytokine-rich methylcellulose and then liquid media. HSPCs were transduced with our exome-wide CRISPR library[24] and selected by flow-sorting for blue fluorescent protein (BFP). Cells were then cultured and collected at several timepoints (Fig. 2a). Sequencing for guide RNA (gRNA) content in genomic DNA of surviving cells on day 30 showed 640 HSPC-essential (dropout) genes at false discovery rate ≤ 20% (Fig. 2b and Supplementary Table 2). We compared these with publicly available CRISPR dropout genes from human cancer cell lines. Strikingly, *DNMT3A*-mutant cell lines OCI-AML2 and OCI-AML3 were the top two enriched cell lines of 325 tested (Extended Data Fig. 2f), demonstrating cross-species validation and genotype specificity.

Pathway analyses of depleted genes showed enrichment in pathways including ribosome, spliceosome, proteasome, cell cycle and OXPHOS (Fig. 2c). To remove pan-essential genes, we compared our *Dnmt3a*^R882H/+^ dropouts with a similar screen performed on non-transformed haematopoietic precursor cell line 7 (HPC-7)[25], which retains many characteristics of primary murine HSPCs[26], and found 201 genes specifically depleted in *Dnmt3a*^R882H/+^; *Cas9*; *Mx1-Cre* HSPCs (Fig. 2d and Supplementary Table 3). Among these 201 genes, 35 belonged to the 'druggable genome' category. These included kinases, chromatin regulators and several genes regulating cellular metabolism, including *Slc25a1*, *Slc6a9*, *Ldlr* and *Ffar3*. In addition, among 34 genes with enzymatic functions, 14 have known roles in metabolism, including OXPHOS and the tricarboxylic acid (TCA) cycle: including *Ndufb11*, *Cs*, *Pdhb*, *Hk3* and *Coq2* (Fig. 2e). Many genes in these categories have available inhibitors (Fig. 2f and Supplementary Tables 4 and 5) and represent attractive targets for pharmacological interventions. Because *Dnmt3a*^R882H/+^ vulnerabilities also correlated with human *DNMT3A*-mutant AML cell line dropouts, we asked whether expression of any *Dnmt3a*^R882H/+^-specific vulnerability genes correlated with AML survival. We identified 17 genes, including *SLC25A1*, *LDLR*, *HK3*, *TGFB1*, *ICMT* and *PSMA4*, where higher expression correlated with significantly poorer patient survival (Supplementary Fig. 1).

## Metabolic vulnerabilities of *Dnmt3a*^R882H/+^ HSPCs

To validate our screen, we used gRNAs against selected 'druggable' candidates and observed that in most instances, gene knockout specifically reduced growth of *Dnmt3a*^R882H/+^ more than WT HSPCs (Fig. 3a,b). As genes implicated in cellular metabolism showed specificity towards *Dnmt3a*^R882H/+^ HSPCs, we focused on them as potential therapeutic targets in *DNMT3A*-R882 CH. We first noted that genes involved in sequential pathways of pyruvate conversion to Acetyl-CoA (*Pdhb*), citrate synthesis (*Cs*) and citrate/malate transport (*Slc25a1*) were *Dnmt3a*^R882H/+^ vulnerabilities important for the TCA cycle. Other mitochondrial targets included electron transfer chain (ETC) components *Ndufb11* and *Coq2*. We focused on the two most significantly depleted *Dnmt3a*^R882H/+^-specific candidates, *Slc25a1* and *Ndufb11* (Supplementary Table 3). To address the consequences of their depletion in vivo, we transduced HSPCs from WT, *Cas9* and *Dnmt3a*^R882H/+^, *Cas9* mice with

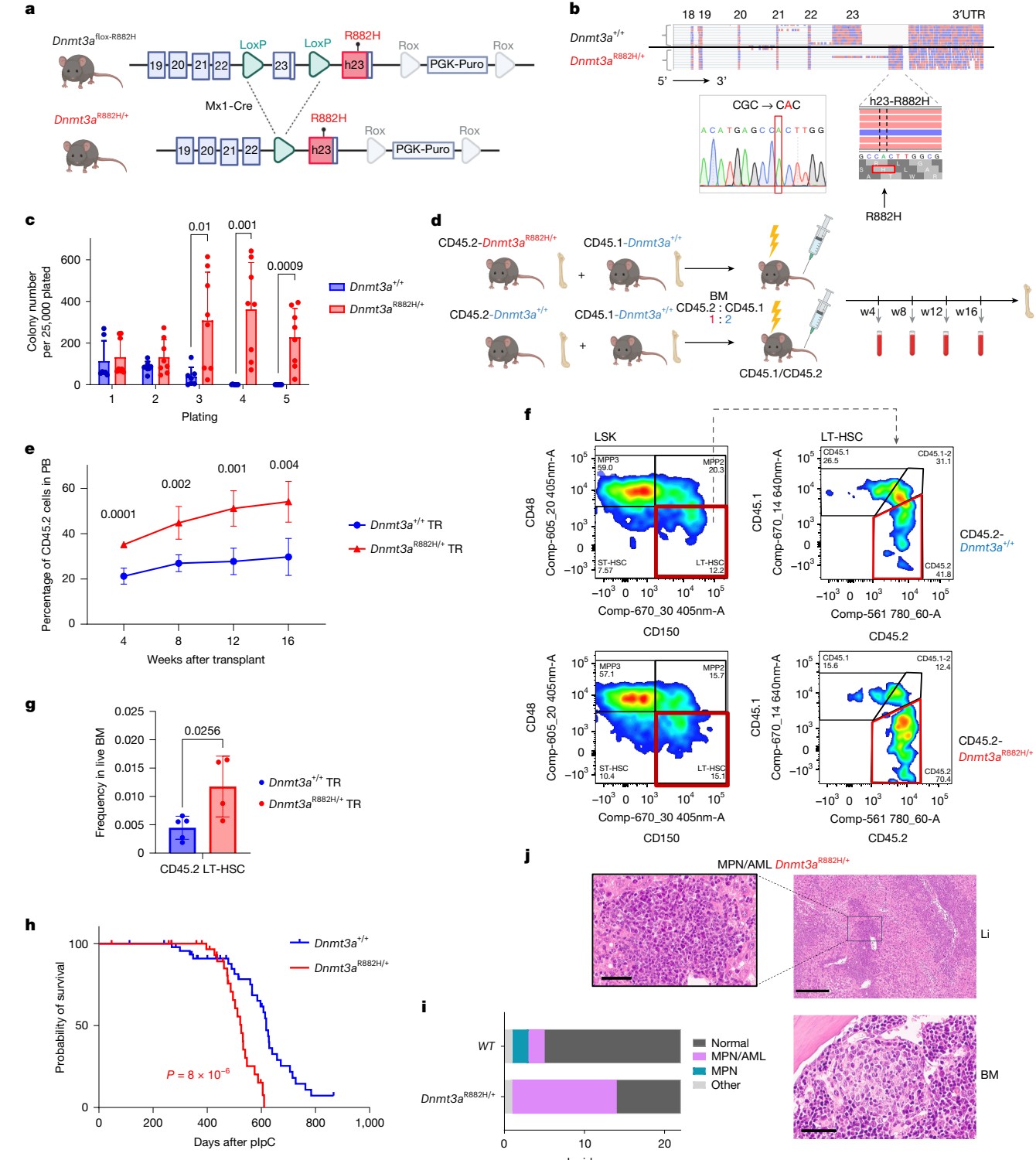

**Fig. 1 | _Dnmt3a_[R882H/+] HSPC shows self-renewal phenotype, enhanced BM repopulation and progression to MPN/AML. a**, Structure of the _Dnmt3a_[R882H] conditional allele. **b**, RNA sequencing reads from _Dnmt3a_[+/+] and _Dnmt3a_[R882H/+] HSPCs were aligned to mouse exons and human exon 23 with _DNMT3A_-R882H mutation. Sanger sequencing was performed on cDNA of _Dnmt3a_[R882H/+] amplified with primers detecting human exon 23. Similar data were observed for _n_ = 3 mice per genotype. **c**, Serial replating of BM-derived colonies from _Dnmt3a_[+/+] (_n_ = 7 mice) and _Dnmt3a_[R882H/+] (_n_ = 8 mice). **d**, Schematic representation of BM competitive transplant strategy of CD45.2-WT and CD45.2-_Dnmt3a_[R882H/+] transplanted together with CD45.1 competitor. **e**, Proportion of _Dnmt3a_[+/+] (_n_ = 5 mice) and _Dnmt3a_[R882H/+] (_n_ = 4 mice) cells in PB post-transplant. **f,g**, Fluorescence-activated cell sorting plots of LT-HSC (Lin[−ve], Sca1[+], c-Kit[+],

Cd48[−]Cd150[+]) and proportion of _Dnmt3a_[+/+] (_n_ = 5 mice) and _Dnmt3a_[R882H/+] (_n_ = 4 mice) transplanted cells in LT-HSC (**f**) with quantification (**g**). **h**, Kaplan–Meier survival curves for _Dnmt3a_[R882H/+] (_n_ = 35) and control (_n_ = 31) mice; _P_ by log-rank (Mantel–Cox) test. **i**, Histopathological diagnoses of moribund mice. Bars depict cancer/normal diagnosis per genotype/total mice with available histology data (_n_ = 44 mice). **j**, Characteristic histology from one mouse with MPN/AML. Myeloid cell/AML infiltration in the liver (Li) and blasts in the setting of myeloid hyperplasia and a hypercellular marrow are shown. Similar data were observed for 13 _Dnmt3a_[R882H/+] mice. w, week. Scale bars, 300 μm for inset (Li), 50 μm for others. In **c**–**e**, **g** and **i**, the mean ± s.d. is shown; _P_ by two-sided _t_-test for comparisons between _Dnmt3a_[R882H/+] and WT. Schematics in **a**,**d** were created using BioRender (https://biorender.com).

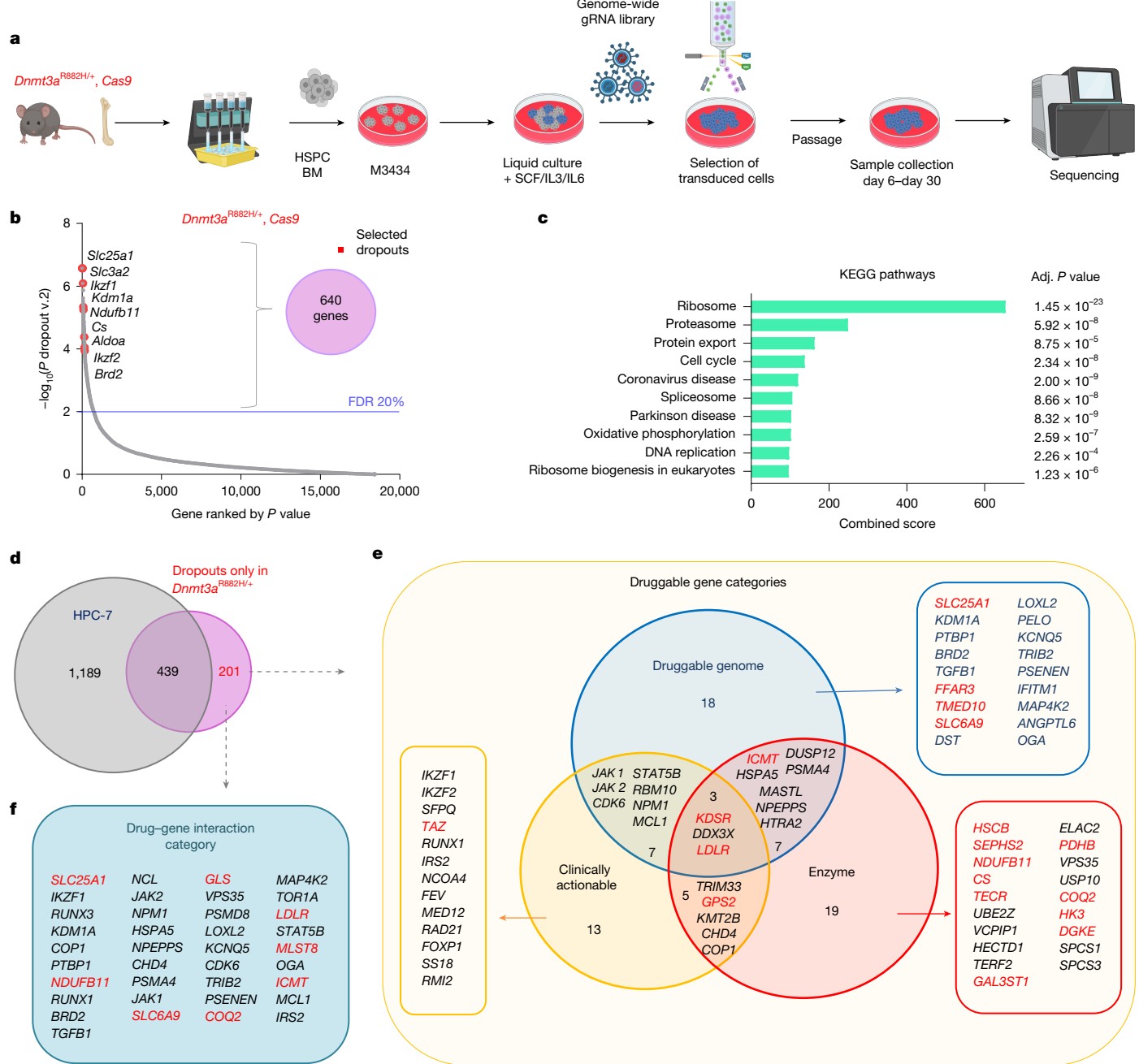

**Fig. 2 | Whole-genome CRISPR screen identifies *Dnmt3a*^R882H/+ drug target candidates. a**, Schematic representation of CRISPR-screen strategy. **b**, Result of the whole-genome CRISPR screen in *Dnmt3a*^R882H/+ HSPC. Selected highly significantly depleted genes are listed. False discovery rate (FDR) was calculated using the MAGeCK statistical package. **c**, KEGG pathway analysis of 640 depleted genes using Enrichr, reporting *P* values adjusted (adj.) for several comparisons. **d**, Venn diagram of *Dnmt3a*^R882H/+ dropouts and HPC-7 dropouts[25].

**e,f**, Classification of 201 dropout genes present only in *Dnmt3a*^R882H/+ cells into potential 'druggable' gene categories (**e**) and drug–gene interaction categories, as defined by DGIdb and a literature search (**f**). Three categories are depicted in **e**. All categories in **f** can be found in Supplementary Table 4, and drug–gene categories are shown in Supplementary Table 5. Schematic in **a** was created using BioRender (https://biorender.com).

lentiviral vectors expressing gRNA for *Slc25a1*, *Ndufb11* and Empty gRNA control. Transduced cells were transplanted into irradiated recipients alongside competitor cells. Four weeks after transplantation, we observed significantly higher reconstitution of *Dnmt3a*^R882H/+ over WT HSCPs transduced with control gRNA. Knockout of *Slc25a1* and *Ndufb11* strongly suppressed the transplantation ability of *Dnmt3a*^R882H/+. Of note, we also observed a negative effect of *Slc25a1* and *Ndufb11* knockout in WT cells, albeit to a lesser extent (Extended Data Fig. 3a–c). With prevention studies in mind, we next tested available pharmacological inhibitors of these two targets: Slc25a1 is directly inhibited by CTPI2[27,28],

whereas Ndufb11 can be inhibited by metformin, a complex I ETC inhibitor that also inhibits mitochondrial glycerophosphate dehydrogenase and IACS-010759, a more specific complex I inhibitor[29]. CTPI2 and metformin significantly reduced colony formation of both preleukaemic and leukaemic *Dnmt3a*^R882H/+; *Cas9* HSPCs (Extended Data Fig. 3d,e). Furthermore, we observed that metformin significantly suppressed the self-renewal potential (plating 3) of *Dnmt3a*^R882H/+ cells (Extended Data Fig. 3f, g), whereas CTPI2 completely abolished enhanced self-renewal of *Dnmt3a*^R882H/+ (platings 2, 3) even at lower concentrations (Extended Data Fig. 3h). We also observed that *NDUFB11* and *COQ2* showed very

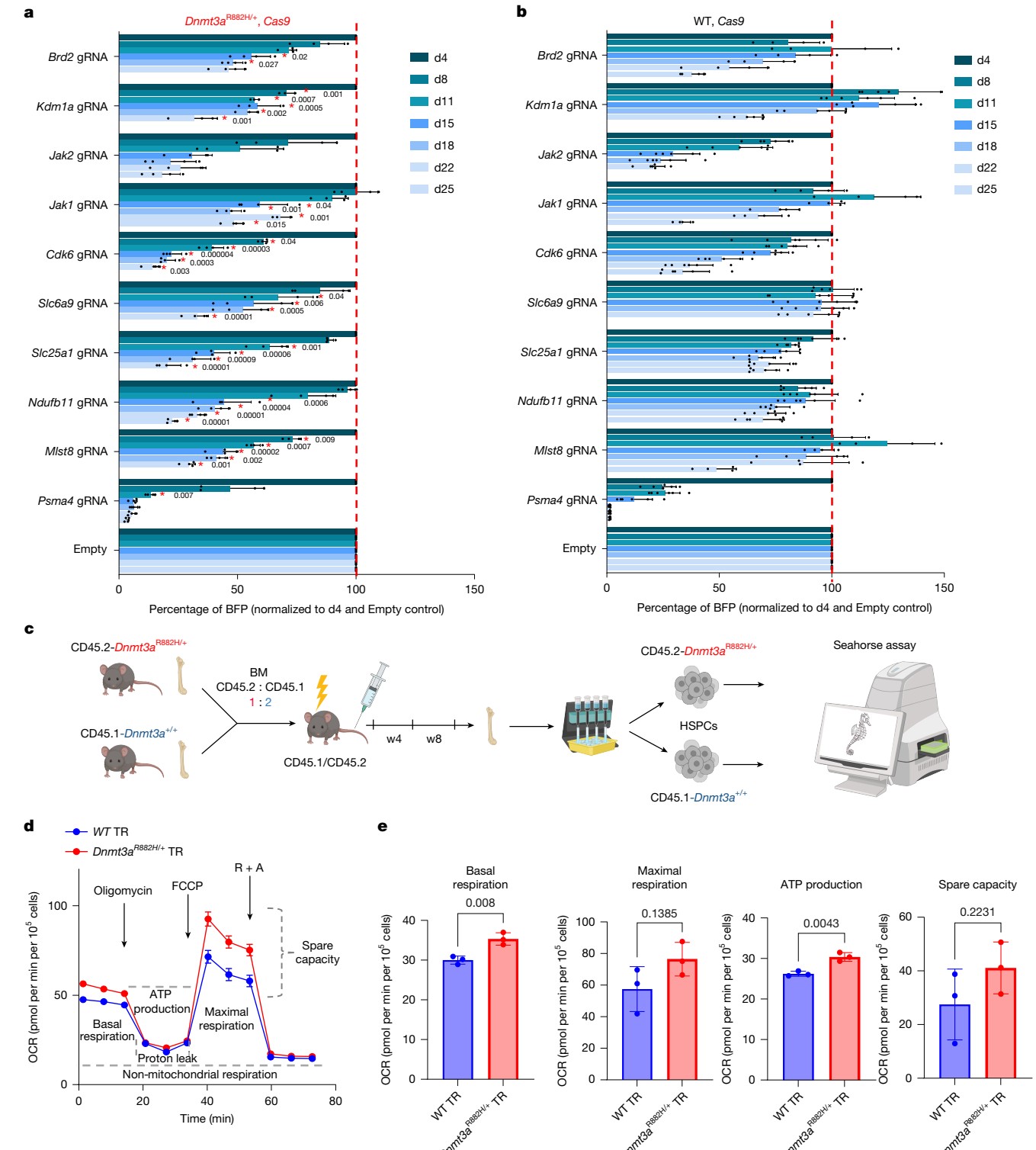

**Fig. 3 | Enhanced mitochondrial respiration in transplanted *Dnmt3a*[R882H/+].**
**a**,**b**, Proliferation of *Dnmt3a*[R882H/+] (**a**) and WT (**b**) HSPCs after editing of the indicated gene. The BFP-positive fraction was compared with the non-transduced population and normalized to day 4 and Empty vector for each gRNA. For **a** and **b**, the mean ± s.d. is shown; *n* = 3 biological replicates for *Kdm1a* gRNA and *n* = 4 biological replicates for the remining gRNAs in *Dnmt3a*[R882H/+]. *n* = 4 biological replicates for *Mlst8, Jak1, Brd2* gRNAs and *n* = 6 biological replicates for the remining gRNAs in WT. Asterisk indicates significant depletion in *Dnmt3a*[R882H/+] versus WT HSPCs calculated for each day. *P* by two-sided *t*-test. **c**, Schematic representation of the experimental setup for mitochondrial respiration analysis of WT and *Dnmt3a*[R882H/+] HSPCs extracted eight weeks post-transplantation performed with the Seahorse analyser. **d**, Example of

OCR in transplanted WT and *Dnmt3a*[R882H/+] HSPCs, measured using a Seahorse extracellular flux analyser; mean ± s.e.m. *n* = 18 for *Dnmt3a*[R882H/+], *n* = 15 for WT representing three independent biological replicates, for each *Dnmt3a*[R882H/+] performed in six replicates, for WT performed in four, five and six replicates respectively. R + A indicates rotenone and antimycin A. **e**, Basal respiration, maximal respiration, ATP production and spare respiration capacity were calculated; mean ± s.d. *P* by two-sided *t*-test. *n* = 3 biological replicates per genotype. Similar results were obtained when *Dnmt3a*[R882H/+] versus WT cells were transplanted separately into recipients (Extended Data Fig. 4g–i). *Higher depletion in *Dnmt3a*[R882H/+] versus WT. d, day. Schematic in **c** was created using BioRender (https://biorender.com). Seahorse picture in **c** adapted with permission from K. Gozdecki.

high dependency scores in human *DNMT3A*-mutant AML cell lines among the 37 myeloid cell lines with available data (Extended Data Fig. 4a), and *SLC25A1* was also depleted in all three *DNMT3A*-mutant lines, albeit more modestly. Furthermore, another two genes involved in citrate synthesis, *PDHB* and *CS*, showed high *DNMT3A*-mutant dependency among myeloid cell lines (Extended Data Fig. 4b).

## Reliance of *Dnmt3a*<sup>R882H/+</sup> HSPCs on OXPHOS

To determine the effect of *Dnmt3a*-R882H on cellular metabolism, we isolated primary HSPCs from *Dnmt3a*<sup>R882H/+</sup> and WT mice and compared the use of glycolysis and OXPHOS for cellular bioenergetics using metabolic flux analysis. First, in homoeostatic conditions, we observed no differences in mitochondrial respiration between *Dnmt3a*<sup>R882H/+</sup> and WT HSPCs (Extended Data Fig. 4c,d). This finding is in consonance with the observation that our *Dnmt3a*-R882H model demonstrates relatively little alteration of steady-state haematopoiesis. However, in response to stressful stimuli, such as stem cell transplantation, inflammation or in vitro culture, *Dnmt3a*<sup>R882H/+</sup> cells display a distinct clonal advantage reminiscent of that seen in individuals with CH. To understand whether this stress-related clonal advantage correlates with altered metabolism, we co-transplanted *Dnmt3a*<sup>R882/+</sup> and WT BM cells into congenic recipients and isolated *Dnmt3a*<sup>R882H/+</sup> and WT HSPCs from the same recipient animals eight weeks after transplantation. This showed enhanced basal mitochondrial respiration and increased ATP production in *Dnmt3a*<sup>R882H/+</sup> versus WT cells (Fig. 3c–e). Concomitantly, glycolysis was unchanged, and glycolytic capacity was reduced in *Dnmt3a*<sup>R882H/+</sup> (Extended Data Fig. 4e,f), indicating a switch in metabolism and greater dependence of mutant cells on mitochondrial respiration. Similarly, WT and *Dnmt3a*<sup>R882H/+</sup> HSPCs transplanted independently into recipient mice also showed enhanced OXPHOS (Extended Data Fig. 4g–i). We then asked whether CTPI2 and metformin could reverse this aberrant respiration, observing that both drugs significantly reduced mitochondrial respiration in *Dnmt3a*<sup>R882H/+</sup> HSPCs (Extended Data Fig. 5a–d).

## Slc25a1 block curtails *Dnmt3a*<sup>R882H/+</sup> HSC expansion

To investigate the effect of Slc25a1 inhibition in vivo, we treated *Dnmt3a*<sup>R882H/+</sup> and WT mice with CTPI2 three times per week for two weeks (Extended Data Fig. 6a). CTPI2 treatment had no effect on the frequencies of LT-HSC, total BM cell or progenitor compartment frequencies, including Lin<sup>−ve</sup>, LSK and multipotent progenitors (MPP) cells (Extended Data Fig. 6b–g). Interestingly, CTPI2 did decrease the frequencies of myeloid cells in *Dnmt3a*<sup>R882H/+</sup> BM but not in WT mice (Extended Data Fig. 6h). Because *Dnmt3a*<sup>R882H/+</sup> HSPCs show enhanced contribution to haematopoiesis after transplantation, we aimed to determine the effect of previous CTPI2 treatment on LT-HSC potential to repopulate irradiated recipient mice. We collected BM from CTPI2/vehicle-treated *Dnmt3a*<sup>R882H/+</sup> and WT mice, mixed them with CD45.1 BM competitor cells in a 1:2 proportion and transplanted them into irradiated recipients (Fig. 4a). Transplant of vehicle-treated *Dnmt3a*<sup>R882H/+</sup> cells showed an enhanced PB contribution in comparison to vehicle-treated WT donors, as expected (Fig. 4b). CTPI2 treatment significantly inhibited the competitive advantage of *Dnmt3a*<sup>R882H/+</sup> cells in PB. Importantly, CTPI2 treatment had no effect on the reconstitution of WT cells (Fig. 4b). Looking at individual haematopoietic lineages, vehicle-treated *Dnmt3a*<sup>R882H/+</sup> cells showed a slightly higher contribution to the B cell lineage, which was reversed by CTPI2 at weeks 8 and 16 (Extended Data Fig. 6i). Conversely, CTPI2 did not affect proportions of T cells and myeloid cells in recipients of either WT or *Dnmt3a*<sup>R882H/+</sup> cells (Extended Data Fig. 6j,k). Blood morphology and indices, including WBC, HGB and red blood cells, were not affected by CTPI2 (Extended Data Fig. 6l–n). PB platelet counts were slightly increased in vehicle-treated *Dnmt3a*<sup>R882H/+</sup> versus WT at weeks 12 and 16. CTPI2 reversed the increased platelet counts in *Dnmt3a*<sup>R882H/+</sup> recipients

to the level of WT (Extended Data Fig. 6o). Strikingly, analysis of the BM haematopoietic compartment of recipients transplanted with cells from vehicle-treated mice showed significantly higher frequencies of *Dnmt3a*<sup>R882H/+</sup> LT-HSC versus WT. By comparison, CTPI2 treatment before transplantation significantly reduced the frequencies of *Dnmt3a*<sup>R882H/+</sup> LT-HSC to WT levels (Fig. 4c,d). Importantly, CTPI2 treatment had no effect on frequencies of WT LT-HSC (Fig. 4c,d), total (CD45.1 and CD45.2 combined) LT-HSC, MPPs, LSK or BM cellularity (Extended Data Fig. 6p–t). These data indicate that *Slc25a1* is a specific therapeutic target in *Dnmt3a*<sup>R882H/+</sup> CH, and its pharmacological targeting prevents expansion of mutant LT-HSC without affecting WT LT-HSC.

## Complex I block curtails *Dnmt3a*<sup>R882H/+</sup> HSC growth

To target Ndufb11 in vivo, we focused on IACS-010759, a clinical-grade, selective small-molecule inhibitor of complex I (ref. 29). We first tested IACS-010759 on colony formation in semisolid media. We isolated HSPCs from transplanted WT mice and *Dnmt3a*<sup>R882H/+</sup> mice. The latter, unlike primary *Dnmt3a*<sup>R882H/+</sup> HSPCs, displayed increased colony formation from first plating. IACS-010759 treatment significantly reduced colony formation in *Dnmt3a*<sup>R882H/+</sup>, sparing WT HSPCs (Fig. 4e,f). To investigate IACS-010759's effect in vivo in the transplant setting, we treated primary *Dnmt3a*<sup>R882H/+</sup> and WT mice with IACS-010759/vehicle for two weeks. BM cells from treated mice were then mixed with competitor cells and transplanted into recipients (Fig. 4g). IACS-010759 had no effect on proportions of total CD45.2-*Dnmt3a*<sup>R882H/+</sup> or WT cells, nor on B, T and myeloid cell reconstitution in the PB (Extended Data Fig. 7a–d). Blood morphology and parameters, including HGB, HCT and PLT, were unchanged in recipients of IACS-010759-treated cells, whereas WBC counts were decreased at weeks 4 and 12, but normalized by week 16 (Extended Data Fig. 7e–h). Analysis of vehicle-treated chimeric BM at 16 weeks showed significantly increased frequencies of LT-HSC from *Dnmt3a*<sup>R882H/+</sup> versus WT cells. Importantly, IACS-010759 treatment significantly decreased the frequency of *Dnmt3a*<sup>R882H/+</sup> but not WT LT-HSC in BM (Fig. 4h). Total (CD45.1 + CD45.2) frequencies of LT-HSC and total BM live cells were unaffected (Extended Data Fig. 7i,j). Because two recent IACS-010759 phase I trials reported unexpected side effects that would preclude use of IACS-010759 to treat CH, at least at the doses used in these trials[30], we tested metformin, a safe, well-tolerated and commonly used complex I inhibitor, which also targets further genes/pathways identified as *Dnmt3a*-R882H-specific vulnerabilities in our screen, including mTOR[31], STAT5[32], TGF-β1[33], GLS[34] and Cyclin D complex[34]. Given the established safety of metformin, we opted to prolong treatment duration. BM cells from CD45.2-*Dnmt3a*<sup>R882H/+</sup> mice and CD45.1-WT mice were mixed in a 1:2 ratio and transplanted into irradiated mice. Five weeks post-transplantation, mice were treated with metformin for six weeks (Fig. 4i). PB analysis before and after treatment showed no altered proportion of *Dnmt3a*<sup>R882H/+</sup> or WT cells (Extended Data Fig. 7k). However, BM analysis six weeks post-treatment showed a significant increase in the frequencies of *Dnmt3a*<sup>R882H/+</sup> LT-HSC in comparison to WT HSPCs in the vehicle-treated controls, which was significantly reversed by metformin treatment (Fig. 4j). Total frequencies of LT-HSC were also significantly decreased by metformin, reflecting its effect on *Dnmt3a*<sup>R882H/+</sup> (Extended Data Fig. 7l). Metformin also decreased LSK frequencies in *Dnmt3a*<sup>R882H/+</sup> but not WT cells (Extended Data Fig. 7m,n); however, the total frequencies of BM *Dnmt3a*<sup>R882H/+</sup> and WT cells were not affected (Extended Data Fig. 7o). Secondary transplantation of BM from metformin-treated chimeric mice showed a significant decrease in the proportion of *Dnmt3a*<sup>R882H/+</sup> cells in PB compared to vehicle (Extended Data Fig. 7p,q).

## *Slc25a1* and complex I loss increase DNA methylation

*SLC25A1* loss in humans is associated with D, L-2hydroxyglutaric aciduria caused by the accumulation of D and L enantiomers of

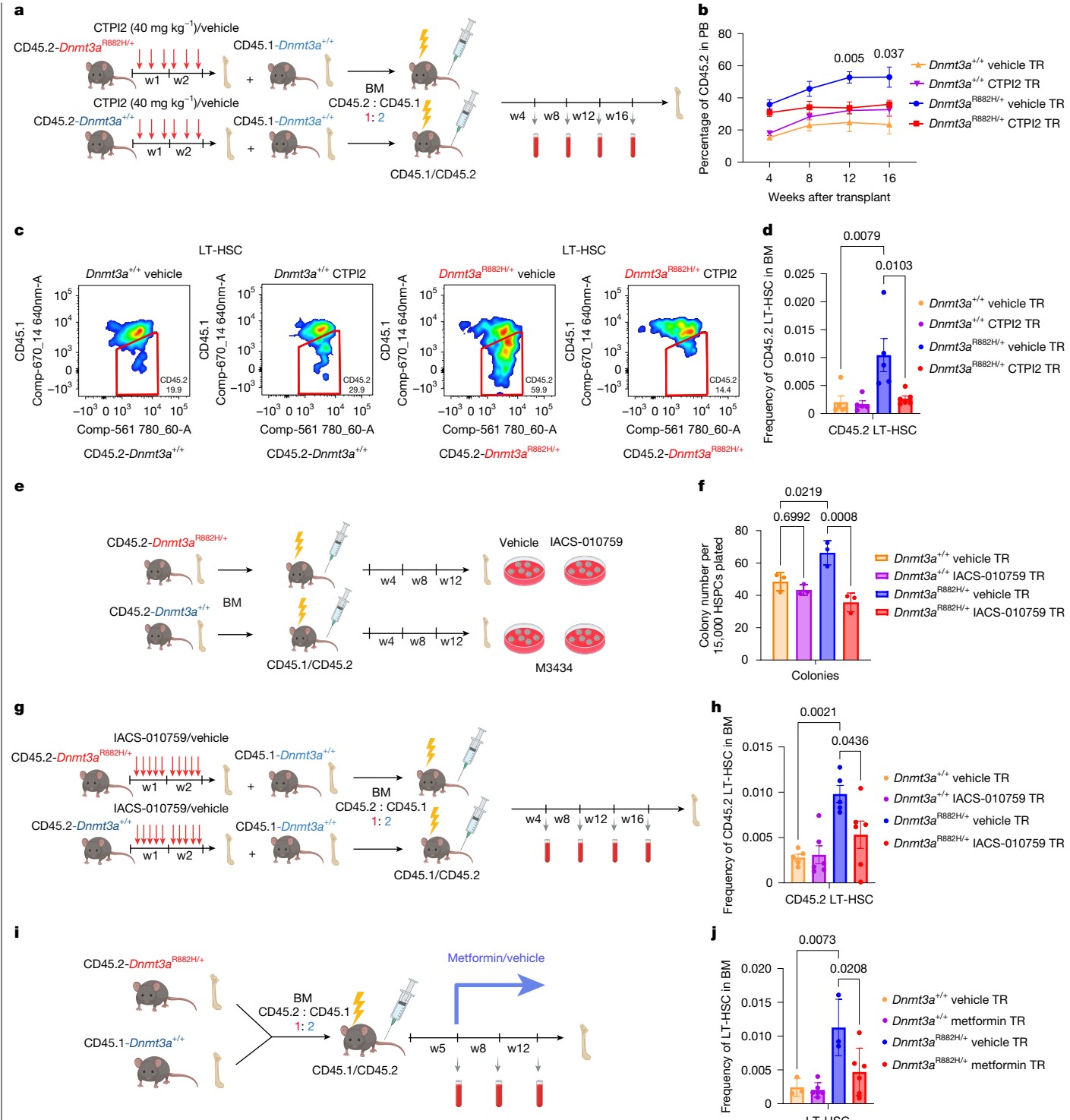

**Fig. 4 | CTPI2 and complex I inhibitors revert clonal advantage of**
**$Dnmt3a^{R882H/+}$ LT-HSC. a**, Scheme of experimental approach. **b**, The proportion
of CD45.2 cells in PB, normalized to the proportion of injected CD45.2 cells.
The mean ± s.e.m. is shown; for $Dnmt3a^{R882H/+}$ CTPI2, $n = 6$, for remining groups,
$n = 5$ mice; $P$ by two-sided $t$-test between $Dnmt3a^{R882H/+}$ vehicle and $Dnmt3a^{R882H/+}$
CTPI2 is shown. **c**, The proportion of transplanted cells in LT-HSC by flow
cytometry at week 16. One sample per group is shown; similar results were
observed for $n = 5$. **d**, Frequencies of CD45.2-LT-HSC in BM at week 16,
normalized to the proportion of injected LT-HSC. The mean ± s.e.m. is shown;
for $Dnmt3a^{R882H/+}$ CTPI2, $n = 6$ mice, for remining groups, $n = 5$ mice. **e**, Schematic
summary of the experimental approach. HSPCs isolated 12 weeks post-
transplant were plated in semisolid media with 100 nM IACS-010759/vehicle for

seven days. **f**, Quantified colonies; the mean ± s.d. is shown; $n = 3$ mice per group.
**g**, Schema of experimental approach for IACS-010759/vehicle treatment.
**h**, Frequencies of CD45.2 LT-HSC in mouse BM at endpoint. The mean ± s.e.m. is
shown; $n = 5$ mice for both vehicle groups, and $n = 6$ for both IACS-010759 groups.
**i**, Schematic representation of metformin treatment model. $Dnmt3a^{R882H/+}$ were
mixed with WT BM cells in 1:2 proportion and transplanted into lethally irradiated
recipient mice. Metformin (125 mg kg⁻¹) or vehicle treatment started from week 5
for six weeks. **j**, Frequencies of transplanted $Dnmt3a^{R882H/+}$ and WT LT-HSC in BM;
the mean ± s.d. is shown; $n = 3$ mice for vehicle group, and $n = 6$ mice per metformin
group. $P$ in **d**, **f**, **h**, **j** by one-way ANOVA with Tukey correction. Schematics in
**a**, **e**, **g**, **i** were created using BioRender (https://biorender.com).

2-hydroxyglutarate (2-HG)[35–37]. Using targeted metabolomics, we confirmed increased levels of 2-HG in both $Dnmt3a^{R882H/+}$ and WT HSPCs on Slc25a1-knockout (Extended Data Fig. 8a,b). Enantiomer analysis showed enrichment in both D- and L-2HG, as reported for humans with $SLC25A1$ loss (Extended Data Fig. 8c,d). Raised 2-HG levels can affect the activity of TET dioxygenases[38] and affect DNA hydroxymethylation (5hmC). We confirmed a global reduction in DNA methylation (5-methylcytosine, 5mC) (Extended Data Fig. 8e,f) and also observed significantly reduced total 5-hydroxymethylcytosine (5hmC) levels in $Dnmt3a^{R882H/+}$ versus WT (Extended Data Fig. 8e,g), reflecting the globally reduced levels of its precursor, 5mC. Interestingly, on $Slc25a1$ knockout, we observed a significant increase in DNA methylation in WT (Extended Data Fig. 8h) and, to a greater extent, $Dnmt3a^{R882H/+}$ cells (Extended Data Fig. 8i). The level of 5hmC on $Slc25a1$ knockout compared to Empty control was unchanged (Extended Data Fig. 8j,k), indicating that the observed increase in DNA methylation is not dependent on TET inhibition by 2-HG. We also tested whether metformin affects DNA methylation in the $Dnmt3a^{R882H/+}$ context. For this, we transplanted $Dnmt3a^{R882H/+}$ BM cells into lethally irradiated recipients and treated them for four weeks with metformin. Interestingly, we observed increased 5mC levels in metformin-treated mice versus vehicle controls (Extended Data Fig. 9a–c). 2-HG levels were not affected by metformin (Extended Data Fig. 9d), indicating that the increase in DNA methylation was not due to TET inhibition. Therefore, in addition to their significant effect on OXPHOS, both $Slc25a1$-knockout and metformin increased 5mC in $Dnmt3a^{R882H/+}$ HSPCs.

## $DNMT3A$-R882-mutant CH in metformin users

Metformin is the most commonly prescribed drug for Type 2 diabetes (T2D), with millions taking this drug worldwide[39]. Encouraged by its ability to curtail $Dnmt3a^{R882H/+}$ LT-HSC expansion, we interrogated the interaction between metformin and CH prevalence among 412,234 UKB participants. Using logistic regression with age, sex, smoking status and the first four genetic principal components as covariates, we found that the risk of $DNMT3A$-CH was substantially reduced among the 11,190 individuals taking metformin, compared to those not taking metformin, at the time of recruitment/blood sampling (OR = 0.86 (95% CI, 0.77–0.96), P = 0.0095) (Fig. 5a,b and Supplementary Table 6). Notably, this association was driven by $DNMT3A$-R882-mutant CH (OR = 0.49 (95% CI, 0.32–0.74), P = 0.00081), with $DNMT3A$ non-R882-CH showing a non-significant reduction trend (OR = 0.91 (95% CI, 0.81–1.03), P = 0.13). We also stratified non-R882 variants of known functionality[40] into high (n = 950) and low (n = 1,550) functionality categories. Again, we found no association between metformin exposure and prevalence of either high- or low-functionality variants (P = 0.38 and P = 0.9, respectively). In addition, there was no association between metformin use and prevalence of other common forms of CH (Fig. 5b). As many individuals on metformin were also taking other antidiabetic medications, most commonly sulphonylureas and insulin, we next restricted the analysis to those taking only metformin and found that they still had a markedly decreased risk of $DNMT3A$-R882-mutant CH (OR = 0.35 (95% CI, 0.18–0.71), P = 0.0034; Fig. 5c and Supplementary Table 6). The effect of metformin was greater for smaller clones (variant allele frequency (VAF) ≤ 0.1), but the same trend was observed for large clones (VAF > 0.1) (Extended Data Fig. 10a,b and Supplementary Table 6). We next wanted to ascertain the prevalence of $DNMT3A$-R882-mutant CH and other CH subtypes among diagnosed diabetics who had not been on metformin at any time before recruitment, but we could not confidently identify these in the UKB, particularly as 90% of individuals diagnosed with T2D in the UK at that time were treated initially with metformin[39]. Instead, we searched for diabetics who were undiagnosed and untreated at the time of recruitment, using two different approaches: (1) individuals with high levels of glycated HGB (HbA1c) at recruitment[41] and (2) those who started

metformin at a later time point ('premetformin'). We observed that neither group (Fig. 5d,e and Supplementary Table 7) had altered rates of $DNMT3A$-R882-mutant CH (OR = 0.90 (95% CI, 0.34–2.40), P = 0.83 and OR = 0.82 (95% CI, 0.45–1.49), P = 0.52, respectively), indicating that T2D per se does not significantly alter $DNMT3A$-R882-mutant CH risk. Also, individuals taking only sulphonylureas or only insulin did not display an altered prevalence of $DNMT3A$-R882-mutant CH (OR = 0.83 (95% CI, 0.21–3.32), P = 0.79 and OR = 1.05 (95% CI, 0.47–2.34), P = 0.91, respectively; Extended Data Fig. 10c,d and Supplementary Table 6). Furthermore, using Mendelian randomization (MR) instruments, we examined the effect of HbA1c[42,43], T2D[44], body mass index (BMI) and waist-to-hip ratio adjusted for BMI[45] on prevalence of $DNMT3A$-R882 CH. Reassuringly, we found no significant associations (Extended Data Fig. 10e–h and Supplementary Table 8), providing further and independent genetic support for the premise that the association between metformin use and reduced $DNMT3A$-R882 CH is causal. Taken together, these findings show that metformin use is associated with significantly reduced risk of $DNMT3A$-R882-mutant CH and propose this well-tolerated drug as a potential intervention to prevent or retard its clonal expansion and, in so doing, reduce the likelihood of progression to AML.

## Metformin restrains human $DNMT3A$-R882 CH

To directly validate our findings experimentally in human primary cells, we plated human blood mononuclear cells from a $DNMT3A$-R882 CH carrier (VAF 12%) for colony formation in semisolid media in the presence of metformin or vehicle control for 13 days (Fig. 5f). Both Sanger (Fig. 5g) and next-generation (Fig. 5h) sequencing of pooled plate DNA showed significantly decreased $DNMT3A$-R882 VAF. Furthermore, genotyping of 96 individual colonies per condition confirmed a significant decrease in $DNMT3A$-R882 versus WT colonies with metformin treatment (Fig. 5i). We also tested the effect of CTPI2 treatment on human primary $DNMT3A$-R882 CH in the same manner and found that it reduced the proportion of $DNMT3A$-R882 (versus WT) myeloid (colony-forming unit–granulocyte/macrophage, CFU-G/M), but not erythroid progenitor colonies (burst/colony-forming unit-erythroid, B/CFU-E; Supplementary Fig. 2).

## Discussion

Most cases of myeloid neoplasm arise through evolution of CH clones over many years or decades[46,47], raising the prospect that timely interventions may prevent malignant progression[48], mirroring a paradigm that is well-established in solid cancers[49,50]. However, unlike solid cancers, the option of physically removing premalignant lesions is not available for the precursors of myeloid neoplasms. Instead, putative therapeutic approaches will need to rely on systemic interventions that curtail or abort malignant progression.

In this manuscript, we focused on $DNMT3A$-R882 mutations, the single most common AML-initiating CH mutation[1]. To understand and target the mechanisms driving clonal expansion, we developed a conditional $Dnmt3a^{R882H}$ mouse model that recapitulates the enhanced HSPC self-renewal and delayed/low-penetrance progression to myeloid neoplasm seen with these mutations in humans. Using a genome-wide CRISPR dropout screen, we identified genetic vulnerabilities, many involved in mitochondrial metabolism, including several genes (for example, $SLC25A1$, $LDLR$ and $HK3$) whose expression correlates with worse AML patient survival. Using metabolic flux analysis, we confirmed the increased reliance on OXPHOS of $Dnmt3a^{R882H}$. Normal HSC require respiration to maintain quiescence[51] but favour anaerobic glycolysis for energy production[52–54], whereas leukaemia stem cells displayed a greater reliance on OXPHOS and mitochondrial function[55–57]. Our studies propose that $Dnmt3a^{R882H/+}$ HSPC partially mirror a leukaemia stem cell pattern in their increased reliance on OXPHOS. The mechanisms

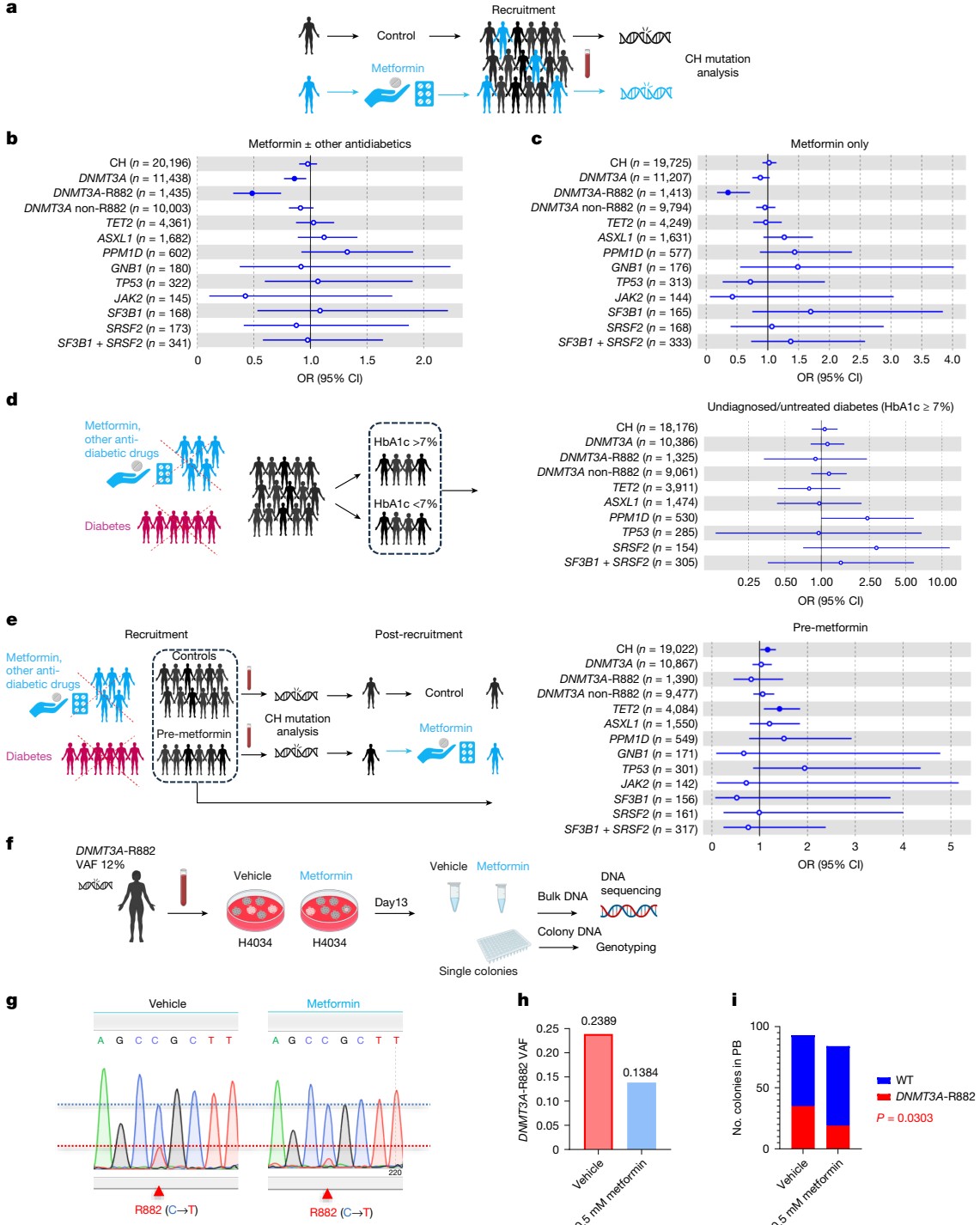

**Fig. 5 | Metformin curtails *DNMT3A*-R882 CH in humans. a**, Schematic representation of the UKB analysis. **b**, Association between metformin and CH risk. In total, 11,190 individuals taking metformin or metformin in combination with other antidiabetic medications and 401,044 individuals not on metformin were analysed. **c**, Association between metformin and CH risk. Only individuals taking metformin were included in the analysis. In total, 5,644 individuals on metformin only and 398,712 individuals not on any form of antidiabetic medications were analysed. **d**, Association between undiagnosed/untreated diabetes and overall CH or gene-specific CH risk. HbA1c > 7% (equivalent of 53 mmol mol⁻¹) was used to identify individuals with undiagnosed/untreated diabetes. Diabetic individuals and individuals taking metformin or other antidiabetic medications at recruitment were excluded. In total, 374,873 individuals with HbA1c data available were included (1,195 individuals with HbA1c > 7% and 373,642 with HbA1c ≤ 7%). **e**, Association between post-recruitment metformin intake and CH risk. Diabetic individuals and individuals

taking metformin or other antidiabetic medications at recruitment were excluded. In total, 3,568 individuals who started on metformin at some time after recruitment (post-recruitment metformin) and 389,153 controls were included for analysis. In **b**–**e**, measures of centre represent the ORs, and the error bars represent the lower and upper bound of the 95% CI of the ORs. ORs and two-sided unadjusted *P* values were derived from logistic regression model with all CH or gene-specific CH as outcome, and with age, sex, smoking and the first four genetic principal components as covariates. Significant *P* values (<0.05) are indicated with full blue circles. **f**, Experimental strategy for testing the effect of metformin on *DNMT3A*-R882 clonal growth (versus WT). **g**,**h**, Sanger sequencing (**g**) and next-generation sequencing (**h**) of DNA collected from bulk colonies. **i**, Comparison of the proportion of WT versus *DNMT3A*-R882 colonies in metformin- versus vehicle-treated cells (*P* by two-sided Chi-square test). Schematics in **a**,**d**,**e**,**f** were created using BioRender (https://biorender.com).

through which *DNMT3A*-R882 mutations may drive a shift towards OXPHOS are inadequately understood and warrant future investigation.

With our focus being the development of therapeutic approaches to avert AML development, we used extant therapeutics to target selected vulnerabilities—namely, the citrate transporter SLC25A1 (with CTPI2) and ETC complex I component NDUFB11 (with metformin or IACS-010759)—and found that these treatments reversed the selective growth advantage of *Dnmt3a*[R882H] over WT LT-HSC in transplanted recipients. Significantly, analysis of UKB data demonstrated a markedly lower prevalence of *DNMT3A*-R882-CH among individuals taking metformin, but not untreated diabetics. Interestingly, we observed an increased risk of *TET2*-CH in diabetics who later started metformin, corroborating the recently reported increased risk of *TET2*-CH, but not *DNMT3A*-CH, in people with high BMI[58]. Also, using MR, we show that genetic instruments for potential confounders such as HbA1c, diabetes, BMI and waist-to-hip ratio do not affect *DNMT3A*-R882-CH prevalence, providing further independent support for a causal relationship between metformin and reduced *DNMT3A*-R882-CH. These findings are compelling and propose metformin as a potential non-toxic intervention to retard or stall *DNMT3A*-R882-CH expansion and in turn avert/delay progression to AML.

The recognition of aberrant cellular energetics as a hallmark of cancer[59] has led to therapeutic targeting of metabolic pathways. In AML, targeting mitochondrial functions including mitochondrial protein synthesis[60], mitochondrial DNA replication[61] and NADH dehydrogenase[29,62] has been proposed as a therapeutic strategy. In fact, both IACS-010759 and metformin have been tested in AML. IACS-010759 has shown impressive efficacy in preclinical models[29]; however, its use as a single agent at relatively high doses was hampered by dose-limiting neurotoxicity in early clinical studies[30]. Metformin can promote apoptosis, induce cell cycle arrest and inhibit cell proliferation in AML[63,64], and its use has been associated with a lower risk of all-cause mortality in cancer patients with coexisting diabetes[65]. Mechanistically, metformin targets complex I[66,67] as well as activating AMPK, leading to downstream effects on cell cycle arrest, autophagy and mTOR inhibition[31,68]. Interestingly, increased mTOR signalling has been observed in *Dnmt3a*[R882H/+23] mice; and of note, the mTOR component *Mlst8* was a vulnerability in our screen. Growth of *Dnmt3a*[R882H/+] cells has also been associated with increased inflammation, in particular TNF signalling[14]. Metformin reduces the levels of cytokines including TNF, IL6 and IL-1β[69,70], which may also hamper the competitive advantage for *Dnmt3a*[R882H/+] HSPCs. We also observed increased 5mC in metformin-treated *Dnmt3a*[R882H/+], in line with previous reports of an increase in global DNA methylation in WT cells linked to changes in *S*-adenosylhomocysteine and *S*-adenosylmethionine[71]. Further studies are required to determine whether this mechanism is also relevant to its effect on *DNMT3A*-R882 CH.

Intriguingly, we also found that *Slc25a1*-ko was associated with raised 5mC as well as moderately increased L- and D-2HG levels. However, we did not find any changes in 5hmC in *Slc25a1*-ko indicating that TET inhibition was not the main mediator of increased 5mC. It should also be noted that *TET2* loss-of-function mutations can co-occur with *DNMT3A* mutations in both CH[1,3,4,72] and myeloid malignancy[73,74], making it unlikely that a simple reciprocal relationship exists between them. This is also supported by the co-occurrence of *DNMT3A* and *IDH1/2* mutations in an AML subset[75]. With regards to other possible mediators of the effect of *Slc25a1* loss, it is noteworthy that a recent study showed increased AMPK in *Slc25a1*[−/−] cells[36], indicating that AMPK activity may also play a role.

Collectively, our findings propose that targeting OXPHOS by existing therapeutics may represent a therapeutic approach for delaying/averting progression of *DNMT3A*-R882-CH to AML. However, before clinical studies are implemented, two critical considerations need to be made: first, can one assume that curtailing or reversing clonal expansion of *DNMT3A*-R882-mutant CH will reduce progression to AML? This premise has not been tested, but several lines of evidence support it,

including observations that both growth rate[46] and clonal size[1,2] are key determinants of AML progression risk. In fact, among UKB participants with *DNMT3A*-R882-mutant CH, those who developed AML had significantly larger CH clones than those who did not[1]. Also, progression from *DNMT3A*-R882-CH to AML happens through acquisition of further mutations, particularly in the *NPM1* gene[73]. Such mutations are more likely to be acquired in larger clones[76]. Therefore, individuals with larger *DNMT3A* clones, in whom AML risk is significantly higher[1,2], could be prioritized for interventions to avert progression. On the other hand, given the strong effect of metformin on smaller clones, consideration could also be given to offering it earlier. Although these observations do not prove the premise, they offer sufficient support to justify a clinical study. Second, in contrast to treating a rapidly fatal disease like AML, preventive treatment administered to someone with a relatively benign state like CH dictates that it should be non-toxic and lack potential long-term side effects. Metformin has been an established treatment for T2D for decades and is currently taken by millions of people worldwide with an acceptable and predictable side-effect profile overall. This, along with our mechanistic, genetic and preclinical data, calls for the initiation of clinical trials of metformin use specifically in individuals with *DNMT3A*-R882-mutant CH[1,47].

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

## Methods

### Mouse models

The in vivo experiments were performed under project licences PPL 80/2564 and PP3797858 issued by the United Kingdom Home Office, in accordance with the Animal Scientific Procedures Act 1986. Murine ethical compliance was approved by the University of Cambridge Animal Welfare and Ethical Review Body. *Dnmt3a*$^{floxR882H/+}$ was constructed by flanking native mouse exon 23 of *Dnmt3a* with *loxP* sites and introducing human exon 23 containing the *DNMT3A*-R882 mutation (in C57/BL6 embryonic stem cells). A PGK-Puro cassette flanked with Rox sites was inserted after the human exon 23. The *Dnmt3a*$^{floxR882H/+}$ mouse model was crossed with *Mx1-Cre* mice[77]. *Cre* expression was induced by intraperitoneal injection of five- to six-week-old mice with pIpC (20 mg kg$^{-1}$, Sigma, P1530): five doses over a period of 10 days. *Cas9*-expressing mice were reported previously[24]. CD45.1 mice: B6.SJL-Ptprca Pepcb/BoyJ mice (Jackson Laboratory, 002014) were used a competitor in transplantation studies. Wild type mice: C57BL/6 J (CD45.2, Jackson Laboratory #000664) were then crossed with CD45.1 to generate CD45.1/CD45.2 recipient mice that were used as transplant recipients between 8 and 18 weeks of age. Mice were housed in specific pathogen-free conditions. All cages were on a 12:12-h light to dark cycle (lights on, 07:30) in a temperature-controlled and humidity-controlled room. Room temperature was maintained at 72 ± 2 °F (22.2 ± 1.1 °C), and room humidity was maintained at 30–70%.

Sample sizes were chosen on the basis of power calculations of expected differences and previous experience with these types of experiments. Sample sizes of $n = 3$ mice per genotype/cell culture were chosen for most experiments with the exception of flow cytometric analysis and mouse experiments where sample size was increased to account for variation between individuals or for the need to carry out experiments at more than one time point. Mice were allocated to the study groups by genotype. Both sexes were used. For transplantation experiments, animals of the same sex and similar age range were randomly assigned to study groups. *Dnmt3a*$^{R882H/+}$ and control mice were 12–16 weeks in most experiments, except aging experiments and characterization of blood compartments at later stages (one-year-old mice), as indicated in the main text. Recipient mice for transplantation experiments were aged 8–16 weeks.

Although the investigators were not blinded to the genotype of the animals, the animal technicians who provided the animal care, supervision and identification of sick animals at humane endpoint (and therefore making the decision to kill sick animals) were blinded. For transplant experiments, flow cytometry data collection and analyses were performed blind (by assigning a number to each animal ID). For in vitro experiments, analyses were performed in batches of animals from both test and control groups using numbers instead of genotypes at the time of data acquisition. For statistical analyses, samples were grouped into test versus control.

### Genotyping

Mouse ear snips were lysed with DirectPCR Lysis Reagent (Viagen, 401-E) according to the manufacturer's instructions. Lysed DNA was used for genotyping each allele with the primers listed in Supplementary Table 9. Polymerase chain reaction (PCR) was performed with REDTaq ReadyMix PCR Reaction Mix (Sigma, R2523) with the following conditions: initial denaturation at 95 °C for 1 min, followed by 35 cycles of 95 °C for 15 s, annealing at 57 °C for 15 s and elongation at 72 °C for 15 s. Final elongation was performed at 72 °C for 10 min. PCR products were visualized on 2.5% agarose gel.

### Histological analysis of mouse tissue

The tissues were fixed in 10% formaldehyde and were subsequently paraffin embedded. Bones were decalcified with 0.38 M EDTA pH 7. Tissue sections (4 μm) were stained with hematoxylin and eosin (Thermo Fisher Scientific). Histology assessment was performed with the Bethesda criteria for mouse haematological tumours[78,79]. The histology slide scanner used for visualization was a Leica Aperio Slide Scanner AT20, running AperioServiceManager.

### Blood-count analysis

Blood-count measurement was performed on a VetabC analyser (Horiba ABX).

### Mouse exome sequencing

DNA for exome sequencing was extracted with a QiaAmp Mini kit (Qiagen, 56304) and submitted to Novogen for the whole-exome sequencing in-house protocol. Briefly, the genomic DNA was randomly sheared into short fragments with a size of 180–280 bp by sonication. The obtained fragments were end repaired, A-tailed and further ligated with Illumina adaptors. The fragments with adaptors were PCR amplified, size selected and purified. The captured libraries were enriched by PCR amplification. The library was created using the SureSelect Mouse All Exon kit (Agilent, G7550B, G7500B) and checked with Qubit and real-time PCR for quantification and a bioanalyzer for size distribution detection. Quantified libraries were pooled and sequenced on the Illumina XPlus platform with a PE150 strategy producing an expected data output of 12 G raw data per sample, according to the effective library concentration and data amount required.

### Whole-exome sequencing data analysis

Whole-exome sequencing reads of AML and control samples were mapped to the mouse genome assembly GRCm39 using BWA v.0.7.18 (ref. 80 (q-bio.GN)) under default parameters, and duplicated reads were flagged using Samtools v.1.9 (ref. 81). Somatic mutations were called by GATK Mutect2 v.4.5 (ref. 82) using AML samples paired with the control sample. Mutation calls were annotated with Ensembl VEP v.112 (ref. 83), and only mutations in exon regions were retained.

### Targeted amplicon sequencing

DNA from bulk colonies was extracted with a DNeasy blood and tissue kit (Qiagen, 69504). DNA was amplified in a 25 μl reaction using HiFi HotStart ReadyMix (Kapa 07958927001) and primers targeting the DNMT3A-R882 region—DNMT3A_Ex23_t1_fp: 5′-ACACTCTTTCCCTAC ACGACGCTCTTCCGATCTCTCTCTGCCTTTTCTCCmCC-3′ and DNMT3A_Ex23_t1_rp: 5′-TCGGCATTCCTGCTGAACCGCTCTTCC GATCTTGTTTAACTTTGTGTCGCTAmCC-3′ at a final concentration of about 4 nM (adjusted for individual primer pairs to attain similar coverage between positions)—and placed in a thermocycler under the following conditions: 95 °C for 3 min, 6 cycles of (98 °C for 20 s, 65 °C for 60 s, 60 °C for 60 s, 55 °C for 60 s, 50 °C for 60 s, 70 °C for 60 s). Following this first round of PCR, samples were kept on ice (to reduce non-specific amplification), and 1 μl of 10 μM i5/i7 index primers were added to each reaction, mixed and placed in a thermocycle under the following conditions: 19 cycles of (98 °C for 20 s, 62 °C for 15 s, 72 °C for 30 s), 72 °C for 60 s. Equal volumes of up to 24 samples (amplified with unique i5/i7 index primer combinations) were pooled, and a 0.55–0.75× double-sided solid-phase reversible immobilization bead cleanup was performed. In most cases, a second solid-phase reversible immobilization bead cleanup (0.75× left-sided) was necessary to reduce contamination of the library (300–400 bp) with adaptors (180–200 bp), as the latter can interfere with sequencing. Libraries were quantified using a Bioanalyzer 2100 (Agilent) and sequenced at 150 bp PE on MiSeq Nano.

### Targeted sequencing data analysis

Targeted sequencing reads were mapped to the human genome assembly GRCh38 using BWA v.0.7.18 (ref. 80 (q-bio.GN)) under default parameters. Samtools mpileup[81], under the parameter -ABQ0, was used to detect mutant reads at the *DNMT3A*-R882 locus.

## Isolation of mouse BM and hematopoietic progenitors

Femur, tibia and hip bones were collected, crushed with a pestle and mortar and lysed with erythrocyte lysis buffer (0.85% $NH_4Cl$; Sigma, A9434). Cells were incubated in the erythrocyte lysis buffer for 5 min at room temperature, spun down, resuspended in PBS containing 2% FBS and filtered through a 40-µm cell strainer (Falcon, 352340). The HSPC compartment was isolated using magnetic-bead selection using a Direct Lineage Cell Depletion Kit (Miltenyi Biotec, 130-110-470) according to the manufacturer's instructions.

## Serial replating assays

For replating assays of mouse cells, 50,000 BM cells were plated in two wells of six-well plates of methylcellulose (Stem Cell Technologies, M3434). The colonies were counted seven days later, and a further 30,000–50,000 cells were reseeded and recounted after one week until no colonies were observed. In drug treatment experiments, the semisolid media were supplemented with the selected drug or vehicle and replenished with each plating. Human primary *DNMT3A*-R882 MNCs PB were plated at ~300,000 per well of a six-well plate of methylcellulose (Stem Cell Technologies, H4034) together with indicated concentrations of metformin/CTPI2 or vehicle control and counted at day 13.

## Human samples

Ethical approval for the human *DNMT3A*-R882 CH sample used in this study was granted by the East of England (Cambridge East) Research Ethics Committee (REC reference: 24/EE/0116). Informed consent was provided by the participant.

## Colony genotyping

Colonies were collected into DirectPCR Lysis Reagent (Viagen, 401-E) and processed according to the manufacturer's instructions. Lysed DNA was used for genotyping of *DNMT3A* with the following primers: F: 5′-CTGAGTGCCGGGTTGTTTAT-3′, R: 5′-GGAAGGGAGCTTGGTTTTGT-3′. PCR was performed with REDTaq ReadyMix PCR Reaction Mix (Sigma, R2523) with the following conditions: initial denaturation at 95 °C for 1 min, followed by 35 cycles of 95 °C for 15 s, annealing at 57 °C for 15 s and elongation at 72 °C for 15 s. Final elongation was performed at 72 °C for 10 min. The presence of *DNMT3A*-R882C generates further restriction sites for the AluI enzyme, and thus PCR products were subsequently digested with the restriction enzyme AluI (New England Biolabs, R0137S) for 1 h at 37 °C and visualized on 2.5% agarose gel.

## Lentiviral-vector production and transduction

Lentiviruses were produced in 293-FT (Invitrogen, R70007) cells with pMD2.G (Addgene, 12259) psPAX2 (Addgene, 12260) and the mouse v.2 whole-genome gRNA library[24] in a 1.5:0.5:1 ratio. Plasmid transfection was performed with Lipofectamine LTX (Invitrogen, 15338100) according to the manufacturer's instruction. Viral supernatants were concentrated by centrifugation at 6,000 rcf, 16 h, at 4 °C. The cells were transduced by spinoculation (60 min, 800*g*, 32 °C) in culture medium supplemented with polybrene (4 µg ml$^{-1}$; Sigma, TR-1003-G) and further incubated overnight at 37 °C. The media was fully changed the following day.

## gRNA cloning and competitive proliferation assay

gRNAs were cloned into a BbsI-digested pKLV2-U6gRNA(BbsI)PGK-puro2ABFP backbone[84]. Sequences of gRNAs used in the study are provided in Supplementary Table 10. For the competitive proliferation assay, the gRNA lentiviral vector was transduced into primary *Dnmt3a*^R882H/+, *Cas9* or WT, *Cas9* HSPCs with 50% of transduction efficiency, which was verified on day 4 post-transduction. Cells were then cultured and passaged three times per week. At each passage, the BFP versus non-BFP cell proportion was compared by flow cytometry.

## Bioenergetics analysis with Seahorse extracellular flux analyser

Cellular oxygen consumption was assessed using the Seahorse XF96 analyser according to the manufacturer's instructions. Briefly, 100,000–125,000 HSPCs were plated into a well of CellTak-coated XF96 cell culture microplates in XF DMEM medium (Agilent, 103575-100). For the Mito-stress test assay, the media was supplemented with 10 mM glucose (Sigma, G8644), 1 mM pyruvate (Agilent, 103578-100) and 2 mM glutamine (Agilent, 103579-100,). For the glycol-stress test assay, the media was supplement with 1 mM pyruvate and 2 mM glutamine. Cells were cultured in a non-CO2 incubator at 37 °C for 1 h. The oxygen consumption rate (OCR) and extracellular acidification rate were measured to assess mitochondrial and glycolytic activity. For the Mito-stress cell assay, OCR was measured in basal conditions, and the cells were then treated sequentially with 1 µM oligomycin (Sigma, 495455) and 2 µM carbonyl cyanide p-(trifluoromethoxy) phenylhydrazone (Sigma, C2920) followed by the addition of a final solution containing 0.5 µM rotenone (Sigma, R8875) and 0.5 µM antimycin A (Sigma, A8674). Glucose-stimulated respiration was measured by the sequential addition of 10 mM glucose, 2 µM oligomycin and 50 mM 2-Deoxy-D-glucose (Sigma, D8375). Drugs provided in manufactures' kits were also used in the study, including the Seahorse XF Cell Mito Stress Test Kit (Agilent, 103015-100) and Seahorse XF Glycolysis Stress Test Kit (Agilent, 103017-100). Results were analysed with Wave software (Agilent Technologies) and normalized to cell concentration. For metformin/CTPI2 treatment experiments, WT and *Dnmt3a*^R882H/+ HSPCs were plated in X-vivo20 media with cytokines and pretreated with selected inhibitors for 2 h. The drug was washed off, and cells were plated into CellTak-coated XF96 cell culture microplates in XF DMEM medium and processed for OCR analysis.

## Genome-wide CRISPR screen, sequencing and data analysis

HSPCs were extracted from two primary *Dnmt3*^R882H/+ mice and used for the screen. HSPCs were plated into M3434 for seven days and then cultured in vitro for another 10 days in X-VIVO 20 media (BE04-448Q, Lonza Bioscience) supplemented with 5% serum (Gibco, 10438-026), 10 ng ml$^{-1}$ IL3 (Peprotech, AF-213-13-1000), 10 ng ml$^{-1}$ IL6 (Peprotech, AF-216-16-1000), 50 ng ml ml$^{-1}$ SCF (Peprotech, AF-250-03-1000) and 1% penicillin–streptomycin–glutamine (Gibco, 12090216). On day 18, we transduced with the genome-wide library, choosing this time to reflect the clonal replating advantage observed with these cells in semisolid media culture (replating 3). For the screen, we used our murine v.2 lentiviral gRNA library containing 90,230 gRNAs against 18,424 genes, as described previously[24,84]. Then, $3 \times 10^7$ cells were transduced with a predetermined volume of the mouse v.2 genome-wide gRNA lentiviral supernatant that gave rise to a 30% transduction efficiency measured by BFP expression. Two days after transduction, BFP expression cells were flow-sorted and cultured for a total of 30 days. On day 6 post-transduction, 20% of the cells were collected for DNA extraction, and the remaining cells were replated in fresh X-vivo20 media with supplements. On day 10, 40% of the cells were collected for DNA extraction and the remaining cells replated. From day 15 to day 30, 50% of the cells were collected for DNA analysis and 50% replated in fresh media. Genomic DNA extraction and Illumina sequencing of gRNAs were described previously[24]. CRISPR sequencing was performed with the HiSeq2500 Illumina system: 19-bp single-end sequencing was performed with the custom sequencing primer 5′-TCTTCCGATCTCTTGTGGAAAGGACGAAACACC G-3′. Enrichment and depletion of guides and genes were analysed using the MAGeCK statistical package[85] by comparing read counts from each cell line with counts from the plasmid as the initial population. gRNA counts of CRISPR screen data are provided in Supplementary Data 1. CRISPP dropout data for human cancer cell lines were obtained from DepMap[86,87]. Gene set enrichment analyses were performed with the Enrichr online software[88]. Drug–gene interactions were investigated using the Drug Gene Interaction database (DGIdb)[89].

## Flow cytometry

The flow cytometry staining and gating strategy for progenitor and differentiated blood panels shown in Fig. 1c,d and Extended Data Fig. 1c–l, were reported previously[90] and are supplied in Supplementary Fig. 3. Briefly, BM cells, post erythrocyte lysis, were blocked with antimouse Cd16/32 (mouse BD FC block, BD Pharmigen, 553142, 1:500 dilution) and 10% mouse serum (Sigma, M5905) for LSK and common lymphoid progenitor (CLP) staining or 10% mouse serum alone for LK staining. LSK, MPP, lymphoid primed multipotent progenitor and LT/ST-HSC flow cytometry staining was performed with antibodies against CD4-PE-Cy5 (BioLegend, 100514, 1:800), Cd5-PE-Cy5 (BioLegend, 100610, 1:600), Cd8a-PE-Cy5 (BioLegend, 100710, 1:800), Cd11b-PE-Cy5 (BioLegend, 101210, 1:400), B220-PE-Cy5 (BioLegend, 103210, 1:200), Ter119-PE-Cy5 (BioLegend, 116210, 1:300), Gr-1-PE-Cy5 (BioLegend, 108410, 1:400), Sca1-PB (BioLegend, 122520, 1:100), c-Kit-APC-Cy7 (eBioscience, 47-1171, 1:200), Cd48-APC (BioLegend, 103411, 1:150), Cd150-PE-Cy7 (BioLegend, 115913, 1:100), Cd34-FITC (BD Pharmigen, 553733, 1:50) and Flt3-PE (eBioscience, 12-1351, 1:50). Granulocyte-monocyte progenitor, MEP and common myeloid progenitor staining was performed with the following biotin-conjugated lineage markers: Mac1, Gr1, Cd3, B220, Ter119 (BD Pharmigen, 559971, 1:300 each), Il7Ra (BioLegend, 121103, 1:300) and streptavidin-PE-Cy7 (BioLegend, 405206, 1:300) alongside Cd34-FITC (BD Pharmigen, 553733, 1:50), Cd16/32-PE (BD Pharmigen, 553145, 1:50), c-Kit-APC (BioLegend, 105812, 1:100) and Sca1-PB (BioLegend, 122520, 1:100). For the detection of the CLP population, cells were stained for Flt3-PE (eBioscience, 12-1351, 1:50), IL7Ra-FITC (BioLegend, 135008, 1:50) and the lineage biotin-conjugated markers Mac1, Gr1, Cd3, B220, Ter119 (BD Pharmigen, 559971) and NK (LSBio, LS-C62548, 1:300) and streptavidin-PE-Cy7 (BioLegend, 405206, 1:300) alongside c-Kit-APC (BioLegend, 105812, 1:100) and Sca1-PB (BioLegend, 122520, 1:100). Differentiated BM, spleen and PB cells were stained with CD11b-APC-Cy7 (BD Pharmigen, 557657, 1:300), B220-PE-Cy5 (BioLegend, 103210, 1:200), Gr1-BV450 (BD Pharmigen, 560603, 1:600) and CD3e-PE (eBioscience, 12-0031-82, 1:300). LT-HSC were defined as Lin⁻c-Kit⁺Sca1⁺Flt3⁻Cd48⁻Cd150⁺CD34⁻. ST-HSC were defined as Lin⁻c-Kit⁺Sca1⁺Flt3⁻Cd48⁻Cd150⁺CD34⁺. MPPs were defined as Lin⁻c-Kit⁺Sca1⁺Flt3⁺. Lymphoid primed multipotent progenitors were defined as Lin⁻c-Kit⁺Sca1⁺Flt3^high. CLPs were defined as Lin⁻Flt3^hiIl7Ra⁺c-Kit^loSca1^lo. Granulocyte-monocyte progenitors were defined as Lin⁻Il7Ra⁻c-Kit⁺Sca1⁻Cd34⁺Cd16/32⁺. Common myeloid progenitors were defined as Lin⁻Il7Ra⁻c-Kit⁺Sca1⁻Cd34⁺Cd16/32⁻. MEPs were defined as Lin⁻Il7Ra⁻c-Kit⁺Sca1⁻Cd34⁻Cd16/32⁻.

For analysis of progenitor compartment in transplant experiments, freshly isolated BM cells post erythrocyte lysis were washed and stained with the following antibodies: biotin-conjugated antibody for B220, Ter119, Mac1, Cd3, Gr1 (BD Biosciences, 559971, 1:300 each), detected with Streptavidin-BV510 (BioLegend, 405233, 1:900), as well as Sca1-PB (BioLegend, 122520, 1:100), c-Kit-AF780 (Thermo Fisher Scientific, 47-1171-82, 1:100), Cd48-BV605 (BioLegend, 103441, 1:200), Cd150-BV650 (BioLegend, 115932, 1:100), Cd135-PE (BD Biosciences, 553842, 1:100), Cd34-FITC (Thermo Fisher Scientific, 11-0431-82, 1:50), Cd16/32 PerCP-Cy5.5 (BioLegend, 101324, 1:100), CD45.1-APC (BioLegend, 110714, 1:100), CD45.2-PE-Cy7 (Thermo Fisher Scientific, 25-0454-82, 1:100) and 7AAD (BioLegend, 420404, 1:100). LT-HSC were defined as 7AAD⁻Lin⁻c-Kit⁺Sca1⁺Cd48⁻Cd150⁺, ST-HSC as 7AAD⁻Lin⁻c-Kit⁺Sca1⁺Cd48⁻Cd150⁻, MPP2: ST-HSC as 7AAD⁻Lin⁻c-Kit⁺Sca1⁺Cd48⁺Cd150⁺, MPP2: MPP3 as 7AAD⁻Lin⁻c-Kit⁺Sca1⁺Cd48⁺Cd150⁻ and LSK as 7AAD⁻Lin⁻c-Kit⁺Sca1⁺. The gating strategy is shown in Supplementary Fig. 4a. CD45.1 and CD45.2 markers were used to distinguish injected cells in each of the gated populations. For differentiated blood panel staining in PB, BM and spleen post-transplantation, cells were stained with the following antibodies: CD45.1-BV605 (BioLegend, 110738, 1:200), CD45.2-FITC (BioLegend, 109806, 1:200), Cd4-PE-Cy5 (BioLegend, 100514, 1:400), Cd8-PE-Cy5 (BioLegend, 100710, 1:400), B220-PE-Cy5 (BioLegend, 103210, 1:400), Gr1-AF700 (BioLegend, 108422, 1:400), Mac1-AF700 (BioLegend, 101222, 1:400) and B220-AF700 (BioLegend, 103232, 1:400). B cells were defined as B220-PE-Cy5⁺B220-AF700⁺, T cells as Cd4- Cd8-PE-Cy5⁺AF700⁻ and myeloid cells as Mac1-Gr1-AF700⁺ PE-Cy5⁻. The gating strategy is shown in Supplementary Fig. 4b. Flow cytometry analysis was performed with an LSRFortessa instrument (BD) and BD FACSDiva Software, and data were analysed in FlowJo software. Figures 1h and 4c show fluorescence-activated cell sorting (FACS) pseudocolour smooth plots, and Supplementary Fig. 4 shows pseudocolour plots.

## Transplantation

BM was isolated from sex/age-matched CD45.2-*Dnmt3a*^R882H/+ or CD45.2-WT post-pIpC and CD45.1-WT, and erythrocytes were lysed with 0.85% ammonium chloride. For competitive transplantation, CD45.2 BM was mixed with CD45.1 BM in a 1:2 proportion and $1 \times 10^6$ cells injected intravenously into lethally irradiated ($2 \times 5.5$ Gy) CD45.1/2 recipient mice. For non-competitive transplant, $1 \times 10^6$ *Dnmt3a*^R882H/+ or WT BM cells were injected intravenously into lethally irradiated CD45.1/2 recipient mice.

## Drug treatment in vivo and in vitro

CTPI2 (2-(4-Chloro-3-nitro-benzenesulfonylamino)-benzoic acid; sc-339832) was administered at 40 mg kg⁻¹ by means of intraperitoneal injection three times per week for two weeks. CTPI2 preparation was reported previously[28]. Briefly, CTPI2 was prepared in 0.47% sodium bicarbonate (NaHCO₃, Biochrom, L-1703) at a final concentration of 11.2 mM; 0.47% NaHCO₃ was used as vehicle control. For in vitro studies, CTPI2 was dissolved either in 0.47% NaHCO₃ or in DMSO.

IACS-010759 (Stratech Scientific, S8731-SEL-100mg) was prepared in 0.5% methylcellulose (Sigma, M0262) as reported before[29]. Briefly, the drug was dissolved in 0.5% methylcellulose solution and sonicated for 8–15 cycles (30 s on, 30 s off) in a Bioruptor Pico instrument (Diagenode), followed by homogenization at 5,000 rpm for 3 min using an IKA Ultra-Turrax homogenizer. IACS-010759 was applied by oral gavage daily (five days on, two days off) at dose 8 mg kg⁻¹; 0.5% methylcellulose was used as vehicle control. For in vitro studies, IACS-010759 was prepared in DMSO.

Metformin hydrochloride (Sigma, PHR1084 or TRC-M258815-100G) was dissolved in PBS and administrated at dose 125 mg kg⁻¹ by means of intraperitoneal injection daily (five days on, two days off) as previously reported[32]. Treatment was performed for six weeks in total. For in vitro studies, metformin was prepared in sterile water. A dose of 125 mg kg⁻¹ per day, when translated to human equivalents using the conversion coefficient proposed by ref. 91, corresponds to 10.4 mg kg⁻¹ per day. Notably, this pharmacological dosage is lower than the levels commonly prescribed for the treatment of T2D.

## Dot blots for 5mC and 5hmC

Genomic DNA was extracted with a Puregene Cell Kit ($8 \times 10^8$) (Qiagen, 158043). The DNA was denatured and spotted onto a Hybond NX membrane (Amersham Biosciences, RPN303 T). The membrane was air dried for 15 min and incubate in 2× saline-sodium citrate buffer for 5 min. The membrane was air dried for 30 min, wrapped in cling film (Bakewell) and UV-crosslinked using the automatic setting on a Stratagene UV Stratalinker 2400 (120,000 µJ cm⁻² for 150 s). The membrane was then blocked with 1% BSA and 5% skim milk in PBS containing 0.2% Tween (PBST) for 1 h at room temperature and washed three times for 5 min with PBST. For 5hmC detection, the membrane was incubated with anti-5hmC Rb antibody (Active Motif, 39769, 1:10,000 dilution) in 3% BSA in PBST overnight at 4 °C. For 5mC detection, the membrane was incubated with anti-5mC Ms antibody (Active Motif, 61479, 1:2 000). The membrane was washed three times for 5 min with PBST and incubated in anti-rabbit (Jackson ImmunoResearch Laboratories Inc., 111-035-003) or anti-mouse (Jackson ImmunoResearch Laboratories Inc.,

115-035-146) HRP-conjugated secondary antibody (diluted 1:20,000 in 1% BSA and 5% skim milk in PBST) for 1 h at room temperature. The membrane was then washed three times for 5 min with PBST and imaged using a SuperSignal West Dura Extended Duration Substrate (Thermo Scientific, 37071). For total DNA detection, the membrane was washed with water and incubated in a 0.1% methylene blue solution containing 0.5 M sodium acetate, pH 5.2 overnight at room temperature. The membrane was then washed three times for 5 min with water and air dried for 30 min. The membrane was scanned and analysed using Image Studio Lite software. Uncropped dot blots and gel images are available in Supplementary Figs. 5–7.

### LC–MS analysis of aqueous metabolites
Liquid chromatography–mass spectrometry (LC–MS) was used to measure metabolites in aqueous extracts from cells. To this end, a Q Exactive Plus orbitrap mass spectrometer coupled to a Vanquish Horizon ultra-high performance liquid chromatography system (both Thermo Fisher Scientific) was used. Customized methods were used to separate and detect distinct metabolites of interest, as detailed below.

### LC–MS sample preparation
Cells were collected into 2-ml tubes, spun down and washed twice with cold PBS. The cell pellet was then resuspended in 500 µl 4:1 methanol water. Aqueous metabolites were extracted from cells using a method modified from ref. 92. Briefly, 800 µl chloroform (Fisher Scientific, 15643700) and 380 µl Milli-Q water was added to cells suspended in 500 µl 4:1 methanol water in 2 ml tubes. This was followed by vortexing, sonication and centrifugation for 5 min at $21,300g$. The top, aqueous phase was then collected from each sample into eppendorf tubes and subsequently dried using an SC210A SpeedVac vacuum centrifuge (Savant, Thermo Fisher Scientific). Dried metabolite extracts were reconstituted in 10 mM ammonium acetate (Fisher Chemical, 10598410) LC sample buffer or derivatized with diacetyl-L-tartaric anhydride (DATAN, Merck, 336040050) for enantiomer analysis (see 'L- and D-2-HG enantiomer analysis') as described previously[93]. When necessary, further internal standards were added to the LC sample buffer: universal $^{15}$N$^{13}$C amino acid mix, succinate $^{13}$C$_4$, AMP $^{15}$N$_5$$^{13}$C$_{10}$, ATP $^{15}$N$_5$$^{13}$C$_{10}$, putrescine D$_8$, dopamine D$_4$.

### LC–MS measurement of 2HG
An ACE Excel C18-PFP column (150 × 2.1 mm, 2.0 µm, Avantor, EXL-1010-1502U) was used to separate species such as TCA cycle intermediates, amino acids and related compounds, as previously[94]. For positive ion mode analyses, mobile phase A consisted of water with 10 mM ammonium formate (Merck, 70221-100G-F) and 0.1% formic acid (Optima grade, Fisher Chemical, 10596814); and for negative ion mode analyses, mobile phase A consisted of water with 0.1% formic acid. Mobile phase B was acetonitrile (Chromasolv, Honeywell, 34851-2.5L) with 0.1% formic acid for both positive and negative ion modes. The C18pfp LC gradient used is detailed in Supplementary Table 11a. The flow rate was 0.5 ml min$^{-1}$, and the injection volume was 3–3.5 µl. The needle wash used was 1:1 water to acetonitrile.

Source parameters used for the orbitrap were an auxiliary gas heater temperature of 450 °C, a capillary temperature of 275 °C, an ion spray voltage of 3.5 kV (2.5 kV for negative ion mode) and a sheath gas, auxiliary gas and sweep gas of 55, 15 and 3 arbitrary units, respectively, with an S-lens radio frequency of 50%. A full scan of 60–900 $m/z$ was used at a resolution of 70,000 ppm in positive ion mode and a full scan of 55–825 $m/z$ at a resolution of 140,000 ppm in negative ion mode.

### L- and D-2-HG enantiomer analysis
An Acquity Premier HSS T3 column (1.8 µm, 2.1 × 100 mm, Waters, 186009468) was used for metabolite extracts derivatized with DATAN, for the purpose of measuring L and D enantiomers of 2-HG, as previously[93]. An internal standard of U-13C algal lyophilized cells

(CK Isotopes, CLM-2065-1) was added to each sample before drying and derivatization; 10 µl of 0.5 µg µl$^{-1}$ was added to each sample post-Folch extraction. Standards of L- and D-2-HG (Merck, 90790-10MG and H8378, respectively) were prepared (100 µM in Milli-Q water), to confirm the retention times of L- and D-2-HG. These standards (100 µl of each) were dried, derivatized and run in parallel alongside the samples. Mobile phase A was 1.5 mM ammonium formate (to pH 3.6 with formic acid), and mobile phase B was acetonitrile with 0.1% formic acid. The LC gradient used is detailed in Supplementary Table 11b. The flow rate was 0.4 ml min$^{-1}$, and the injection volume was 5 µl for samples and 1 µl for L- and D-2-HG standards. The needle wash used was 1:1 water to acetonitrile.

Source parameters used for the orbitrap were as for the C18pfp method. In addition, parallel reaction monitoring of the transitions $m/z$ 363.0569 (2-HG + DATAN) to 147.0299 (2-HG) and $m/z$ 368.0737 (2-HG $^{13}$C$_5$ + DATAN) to 152.0467 (2-HG $^{13}$C$_5$) was performed with a collision energy of 20 in negative ion mode.

### LC–MS data processing
Targeted analysis of LC–MS data was performed using Thermo Scientific Xcalibur (Quan Browser and Qual Browser) as previously[94]. Peak areas were normalized to an appropriate internal standard where possible. For 2HG, area ratios to $^{13}$C$^{15}$N glutamate internal standard were calculated. For L- and D-2HG enantiomer analysis, the $m/z$ of 147.0299 (2-HG) was extracted as previously[93], in addition to the $m/z$ of 152.0467 for the 2-HG $^{13}$C$_5$ internal standard. Area ratios of L- and D-2-HG to respective $^{13}$C$_5$ labelled L- and D-2HG were calculated.

### TCGA-AML data analysis
Gene expression and survival data of the TCGA-AML cohort (151 samples) were downloaded from the GDA data portal[95]. To plot the Kaplan–Meier curves, samples were divided into two groups according to the gene's expression level: high-expression group, expression levels above the median; and low expression group, expression below the median. Survival curves and statistics were generated by the 'survival' package in R[96].

### UKB analysis
UKB is a prospective cohort of about 500,000 adults, aged between 40 to 70 years and recruited between 2006 and 2010[41,97]. We identified 419,389 individuals of genetically determined European ancestry from the UKB with whole-exome sequencing data passing quality control criteria[98]. Individuals with CH were identified by somatic variant calling of established driver mutations, as previously described[72,98]. Among the 419,226 individuals with CH data available, we retained 416,118 individuals without prerecruitment blood cancer diagnosis. Eight hundred and forty-six type 1 diabetes individuals were identified as previously described[99] and were removed. Finally, 412,234 individuals with complete covariate data were included for analysis. Diabetes status and intake of antidiabetic medications (metformin, sulphonylureas, insulin, thiazolidinediones, meglitinides and acarbose) were ascertained from self-reported data. More individuals on metformin were ascertained from general practitioner records. In total, 11,190 individuals were identified as metformin users at recruitment and 19,287 individuals were identified as diabetic at recruitment, of whom 11,001 (57.0%) were on metformin, consistent with a recent study[100]. Circulating HbA1c levels were measured for each UKB participant at recruitment and used to identify undiagnosed diabetics (HbA1c ≥ 7%). We used logistic regression to model the association between antidiabetic medication at recruitment (predictor) and overall CH or gene-specific CH (outcome) and included age, sex, smoking and the first four genetic principal components as covariates. Metformin, sulphonylureas (glibenclamide, gliclazide, glipizide, glimepiride, tolbutamide) and insulin were included as antidiabetic medications in the association analysis. The UKB study has approval from the North-West Multi-centre Research Ethics Committee (11/NW/0382). The present study has been conducted under approved UKB application numbers 56844 and 26041. As we did not find

an association between metformin use and *DNMT3A* non-R882 CH, we classified *DNMT3A* non-R882 mutations into high- and low-functionality variants as determined by ref. 40 and assessed the association between metformin and *DNMT3A* variants stratified by functionality.

## MR analysis

For polygenic MR analysis, independent germline genetic variants associated with HbA1c, BMI, T2D and BMI-adjusted waist-to-hip ratio were retrieved from previous publications that included data from the UKB[42–45]. For HbA1c, we also used the instruments from ref. 42 with independent betas from the Meta-Analyses of Glucose and Insulin-related traits Consortium[43] (two-sample MR). Genetic associations using these genetic instruments with CH as the outcome were performed using Firth logistic regression implemented by REGENIE software[101], assuming an additive effect, adjusted for age, sex and the first 10 genetic principal components. MR analyses were performed using the TwoSampleMR v.0.5.6 R package[102,103] (valid in both one-sample and two-sample settings[104]), with glycaemic-related traits as the exposures and CH as the outcome, and the test statistics reported were derived from inverse variance weighting. MR summary statistics are provided in Supplementary Table 8.

## Illustrations and graphs

Schematic illustrations were made with BioRender under University of Cambridge – Central License's Plan. The image of a seahorse in Fig. 3c and Extended Data Fig. 4g was hand-drawn by Kazimierz Gozdecki. Data were analysed and plotted using GraphPad Prism v.10. IGV software[105] was used to visualize the data shown in Fig. 1b.

## Statistical analysis

All statistical analyses were performed with two-sided Student's *t*-test, log-rank (Mantel–Cox), one-way analysis of variance (ANOVA), two-way ANOVA or Chi-square test, as specified in figure legends. Samples were tested for normal distribution before statistical analysis. Error bars represent the s.e.m. or the s.d. *P* values ≤ 0.05 were considered statistically significant. Representative data/images were replicated as specified in the relevant figure legend.

## Reporting summary

Further information on research design is available in the Nature Portfolio Reporting Summary linked to this article.

## Data availability

CRISPR screen raw data have been deposited in the Gene Expression Omnibus with accession number GSE259404. Whole-exome and targeted sequencing data are available in Sequence Read Archive BioProject PRJNA1160274. Metabolomic data were deposited into the MetaboLights depository (https://www.ebi.ac.uk/metabolights/) with accession number MTBLS12201. TCGA-AML cohort (151 samples) were downloaded from the GDA data portal: https://portal.gdc.cancer.gov/projects/TCGA-LAML. For UKB, individual-level data are under controlled accessed to protect sensitive information of the study participants. Individual-level UKB data may be requested by means of application to the UKB. All whole-exome sequencing data described in our study are available to registered researchers through the UKB data access protocol. Exomes can be found in the UKB showcase portal: https://biobank.ndph.ox.ac.uk/showcase/label.cgi?id=170. Additional information about data access registration is available at https://www.ukbiobank.ac.uk/enable-your-research/register. The DepMap 19Q4 Public dataset is available at Figshare (https://figshare.com/articles/dataset/DepMap_19Q4_Public/11384241/3)[106]; the Meta-Analyses of Glucose and Insulin-related traits Consortium dataset is available at http://magicinvestigators.org/; and the DGIdb database is available at https://dgidb.org/. Source data are provided with this paper.

## Code availability

The Survival analysis R package v.3.5-8 is available from the CRAN (https://CRAN.R-project.org/package=survival) and the Enrichr online software is available at https://maayanlab.cloud/Enrichr/. Data processing and analyses were conducted using publicly available code and software, as outlined in the Methods and Reporting Summary.

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

**Acknowledgements** This study is funded by a Blood Cancer UK Grant (grant no. 21006) to B.J.P.H. and G.S.V., a joint Blood Cancer UK – Leukemia & Lymphoma Society Specialized Centre of Research Grant (grant no. 7035-24), a Cancer Research UK Early Detection Grant (grant no. EDDCPJT\100010) and a Wellcome Trust grant (grant no. WT098051) to G.S.V. G.S.V. is funded by a Cancer Research UK Senior Cancer Fellowship (grant no. C22324/A23015), and work in his lab is also funded by the European Research Council, Kay Kendall Leukaemia Fund, Blood Cancer UK and the Wellcome Trust. Work in B.J.P.H.'s laboratory is funded by Cancer Research UK (grant nos. C18680/A25508 and DRCRPG847-Nov22/100014) and the Medical Research Council (grant no. MR-X008371). This research was supported by the NIHR Cambridge Biomedical Research Centre (grant nos. BRC-1215-20014 and NIHR203312) and the Cancer Research UK Cambridge Centre (Cancer Research UK Major Centre Award nos. C9685/A25117 and CTRQQR-2021\100012) and was funded in part by the Wellcome Trust, who supported the Cambridge Stem Cell Institute (grant nos. 203151/Z/16/Z and 226795/Z/22/Z). The views

expressed are those of the authors and not necessarily those of the NIHR or the Department of Health and Social Care. For the purpose of Open Access, the author has applied for a CC BY public copyright licence to any Author Accepted Manuscript version arising from this submission. A.K. is supported by Wellcome Trust investigator grant no. 222497/Z/21/Z and Leona M & Harry B Helmsley Charitable Trust grant no. R-2408-07256. P.M.Q. is supported by the Ramon y Cajal Program (grant no. RYC2022-036793-I), funded by MICIU/AEI/10.13039/501100011033 and cofunded by FSE+, and his work is funded by ISCIII (grant no. PI22/00218), cofunded by the EU. C.B. is supported by the Ramon y Cajal program (RYC2021-031291-I) funded by MICU/AEI/10.13039/50100011033 and cofunded by European Union NextGenerationEU/PRTR. K.T. is supported by Wellcome Trust (grant nos. RG83195, G106133, G127005 and G127005), UKRI Medical Research Council (grant no. RG83195) and Leukaemia UK (grant no. G117699). E.Y. is supported by Leukaemia UK John Goldman Fellowship (grant no. G127956). We thank M.L. Avantaggiati and her team for their guidance on CTPI2 preparation for in vivo study. We thank J.R. Marszalek and his team for detailed information on IACS-010759 preparation for in vivo study. We thank the Cytometry Core Facility at the Wellcome Sanger Institute for their help with FACS sorting experiments and the Anne McLaren Biomedical Research Facility for their help with mouse experiments. We thank K. Gozdecki for hand-drawing selected illustrations for the manuscript.

**Author contributions** M. Gozdecka, B.J.P.H. and G.S.V. conceived the study, designed experiments and prepared the manuscript. M. Gozdecka and M.D. conducted most of the experiments. S.W., M.A.F., J.M. and S.P. performed UKB data analysis. M. Gu, V.M.S., P.C., P.M.Q., S.K., T.G. and W.G.D. performed bioinformatic analysis and data deposition. R.J.S. performed LC–MS and data analysis. J.R., A.D., E.Y., L.B., M.A.M., R.B.B., G.L.G., S.J.H., J.L.C., N.N., R.A., D.C.P., P.G., L.M., A.P.R., R.W., G.G., C.B., K.T., C.F., J.A.N., A.K. and G.S.V. contributed to in vitro and in vivo experiments and provided scientific expertise. A.M. and G.D. contributed to the generation of the mouse model. P.W., G.S.V. and B.J.P.H. conducted histological diagnosis. K.Y. conducted CRISPR analysis. All authors reviewed and agreed with the final submission.

**Competing interests** G.S.V. is a consultant for STRM.BIO and receives a research grant from AstraZeneca. S.W., J.M. and M.A.F. are current employees and/or stockholders of AstraZeneca. K.T. has received consultancy fees, stock options and research funding from Storm Therapeutics Ltd., Cambridge, UK. K.T. is a cofounder of Btwo3 Therapeutics LLC, USA and TEP Therapeutics LLC, USA. The other authors declare no competing interests.

**Additional information**
**Correspondence and requests for materials** should be addressed to Malgorzata Gozdecka, Brian J. P. Huntly or George S. Vassiliou.

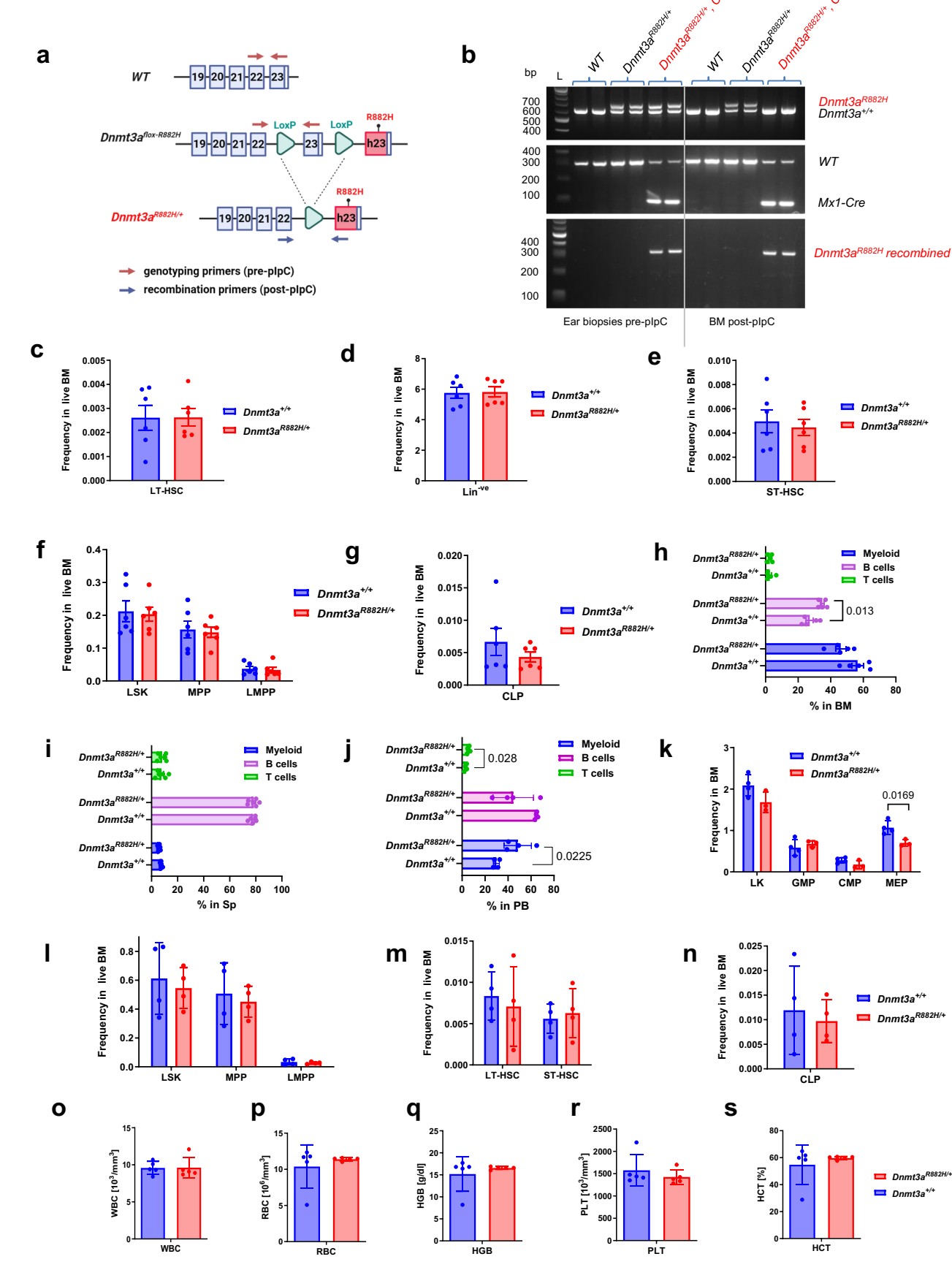

**Extended Data Fig. 1** | See next page for caption.

**Extended Data Fig. 1 | Characterisation of *Dnmt3a*^R882H/+ mouse model.**
(**a**) Schematic representation of genotyping (red arrows) and recombination primers (blue arrows) binding to WT, *Dnmt3a*^flox-R882H and *Dnmt3a*^R882H alleles. (**b**) Genotyping confirmed the presence of the *Dnmt3a*^R882H allele pre-pIpC in both *Dnmt3a*^R882H/+ and *Dnmt3a*^R882H/+ *Cre* genotypes (top panel, left half). After Cre induction with pIpC the *Dnmt3a*^R882H allele completely disappears from *Dnmt3a*^R882H/+ *Cre* mice, indicating complete recombination in BM (top panel, write half). Mid panel illustrates the presence of the *Mx1-Cre* allele in *Dnmt3a*^R882H/+ *Cre* genotypes. Lower panel shows recombination in *Dnmt3a*^R882H/+, *Cre* mice post-pIpC. A low level of recombination is also observed in *Dnmt3a*^R882H/+, *Cre* mice pre-pIpC (lower panel, left half). However, the full recombination was not observed pre-pIpC *Dnmt3a*^R882H/+ *Cre*, indicated by the presence of *Dnmt3a*^R882H allele (top panel, left half). (**c**) Frequency of LT-HSC, (**d**) Lin-ve, (**e**) ST-HSC, (**f**) LSK, MPP, LMPP and (**g**) CLP progenitors in BM of *Dnmt3a*^R882H/+ and WT mice 6-weeks post-pIpC, $n = 6$ mice per group in c-g (**h**) Proportion of B, T and myeloid cells in BM, $n = 5$ mice per group and (**i**) Sp of *Dnmt3a*^R882H/+ and WT mice 6-weeks post-pIpC, $n = 6$ mice per group. The mean ± SEM is shown in c-i (**j**) Proportion of myeloid, B and T cells in PB, $n = 4$ mice per group. (**k**) Frequencies of LK, GMP, CMP, MEP, $n = 4$ mice for WT and $n = 3$ mice for *Dnmt3a*^R882H/+ (**l**) Frequencies of LSK, MPP, LMPP (**m**) LT-HSC, ST-HSC and (**n**) CLP in BM of *Dnmt3a*^R882H/+ and WT mice 1-year post-pIpC for j-*n*. The mean ± SD is shown, $n = 4$ mice per group for (**l-n**). (**o**) WBC, (**p**) RBC, (**q**) HGB, (**r**) PLT, (**s**) HCT of *Dnmt3a*^R882H/+ and WT mice 20 weeks post-pIpC, $n = 5$ mice per group for (**o-s**). *P* in (**h, j, k**) by two-sided t-test. MPP, multipotent progenitors; LMPP, lymphoid primed multipotent progenitors; ST-HSC, short-term HSC; GMP, granulocyte-monocyte progenitors; CMP, common myeloid progenitors, MEP megakaryocyte-erythroid progenitors; CLP, common lymphoid progenitors; RBC, red blood cells. Schematic in **a** was created using BioRender (https://biorender.com).

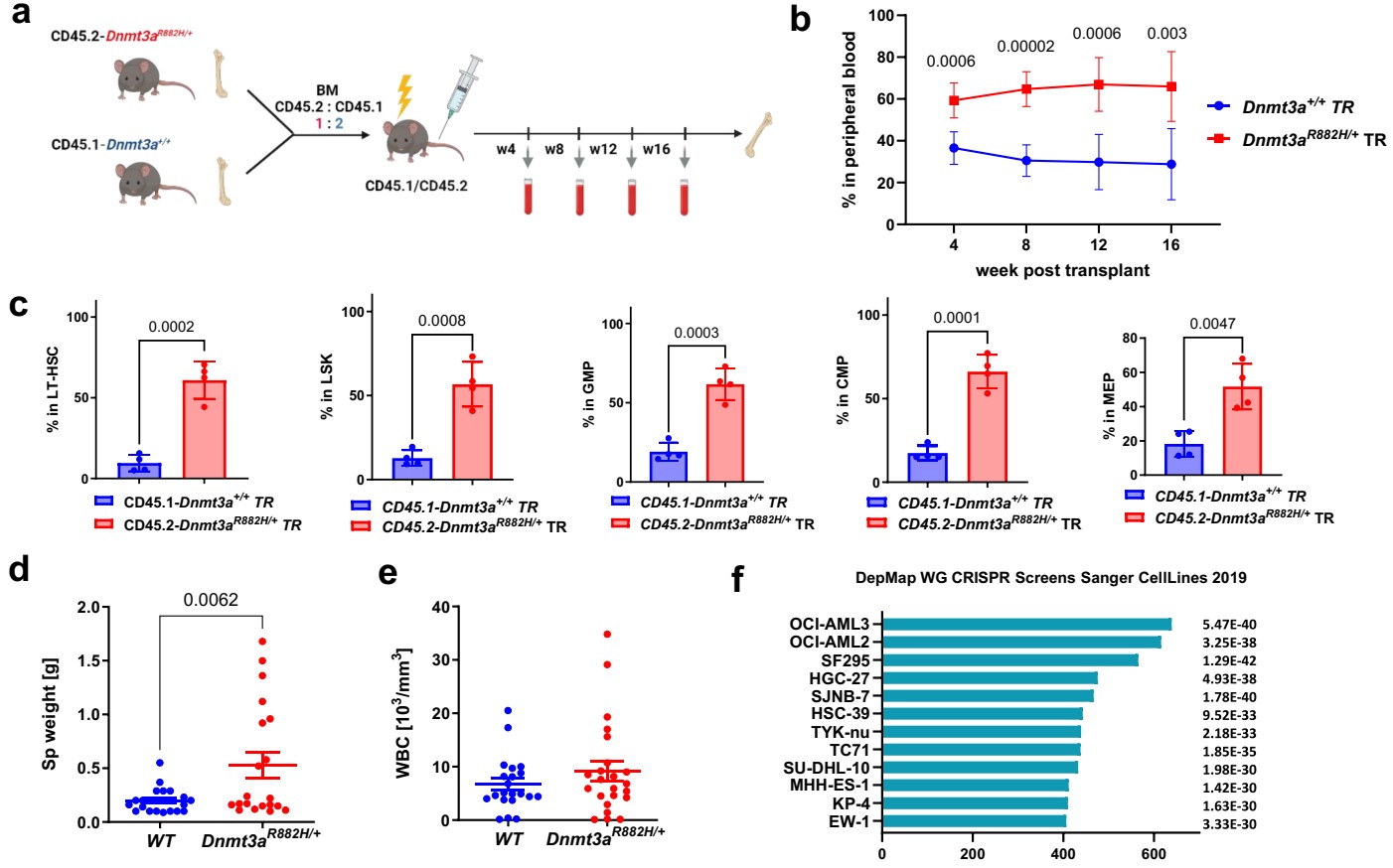

**Extended Data Fig. 2 | Enhanced competitive BM repopulation of** ***Dnmt3a***[R882H/+]**.** (**a**) Schematic representation of the experimental approach for *Dnmt3a*[R882H/+] (CD45.2) versus WT (CD45.1) competitive transplant. (**b**) Proportion of *Dnmt3a*[R882H/+] and WT cells in PB of recipients post-transplant. The mean ± SD is shown, *n* = 8 mice, *P* by two-sided t-test. *P* displayed for *Dnmt3a*[R882H/+] TR vs WT TR comparison. (**c**) Proportion of *Dnmt3a*[R882H/+] and WT cells in LT-HSC, LSK, GMP, CMP, MEP in BM at end point. The mean ± SD is shown, *n* = 4 mice, *P* by two-sided t-test. (**d**) Spleen weights *n* = 23 for WT and

*n* = 20 for *Dnmt3a*[R882H/+] mice and (**e**) WBC from WT (*n* = 21) and *Dnmt3a*[R882H/+] (*n* = 23) mice at aging end point. The mean ± SEM is shown, *P* by two-sided t-test. (**f**) *Dnmt3a*[R882H/+] vulnerability genes are highly enriched for dropouts observed in human *DNMT3A*-mutant cell lines. Enrichment analysis was performed with the Enrichr online software. Schematic in **a** was created using BioRender (https://biorender.com). GMP, granulocyte-monocyte progenitor; CMP, common myeloid progenitor.

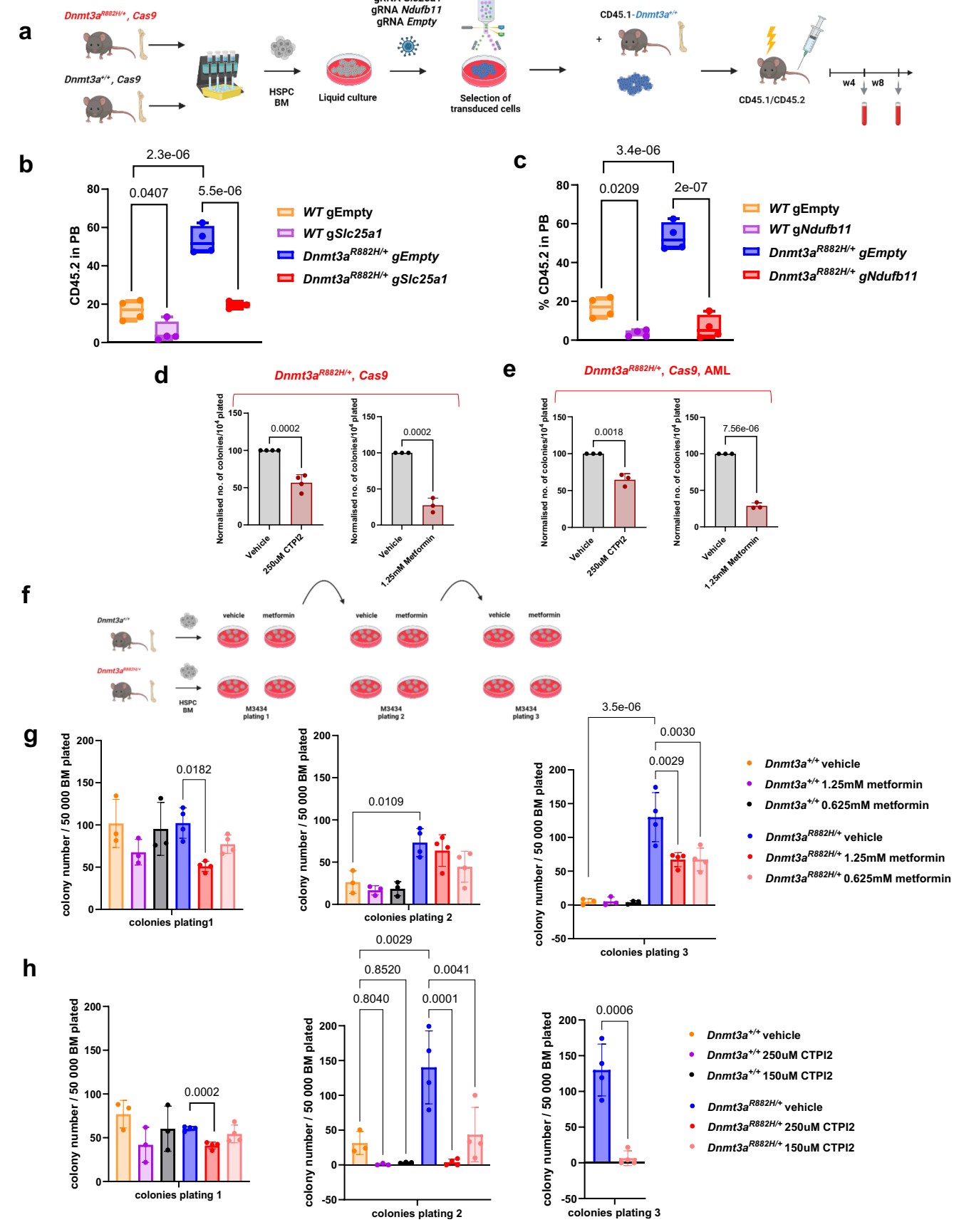

**Extended Data Fig. 3 | Effect of selected inhibitors on $Dnmt3a^{R882H/+}$ self-renewal potential.** (**a**) Experimental schema used to determine the impact of gene knockout on competitive repopulation potential of $Dnmt3a^{R882H/+}$ vs $Dnmt3a^{+/+}$ HSPCs (**b-c**) Proportion of CD45.2 (WT and $Dnmt3a^{R882H/+}$) cells in PB upon (**b**) *Slc25a1* and (**c**) *Ndufb11* knockout. $n = 4$ mice per group, box-and-whiskers plots displaying median and interquartile range (IQR), with whiskers to min and max values. *P* by one-way-ANOVA with Tukey correction. (**d**) Colony formation of $Dnmt3a^{R882H/+}$, $n = 4$ mice per condition in left panel and $n = 3$ mice per condition for right panel, and (**e**) leukaemic $Dnmt3a^{R882H/+}$ HSPCs plated in the presence of selected inhibitors. Colony number was normalised to vehicle control. The mean ± SD is shown, $n = 3$ mice, *P* by two-sided t-test. (**f**) Schematic representation of colony replating strategy. (**g**) Colony number at each plating during metformin and (**h**) CTPI2 treatment. $n = 3$ mice for WT and $n = 4$ for $Dnmt3a^{R882H/+}$ in each condition in g-h, the mean ± SD is shown, *P* by one way ANOVA, and in h plating 3 by two-sided t-test. Schematics in **a**,**f** were created using BioRender (https://biorender.com).

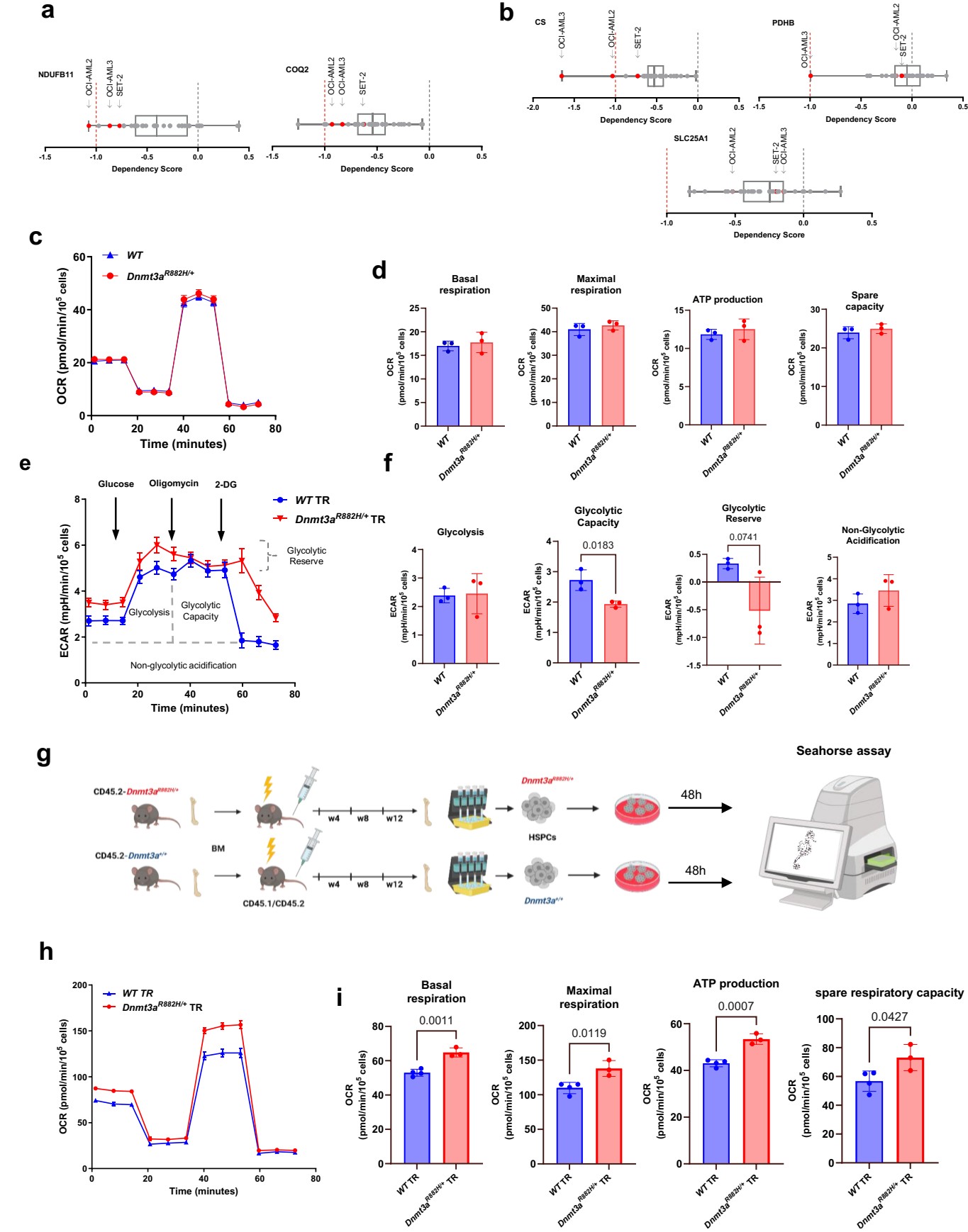

**Extended Data Fig. 4** | See next page for caption.

**Extended Data Fig. 4 | Cellular respiration in homoeostasis and transplantation setting.** (**a**) CRISPR dependency scores for *NDUFB11*, *COQ2*, (**b**) Similar score for *CS*, *PDHB*, *SLC25A1* across all myeloid cancer cell lines. Arrows identify the *DNMT3A*-mutated cell lines. (a-b) box-and-whiskers plots displaying median and interquartile range (IQR), with whiskers to min and max values. Data for a-b were obtained from DepMap, $n$ = 37 cell lines. (**c**) Oxygen consumption rate in primary WT and *Dnmt3a*[R882H/+] HSPCs, was measured using a Seahorse analyser, mean ± SEM is shown. $n$ = 11 WT and $n$ = 12 *Dnmt3a*[R882H/+], representing 3 independent biological replicates per genotype each performed in 4 replicates for *Dnmt3a*[R882H/+] and in 3, 4, and 4 replicates for WT. One of three independent experiments are shown. (**d**) Basal respiration, maximal respiration, ATP-production, and spare respiration capacity; mean ± SD is shown. $n$ = 3 biological replicates per genotype. (**e**) The extracellular acidification rate (ECAR) was measured in transplanted WT and *Dnmt3a*[R882H/+] HSPCs by the Seahorse analyser, mean ± SEM is shown. $n$ = 7 for WT and $n$ = 18 *Dnmt3a*[R882H/+], 3 biological replicates per genotype were performed in

6 replicates for *Dnmt3a*[R882H/+], and 3, 3 and 1 replicate for WT. One of three independent experiments are shown. (**f**) Glycolysis, glycolytic capacity, glycolytic reserve, and non-glycolytic acidification were compared between transplanted WT and *Dnmt3a*[R882H/+] HSPCs. Mean ± SD is shown. $P$ by two-sided t-test. $n$ = 3 biological replicates per genotype. (**g**) Schematic representation of the experimental setup. WT and *Dnmt3a*[R882H/+] BM were transplanted separately into recipient mice. HSPCs were extracted 12 weeks post-transplantation, cells were plated in culture for 48 h and analysed with Seahorse analyser. (**h**) OCR in transplanted WT and *Dnmt3a*[R882H/+] HSPCs, was measured using a Seahorse extracellular flux analyser, mean ± SEM. $n$ = 24 for WT and $n$ = 18 for *Dnmt3a*[R882H/+], indicating four and three biological replicates respectively, each performed in 6 replicates. (**i**) Basal respiration, maximal respiration, ATP-production and spare respiration capacity was calculated; mean ± SD. $P$ by two-sided t-test. $n$ = 4 biological replicates for WT and 3 biological replicates for *Dnmt3a*[R882H/+]. Schematic in **g** was created using BioRender (https://biorender.com). Seahorse picture in **g** adapted with permission from K. Gozdecki.

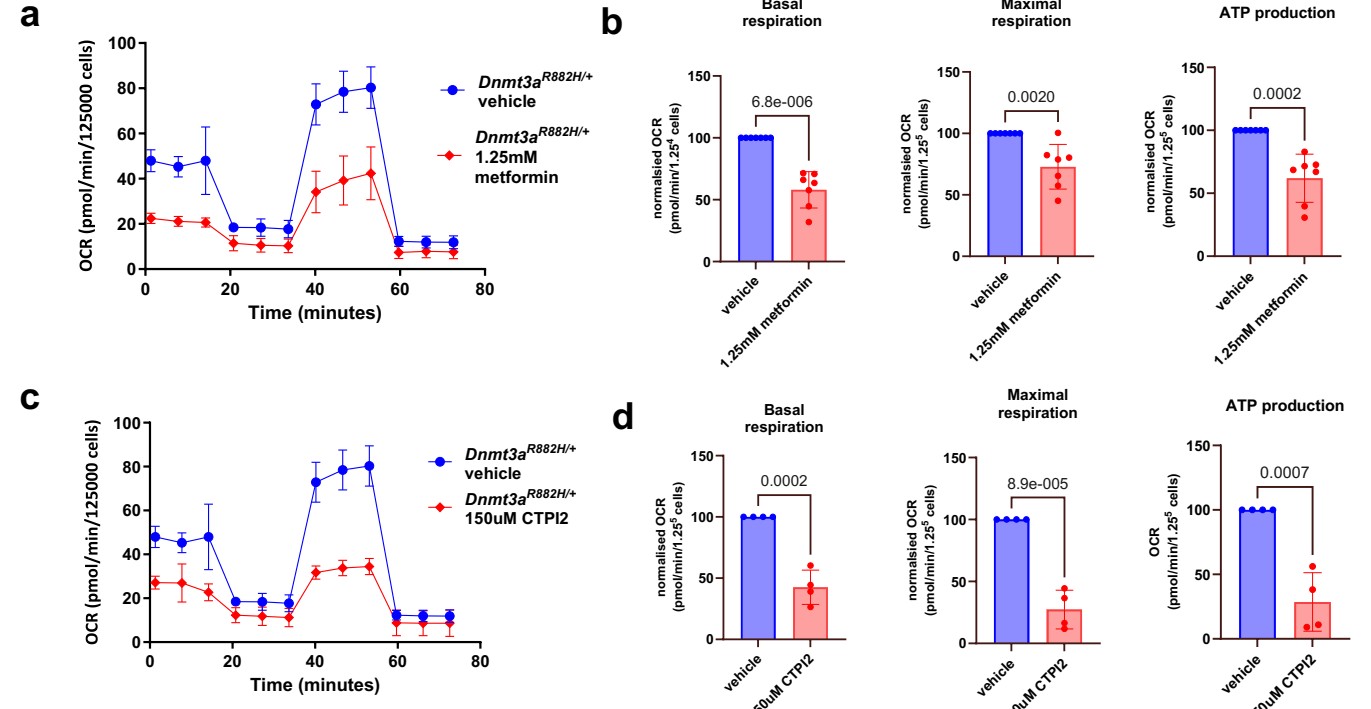

**Extended Data Fig. 5 | The effect of metformin and CTPI2 on mitochondrial respiration in *Dnmt3a*^R882H/+ HSPCs.** (**a**) Oxygen consumption rate in *Dnmt3a*^R882H/+ treated in vitro with 1.25 mM metformin for 2 h, mean ± SD. $n = 11$ for vehicle and $n = 12$ for metformin, indicating three biological replicates per condition, for metformin performed in 4 replicates and for vehicle, two performed in 3 and one in 4 replicates. One of three independent experiments is shown. (**b**) Basal, maximal respiration and ATP-production were compared between metformin and vehicle treatment, mean ± SD. *P* by two-sided t-test. Data were normalised to each vehicle control. $n = 7$ biological replicates per condition from three independent experiments. (**c**) Oxygen consumption rate in *Dnmt3a*^R882H/+ treated with 150uM of CTPI2 for 2 h, mean ± SD. $n = 11$ for vehicle and $n = 10$ for CTPI2, indicating three biological replicates per condition, for CTPI2 performed in 4, 3 and 3 replicates and for vehicle, two performed in 3 and one in 4 replicates. One of two independent experiments is shown (**d**) Basal, maximal respiration and ATP-production were compared between CTPI2 and vehicle, mean ± SD, *P* by two-sided t-test. Data were normalised to each vehicle control. $n = 4$ biological replicates per condition from two independent experiments.

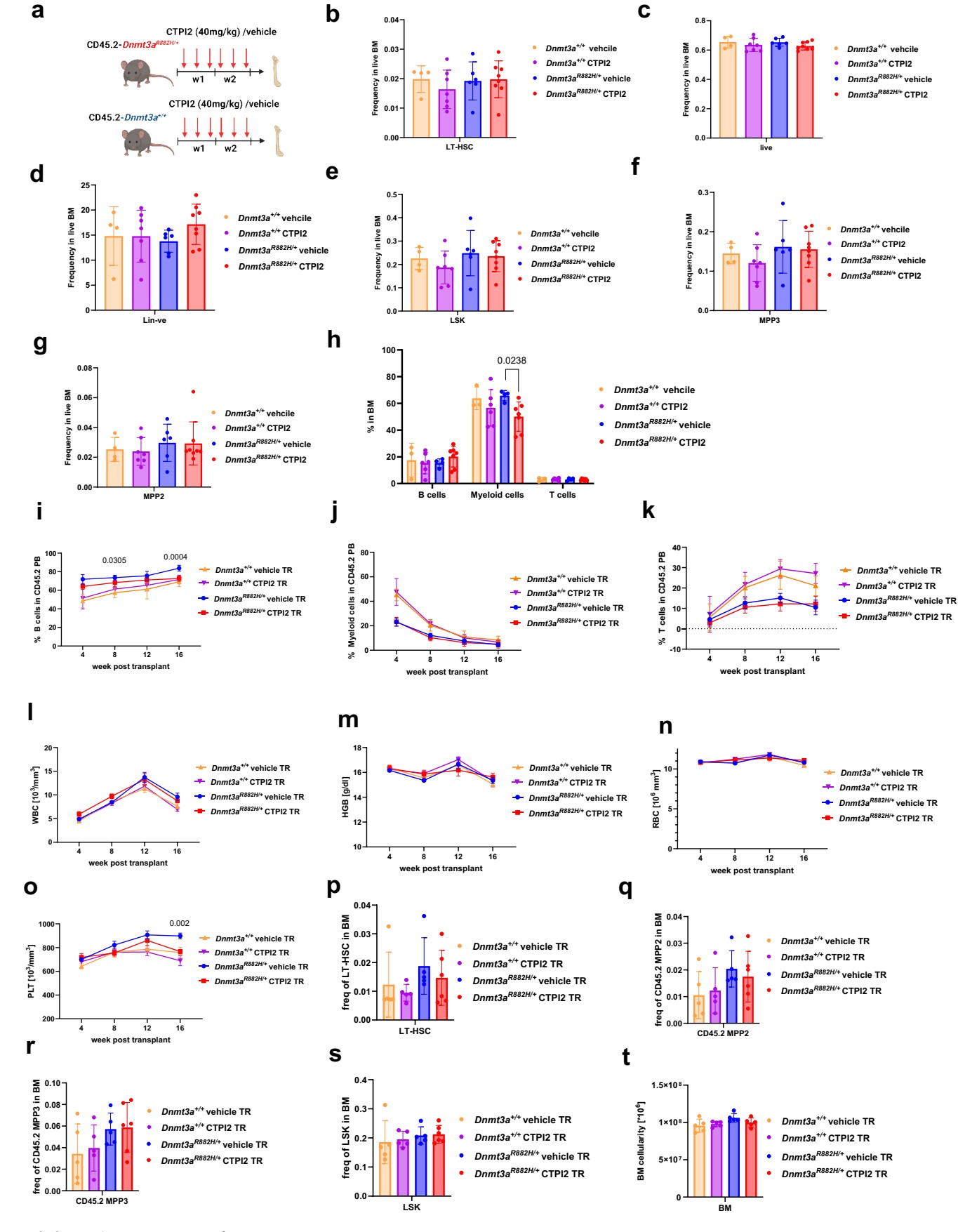

**Extended Data Fig. 6** | See next page for caption.

**Extended Data Fig. 6 | Characterisation of the blood progenitor and differentiated cell compartments post-CTPI2 treatment.** (**a**) Schematic representation of the treatment of primary mice with CTPI2. (**b**) Frequencies of LT-HSC (**c**) live, (**d**) Lin-ve, (**e**) LSK, (**f**) MPP3 (**g**) MPP2 and (**h**) proportion of differentiated cells in primary WT and $Dnmt3a^{R882H/+}$ mice treated with CTPI2/vehicle for 2 weeks. Mean ± SD is shown, in (b-g) $n = 4$ for WT vehicle, $n = 7$ for WT CTPI2, $n = 6$ for $Dnmt3a^{R882H/+}$, and $n = 8$ mice for $Dnmt3a^{R882H/+}$ CTPI2 group. In (h) $n = 3$ for WT vehicle, $n = 6$ for WT CTPI2, $n = 4$ for $Dnmt3a^{R882H/+}$, and $n = 7$ mice for $Dnmt3a^{R882H/+}$ CTPI2 group. (**i**) PB proportion of B (**j**) Myeloid (**k**) T cells in CTPI2/vehicle treated and transplanted CD45.2-WT and $Dnmt3a^{R882H/+}$. (**l**) WBC, (**m**) HGB, (**n**) red blood cells (RBC), (**o**) PLT in CTPI2/vehicle treated and transplanted CD45.2-WT and $Dnmt3a^{R882H/+}$. (**p**) Frequencies of total (CD45.1 and CD45.2) LT-HSC, (**q**) MPP2, (**r**) MPP3, (**s**) LSK and (**t**) BM cellularity in CTPI2/vehicle treated and transplanted WT and $Dnmt3a^{R882H/+}$ at end point. Mean ± SD is shown, $n = 6$ for $Dnmt3a^{R882H/+}$ CTPI2 and $n = 5$ mice for remining groups for (i-t). $P$ by two-sided t-test, in (h), (i) and (o), $P$ is displayed for comparison of $Dnmt3a^{R882H/+}$ vehicle TR vs $Dnmt3a^{R882H/+}$ CTPI2 TR. Schematic in **a** was created using BioRender (https://biorender.com).

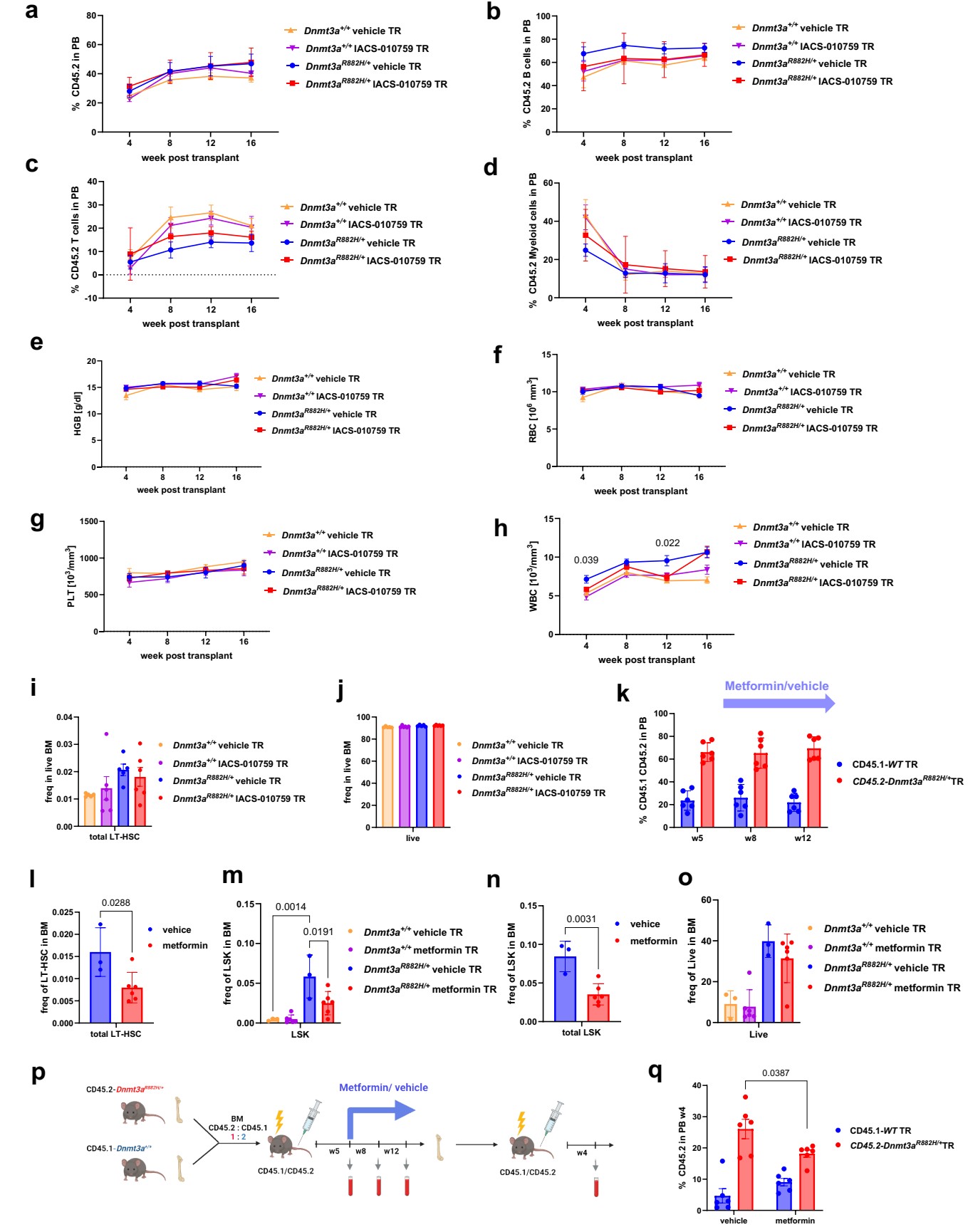

**Extended Data Fig. 7** | See next page for caption.

**Extended Data Fig. 7 | Characterisation of the blood progenitor and differentiated cell compartments post-IACS-010759 treatment.**
(**a**) Proportion of CD45.2 (**b**) B (**c**) T and (**d**) Myeloid cells in PB of IACS-010759/vehicle treated and transplanted CD45.2-WT and $Dnmt3a^{R882H/+}$ recipient mice (**e**) HGB, (**f**) red blood cells (RBC), (**g**) PLT and (**h**) WBC in IACS-010759/vehicle treated and transplanted WT and $Dnmt3a^{R882H/+}$. (a-d) Mean ± SD is shown, $n = 6$ mice for IACS-010759 and $n = 5$ mice for vehicle groups. $P$ by two-sided t-test, in (h) $P$ displayed for comparison of $Dnmt3a^{R882H/+}$ vehicle TR vs $Dnmt3a^{R882H/+}$ IACS-010759 TR. (**i**) Frequencies of total (CD45.1 and CD45.2) LT-HSC and (**j**) live cells in BM of IACS-010759/vehicle treated and transplanted WT and $Dnmt3a^{R882H/+}$. (e-j) Mean ± SEM is shown, $n = 5$ mice for vehicle and $n = 6$ mice for IACS-010759 groups. (**k**) Proportion of transplanted $Dnmt3a^{R882H/+}$ and WT cells in PB before (w5) and during metformin treatment. Mean ± SD is shown, $n = 6$ mice. (**l**) Frequencies of total LT-HSC in BM (**m**) Frequencies of TR $Dnmt3a^{R882H/+}$ and WT LSK in BM, (**n**) total frequencies of LSK in BM (**o**) Frequencies of live $Dnmt3a^{R882H/+}$ and WT TR cells in BM, the mean ± SD is shown. For (l-o) $n = 3$ mice for vehicle group and $n = 6$ mice per metformin group. $P$ in m by one-way ANOVA with Tukey correction; $P$ in l and $n$ by two-sided t-test. (**p**) Secondary transplantation strategy. (**q**) Quantification of the proportion of WT and $Dnmt3a^{R882H/+}$ cells in PB of mice treated with metformin/vehicle post transplantation. Mean ± SEM is shown, $n = 6$ mice per group, $P$ by two-sided t-test. Schematic in **p** was created using BioRender (https://biorender.com).

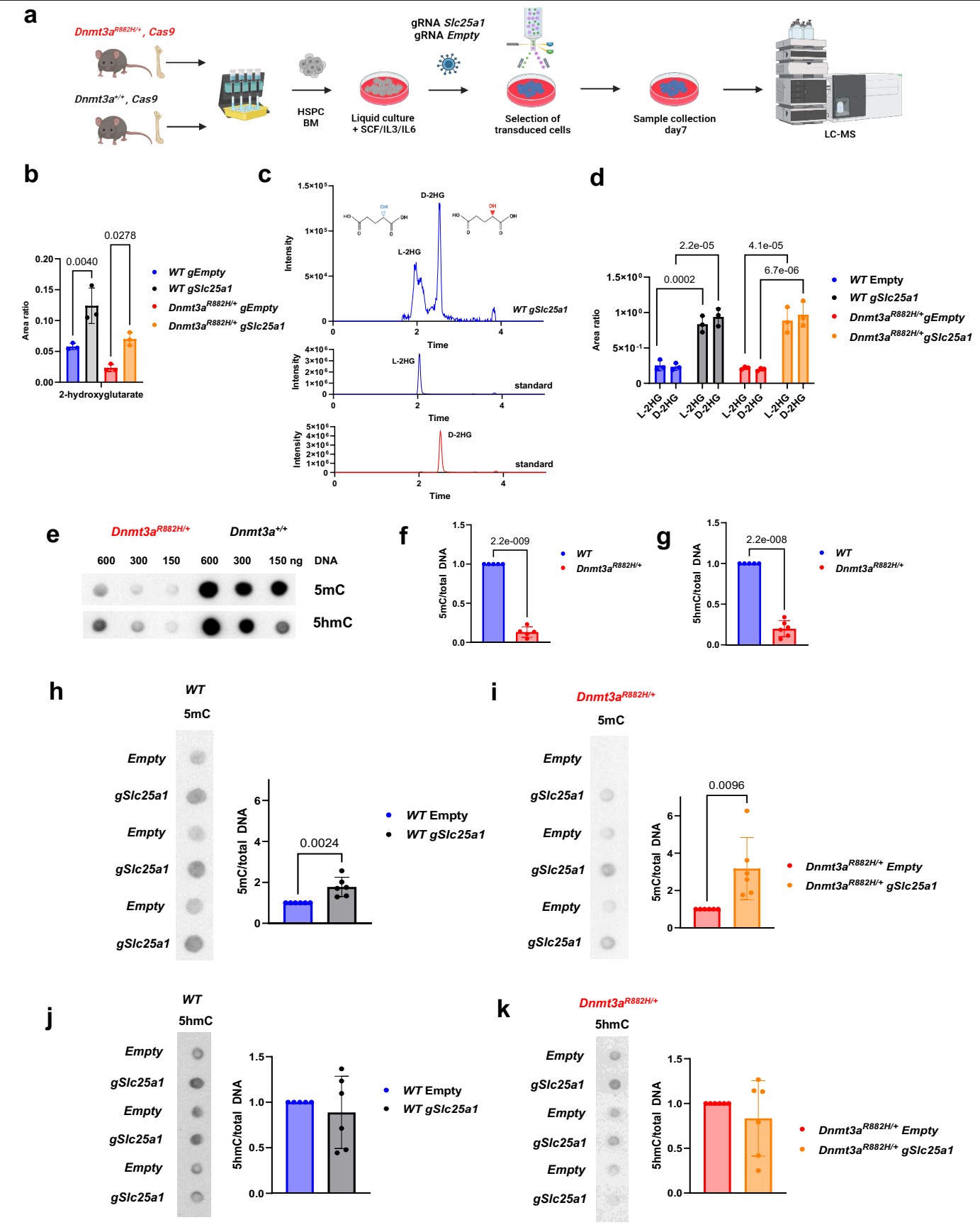

**Extended Data Fig. 8** | See next page for caption.

**Extended Data Fig. 8 | Increased L,D-2HG levels and 5mC in *Dnmt3a*$^{R882H/+}$ upon *Slc25a1* knockout.** (**a**) Schematic representation of experimental strategy. (**b**) The levels of 2HG in *Slc25a1* ko/control WT and *Dnmt3a*$^{R882H/+}$ cells. Mean ± SD is shown, $n = 3$ biological replicates per group, *P* by one-way ANOVA. (**c**) Chromatogram indicating L and D-2HG in exemplary WT sample upon knockout of *Slc25a1*. L-2HG and D-2HG standards are presented below. Similar data were observed for $n = 3$ biological replicates. (**d**) Levels of L-2HG and D-2HG in each sample by LC-MS. Mean ± SD is shown, $n = 3$ biological replicates per group, *P* by two-way ANOVA. (**e**) DNA methylation and hydroxymethylation in WT and *Dnmt3a*$^{R882H/+}$ HSPCs by dot blot. (**f**) Quantification of 5mC, $n = 5$ biological replicates per group, and (**g**) 5hmC levels in *Dnmt3a*$^{R882H/+}$ relatively to WT. $n = 5$ for WT and $n = 6$ biological replicates for *Dnmt3a*$^{R882H/+}$. (**h**) Dot blot and quantification of 5mC levels in *WT Slc25a1* null cells compared to WT Empty controls, and (**i**) *Dnmt3a*$^{R882H/+}$ *Slc25a1* null cells compared to *Dnmt3a*$^{R882H/+}$ Empty controls. (**j**) 5hmC levels in WT and (**k**) *Dnmt3a*$^{R882H/+}$ compared to Empty controls. (f-k) Mean ± SD is shown, in (j) $n = 5$ biological replicates for WT Empty, and in remining groups in (h-k) $n = 6$ biological replicates, *P* by two-sided t-test. Schematic in **a** was created using BioRender (https://biorender.com).

**a**

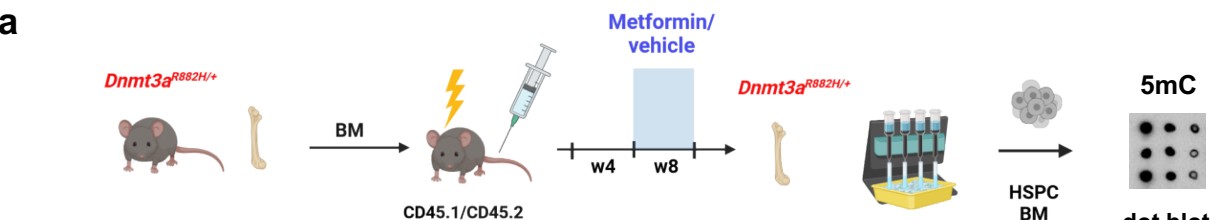

**b**

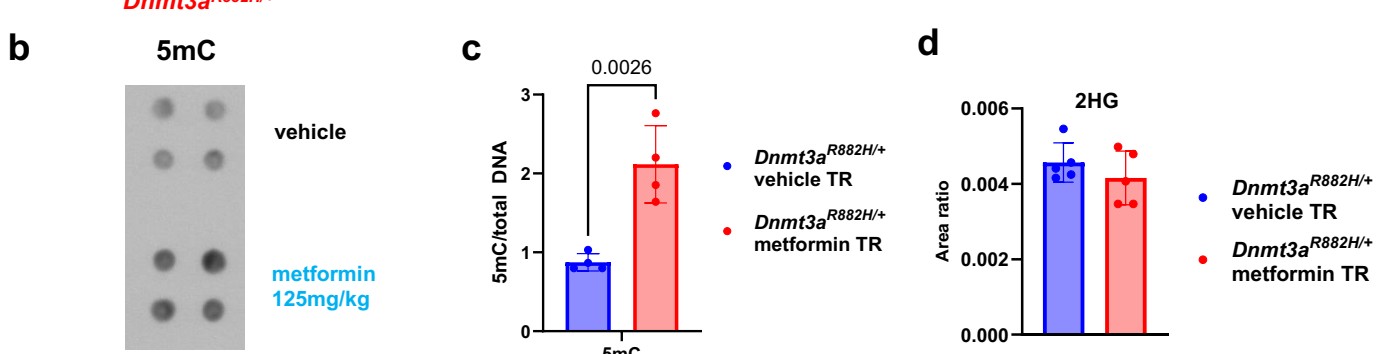

**Extended Data Fig. 9 | Increased 5mC in *Dnmt3a*ᴿ⁸⁸²ᴴ/⁺ HSPCs treated with metformin in vivo.** (**a**) Schematic representation of the experimental strategy. (**b**) 5mC dot blot and (**c**) quantification. (**d**) 2HG levels by targeted LC-MS.

Mean ± SD is shown, *n* = 4 biological replicates per group, in b - each dot represents different biological replicate. *P* by two-sided t-test. Schematic in **a** was created using BioRender (https://biorender.com).

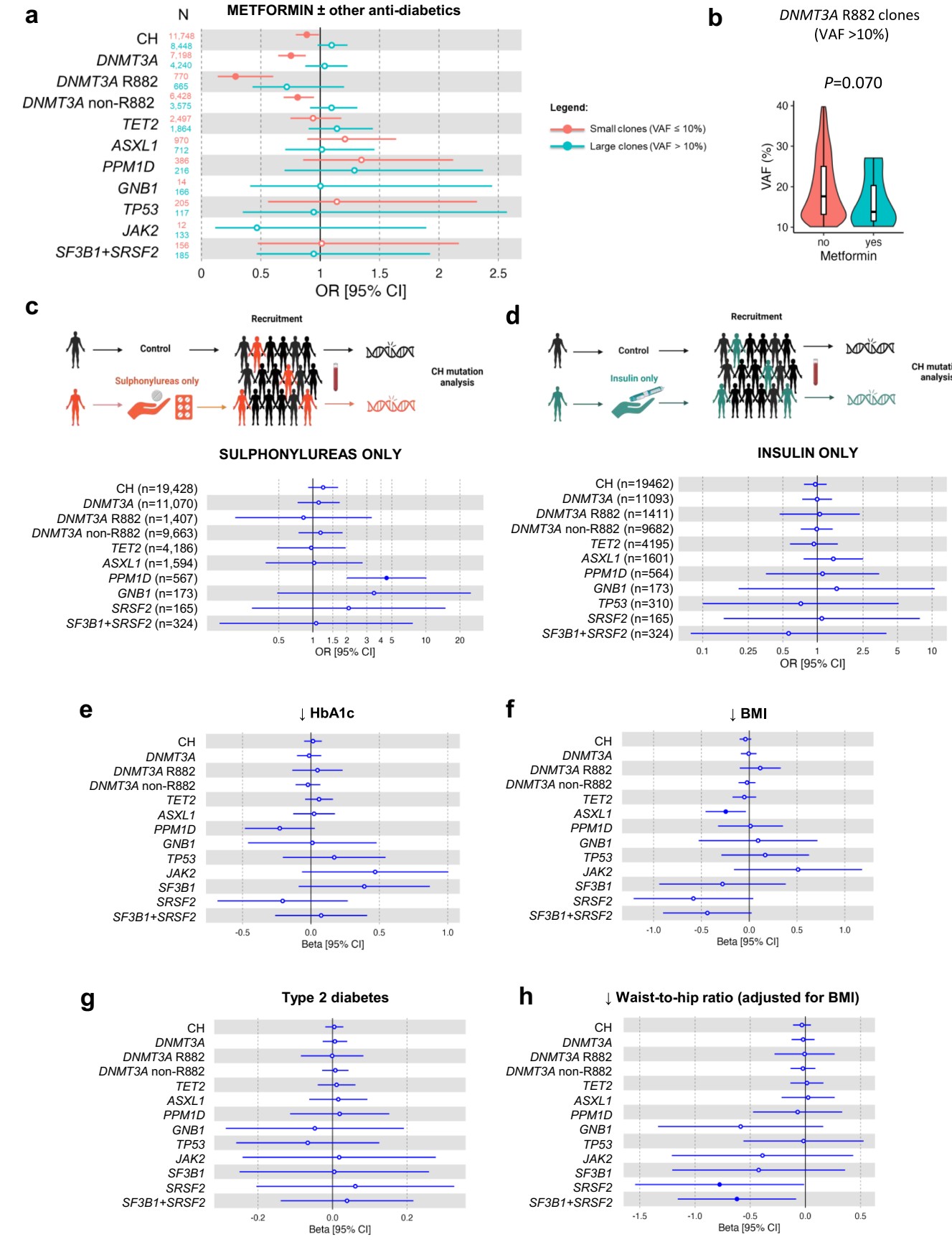

**Extended Data Fig. 10 |** See next page for caption.

**Extended Data Fig. 10 | Lack of association of sulphonylureas and insulin administration or genetically predicted glycemic-related traits with *DNMT3A*-R882 CH.** (**a**) Association between metformin and overall CH or gene-specific CH risk. CH split by small clone (VAF ≤ 10%) and large clone (VAF > 10%). In total, 11,190 individuals taking metformin or metformin in combination with other antidiabetic medications and 401,044 individuals not on metformin were included in the analysis. (**b**) Distribution of large *DNMT3A*-R882 clones (VAF > 10%) among 15 individuals taking metformin or metformin in combination with other anti-diabetic medications and 650 individuals not on metformin. The *P* value was derived using linear regression with VAF as outcome and metformin as predictor, adjusted for age, sex, smoking status, and first four genetic principal components (PCs). Boxplots represent the median, first and third quartiles, with whiskers representing 1.5 times the interquartile range. (**c**) Association between sulphonylureas (glibenclamide, gliclazide, glipizide, glimepiride, tolbutamide) and overall CH or gene-specific CH risk. Only individuals taking sulphonylureas as the only form of anti-diabetic medication were included in the analysis. In total, 571 individuals on sulphonylureas only and 398,712 individuals not on anti-diabetic medications were included for analysis. We observed significant association for *PPM1D* CH: OR = 4.49 [95 CI% = 1.99-10.12], *P* = 0.00029. (**d**) Association between insulin and overall CH or gene-specific CH risk. Only individuals taking insulin as the only form of anti-diabetic medication were included in the analysis. In total, 1,488 individuals on insulin only and 398,712 individuals not on anti-diabetic were included for analysis. For (**a**,**c**,**d**) odds ratios and two-sided unadjusted *P* values were derived from logistic regression model with all CH or gene-specific CH as outcome, and with age, sex, smoking, and the first four genetic principal components as covariates. Measures of centre represent the odds ratios, and the error bars represent the lower and upper bound of the 95% confidence interval of the odds ratios. Significant *P* values (<0.05) are indicated with full circles. (**e-h**) MR analyses of the effect of HbA1c, BMI, T2D, BMI-adjusted WTHR (exposure) on overall CH and gene-specific CH risk (outcome). In total, 392,186 individuals not on metformin at recruitment and passed sample-level QC for genotyping array (excluded samples with genotype missingness > 5%, samples with non-XX or -XY chromosome configuration, and samples with high heterozygosity) were included for analysis here. Measures of centre represent the beta coefficients, and the error bars represent the lower and upper bound of the 95% confidence interval of the beta coefficients. Beta coefficients and unadjusted *P* values were derived from inverse variance weighting method. Significant *P* values (<0.05) are indicated with full circles. Schematics in **c**,**d** were created using BioRender (https://biorender.com).

Brian Huntly
George Vassiliou

# Reporting Summary

## Statistics

For all statistical analyses, confirm that the following items are present in the figure legend, table legend, main text, or Methods section.

| n/a | Confirmed | |
|---|---|---|
| ☐ | ☒ | The exact sample size (*n*) for each experimental group/condition, given as a discrete number and unit of measurement |
| ☐ | ☒ | A statement on whether measurements were taken from distinct samples or whether the same sample was measured repeatedly |
| ☐ | ☒ | The statistical test(s) used AND whether they are one- or two-sided *Only common tests should be described solely by name; describe more complex techniques in the Methods section.* |
| ☐ | ☒ | A description of all covariates tested |
| ☐ | ☒ | A description of any assumptions or corrections, such as tests of normality and adjustment for multiple comparisons |
| ☐ | ☒ | A full description of the statistical parameters including central tendency (e.g. means) or other basic estimates (e.g. regression coefficient) AND variation (e.g. standard deviation) or associated estimates of uncertainty (e.g. confidence intervals) |
| ☒ | ☐ | For null hypothesis testing, the test statistic (e.g. $F$, $t$, $r$) with confidence intervals, effect sizes, degrees of freedom and $P$ value noted *Give P values as exact values whenever suitable.* |
| ☒ | ☐ | For Bayesian analysis, information on the choice of priors and Markov chain Monte Carlo settings |
| ☒ | ☐ | For hierarchical and complex designs, identification of the appropriate level for tests and full reporting of outcomes |
| ☒ | ☐ | Estimates of effect sizes (e.g. Cohen's *d*, Pearson's *r*), indicating how they were calculated |

*Our web collection on statistics for biologists contains articles on many of the points above.*

## Software and code

Policy information about availability of computer code

| Data collection | - CRISPR sequencing was performed with HiSeq2500 system, Illumina.<br>- BD FACSDiva Software v9.0<br>- Histology slide scanner: Leica Aperio Slide Scanner AT20, running AperioServiceManager version 12.4.3.5008.<br>- LC-MS was performed with a Thermo Fisher Scientific Vanquish Horizon UHPLC System coupled to a Q Exactive Plus MS (Orbitrap MS 2.11 build 3006). The following software applications were used: Foundation 3.1 SP6, FreeStyle 1.5, Xcalibur 4.2 SP1 (version 4.2.47) and Thermo Scientific SII for Xcalibur (version 1.5.0.10747)<br>- Whole-exome sequencing reads of AML and control samples were mapped to the mouse genome assembly GRCm39 using BWA version 0.7.18 (arXiv:1303.3997 [q-bio.GN]) under default parameters and duplicated reads were flagged using Samtools version 1.9 |
|---|---|
| Data analysis | - FlowJo (version 10.8.2)<br>- Seahorse data analysis was performed with Wave (version 2.6.3)<br>- CRISPR screen data was analysed with MAGeCK statistical package (version 0.5.3)<br>- Survival analysis was performed with R package (version 3.5-8) https://CRAN.R-project.org/package=survival.<br>- R package (version 4.2.2): For modelling anti-diabetic medications and clonal hematopoiesis (CH) using logistic regression.<br>- LC-MS data analysis was performed with Thermo Scientific Xcalibur 4.1 (version 4.1.31.9)<br>- Data were analysed and graphed using GraphPad Prism version 10 (v10.4.1)<br>- IGV software was used for visualizing data shown in Fig.1b. (version 2.3.0)<br>- MR analyses were performed using TwoSampleMR (v0.5.6 R) package<br>- Somatic mutations were called by GATK Mutect2 version 4.587<br>- Mutation calls were annotated with Ensembl VEP version 112 |

- Dot blots were scanned and analysed using Image Studio Lite software version 5.2

For manuscripts utilizing custom algorithms or software that are central to the research but not yet described in published literature, software must be made available to editors and reviewers. We strongly encourage code deposition in a community repository (e.g. GitHub). See the Nature Portfolio guidelines for submitting code & software for further information.

## Data

Policy information about availability of data

All manuscripts must include a data availability statement. This statement should provide the following information, where applicable:
- Accession codes, unique identifiers, or web links for publicly available datasets
- A description of any restrictions on data availability
- For clinical datasets or third party data, please ensure that the statement adheres to our policy

- CRISPR screen raw data have been deposited in Gene Expression Omnibus with accession number GSE259404.
- Whole-exome and targeted sequencing data are available in Sequence Read Archive BioProject PRJNA1160274.
-Metabolomic data were deposited into MetaboLight depository (https://www.ebi.ac.uk/metabolights/) with accession number: MTBLS12201
- TCGA-AML cohort (151 samples) were downloaded from the GDA data portal. GDA portal accession weblink: https://portal.gdc.cancer.gov/projects/TCGA-LAML
-For UKB, individual-level data are under controlled accessed to protect sensitive information of the study participants. Individual-level UKB data may be requested via application to the UKB. All whole-exome sequencing data described in our study are available to registered researchers through the UKB data access protocol. Exomes can be found in the UKB showcase portal: https://biobank.ndph.ox.ac.uk/showcase/label.cgi?id=170. Additional information about data access registration is available at https://www.ukbiobank.ac.uk/enable-your-research/register.
- Whole-exome sequencing reads of AML and control samples were mapped to the mouse genome assembly GRCm39 using BWA version 0.7.18 (arXiv:1303.3997 [q-bio.GN]) under default parameters and duplicated reads were flagged using Samtools version 1.9
- Enrichr online software webpage link: https://maayanlab.cloud/Enrichr/
- DGIdb webpage link: https://dgidb.org/
-DepMap, Broad (2019) webpage link: https://figshare.com/articles/dataset/DepMap_19Q4_Public/11384241/3
- Survival Analysis R package, version 3.5-8, weblink: https://CRAN.R-project.org/package=survival.
- Meta-Analyses of Glucose and Insulin-related traits Consortium (MAGIC) http://magicinvestigators.org/

## Research involving human participants, their data, or biological material

Policy information about studies with human participants or human data. See also policy information about sex, gender (identity/presentation), and sexual orientation and race, ethnicity and racism.

| | |
|---|---|
| Reporting on sex and gender | For UKB cohort, sex concordance was determined by comparing clinically-reported sex against chromosome X:Y consensus coding sequence coverage ratio. Only sex, but not gender, was reported by this study.<br>- Human DNMT3A-R882 CH who donated PB sample used in this study was 41 year old female |
| Reporting on race, ethnicity, or other socially relevant groupings | For the UK Biobank cohort, ancestry of study participants were determined using the peddy software. Europeans individuals from UKB were defined with peddy-inferred European probability of at least 95% and were used for regression analyses.<br>- The DNMT3A-R882 CH participant was of European ancestry. |
| Population characteristics | UKB is a prospective cohort of approximately 500,000 adults, aged between 40 to 70 years, with genetic (whole-genome sequencing, whole-exome sequencing, SNP array) and phenotypic, proteomic, and matobolomic data available. 54.4% women, 45.6% men. Details for UKB have been described in Szustakowski et al. (Nature Genetics, 2021) and Bycroft et al. (Nature, 2018).<br>- Human DNMT3A-R882 CH participant was 41 year old female |
| Recruitment | For the UKB cohort, participants were aged between 40 to 70 years, and recruited between 2006 and 2010.  The UKB has a degree of healthy volunteer bias.<br>The DNMT3A-R882 CH carrier attends the Cambridge Clonal Haematopoiesis Clinic, where she was approached and recruited into this study in 2022. |
| Ethics oversight | -UKB study has approval from the North-West Multi-centre Research Ethics Committee (11/NW/0382).<br>-Ethical approval for human DNMT3A-R882 CH sample used this study was granted by the East of England (Cambridge East) Research Ethics Committee (REC reference: 24/EE/0116). Informed consent was provided by the participant. |

Note that full information on the approval of the study protocol must also be provided in the manuscript.

# Field-specific reporting

Please select the one below that is the best fit for your research. If you are not sure, read the appropriate sections before making your selection.

☒ Life sciences          ☐ Behavioural & social sciences          ☐ Ecological, evolutionary & environmental sciences

For a reference copy of the document with all sections, see nature.com/documents/nr-reporting-summary-flat.pdf

# Life sciences study design

All studies must disclose on these points even when the disclosure is negative.

| | |
|---|---|
| Sample size | Sample sizes were chosen based on power calculations of expected differences and prior experience with these types of experiments. Sample sizes of n=3 mice per genotype/cell culture were chosen for most experiments with the exception of flow cytometric analysis and mouse experiments where sample size was increased to account for variation between individuals or for the need to carry out experiments at more than one timepoint. |
| Data exclusions | No data were excluded from the analysis |
| Replication | Most experiments were replicated 2-3 times. Independent biological and technical replicates were used per experiment, exact information is included in each figure legend. |
| Randomization | Mice were allocated to the study groups by genotype. For transplantation experiments animals of the same sex and similar age range were randomly assigned to study groups. For in vitro experiments mouse samples were also selected by genotype. |
| Blinding | Although the investigators were not blinded to the genotype of the animals, the animal technicians who provided the animal care, supervision and identification of sick animals at humane endpoint (and therefore making the decision to sacrifice sick animals) were blinded in order to eliminate survival bias. For transplant experiments, flow cytometry data collection and analyses were performed blind (by assigning a number to each animal ID). For in vitro experiments analyses were performed in batches of animals from both test and control groups using numbers instead of genotypes at the time of data acquisition. For statistical analyses samples were grouped into test vs control. |

# Behavioural & social sciences study design

All studies must disclose on these points even when the disclosure is negative.

| | |
|---|---|
| Study description | N/A |
| Research sample | *State the research sample (e.g. Harvard university undergraduates, villagers in rural India) and provide relevant demographic information (e.g. age, sex) and indicate whether the sample is representative. Provide a rationale for the study sample chosen. For studies involving existing datasets, please describe the dataset and source.* |
| Sampling strategy | *Describe the sampling procedure (e.g. random, snowball, stratified, convenience). Describe the statistical methods that were used to predetermine sample size OR if no sample-size calculation was performed, describe how sample sizes were chosen and provide a rationale for why these sample sizes are sufficient. For qualitative data, please indicate whether data saturation was considered, and what criteria were used to decide that no further sampling was needed.* |
| Data collection | *Provide details about the data collection procedure, including the instruments or devices used to record the data (e.g. pen and paper, computer, eye tracker, video or audio equipment) whether anyone was present besides the participant(s) and the researcher, and whether the researcher was blind to experimental condition and/or the study hypothesis during data collection.* |
| Timing | *Indicate the start and stop dates of data collection. If there is a gap between collection periods, state the dates for each sample cohort.* |
| Data exclusions | *If no data were excluded from the analyses, state so OR if data were excluded, provide the exact number of exclusions and the rationale behind them, indicating whether exclusion criteria were pre-established.* |
| Non-participation | *State how many participants dropped out/declined participation and the reason(s) given OR provide response rate OR state that no participants dropped out/declined participation.* |
| Randomization | *If participants were not allocated into experimental groups, state so OR describe how participants were allocated to groups, and if allocation was not random, describe how covariates were controlled.* |

# Ecological, evolutionary & environmental sciences study design

All studies must disclose on these points even when the disclosure is negative.

| | |
|---|---|
| Study description | N/A |
| Research sample | *Describe the research sample (e.g. a group of tagged Passer domesticus, all Stenocereus thurberi within Organ Pipe Cactus National Monument), and provide a rationale for the sample choice. When relevant, describe the organism taxa, source, sex, age range and any manipulations. State what population the sample is meant to represent when applicable. For studies involving existing datasets, describe the data and its source.* |
| Sampling strategy | *Note the sampling procedure. Describe the statistical methods that were used to predetermine sample size OR if no sample-size calculation was performed, describe how sample sizes were chosen and provide a rationale for why these sample sizes are sufficient.* |

| Data collection | *Describe the data collection procedure, including who recorded the data and how.* |
|---|---|
| Timing and spatial scale | *Indicate the start and stop dates of data collection, noting the frequency and periodicity of sampling and providing a rationale for these choices. If there is a gap between collection periods, state the dates for each sample cohort. Specify the spatial scale from which the data are taken* |
| Data exclusions | *If no data were excluded from the analyses, state so OR if data were excluded, describe the exclusions and the rationale behind them, indicating whether exclusion criteria were pre-established.* |
| Reproducibility | *Describe the measures taken to verify the reproducibility of experimental findings. For each experiment, note whether any attempts to repeat the experiment failed OR state that all attempts to repeat the experiment were successful.* |
| Randomization | *Describe how samples/organisms/participants were allocated into groups. If allocation was not random, describe how covariates were controlled. If this is not relevant to your study, explain why.* |
| Blinding | *Describe the extent of blinding used during data acquisition and analysis. If blinding was not possible, describe why OR explain why blinding was not relevant to your study.* |

Did the study involve field work? ☐ Yes ☒ No

## Field work, collection and transport

| Field conditions | N/A |
|---|---|
| Location | *State the location of the sampling or experiment, providing relevant parameters (e.g. latitude and longitude, elevation, water depth).* |
| Access & import/export | *Describe the efforts you have made to access habitats and to collect and import/export your samples in a responsible manner and in compliance with local, national and international laws, noting any permits that were obtained (give the name of the issuing authority, the date of issue, and any identifying information).* |
| Disturbance | *Describe any disturbance caused by the study and how it was minimized.* |

# Reporting for specific materials, systems and methods

We require information from authors about some types of materials, experimental systems and methods used in many studies. Here, indicate whether each material, system or method listed is relevant to your study. If you are not sure if a list item applies to your research, read the appropriate section before selecting a response.

### Materials & experimental systems

| n/a | Involved in the study |
|---|---|
| ☐ | ☒ Antibodies |
| ☐ | ☒ Eukaryotic cell lines |
| ☒ | ☐ Palaeontology and archaeology |
| ☐ | ☒ Animals and other organisms |
| ☒ | ☐ Clinical data |
| ☒ | ☐ Dual use research of concern |
| ☒ | ☐ Plants |

### Methods

| n/a | Involved in the study |
|---|---|
| ☒ | ☐ ChIP-seq |
| ☐ | ☒ Flow cytometry |
| ☒ | ☐ MRI-based neuroimaging |

## Antibodies

| Antibodies used | Antibody used for FACS:<br><br>B220-Biotin, BD Biosciences, #559971, clone: RA3-6B2, lot: 6169942<br>Ter119-Biotin, BD Biosciences, #559971, clone: TER-119, lot: 6169946<br>Mac1-Biotin, BD Biosciences, #559971, clone: M1/70, lot: 6169944<br>CD3-Biotin, BD Biosciences, #559971, clone: 145-2C11, lot: 6169941<br>Gr1-Biotin, BD Biosciences, #559971, clone: RB6-8C5, lot: 6169943<br>Sca1-PB, BioLegend, #122520, clone: E13-161.7, lot: B385332<br>c-kit AF780, Thermo Fisher Scientific, #47-1171-82, clone: 2138, lot: 2577317<br>CD48-BV605, BioLegend, #103441, clone: HM48-1, lot: B308985<br>CD150-BV650, BioLegend, #115932, clone: TC15-12F12.2, lot: B390003<br>Cd135 PE, BD Biosciences, #553842, clone: A2F10.1, lot: 1354615<br>Cd34 FITC, Thermo Fisher Scientific, #11-0431-82, clone: RAM34, lot: 2518336<br>Cd16/32 PerCP PeCy5.5, BioLegend, #101324, clone: 93, lot: B378771 |
|---|---|

CD45.1 APC, BioLegend, #110714, clone: A20, lot: B254042
CD45.2 Pe-Cy7, Thermo Fisher Scientific, #25-0454-82, clone: 104, lot: 2628857
CD45.1 BV605, BioLegend, #110738, clone: A20, lot: B386453
CD45.2 FITC, BioLegend, #109806, clone: 104, lot: B395596
CD4 Pe-Cy5, BioLegend, #100514, clone: RM4-5, lot: B341915
Cd8 Pe-Cy5, BioLegend, #100710, clone: 53-6.7, lot: B369730
B220 Pe-Cy5, BioLegend, #103210, clone: RA3-6B2, lot: B354953
Gr1 AF700 , BioLegend, #108422, clone: RB6-8C5, lot: B386943
Mac1 AF700 , BioLegend, #101222, clone: M1/70 , lot: B350819
B220AF700 , BioLegend, #103232, clone: RA3-6B2, lot: B375847
7 AAD , BioLegend, #420404, lot: B383019
streptavidin BV510, BioLegend, #405233, lot: B362883
CD4 (Biolegend, #100514, clone: RM4-5, lot: B171933)
CD5 (Biolegend, #100610, clone: 53-7.3, lot: B178256)
CD8a (Biolegend, # 100710, clone: 53-6.7, lot: B166422)
CD11b (Biolegend, #101210, clone: M1/70, lot: B167424)
B220 (Biolegend, #103210, clone: RA3-6B2, lot: E07569-1635)
TER-119 (Biolegend, #116210, clone: TER-119, lot: B169021)
GR-1 (Biolegend, #108410, clone: RB6-8C5, lot: B179158)
SCA-1 (Biolegend, #122520, clone: E13-161.7, lot: B174209)
CD117 (eBioscience, # 47-1171-82, clone: 2B8, lot: E08461-1637)
CD48 (Biolegend, #103411, clone: HM-48-1, lot: B214727)
CD150 (Biolegend, #115913, clone: TC15-12F12.2, lot: B172059)
CD34 (BD Pharmigen, #553733, clone: RAM34, lot: 4319145)
FLT3 (BD Pharmigen, #553842, clone: A2F10.1, lot: 6148682)
IL7Ra (Biolegend, #135008, clone: A7R34, lot: B190405)
IL7Ra (Biolegend, #121103, clone: SB/199, lot: B176923)
Streptavidin (Biolegend, #405206, lot: B182997)
CD16/32 (BD Pharmigen, #553145, clone: 2.4G2, lot: 3123812)
c-KIT (BioLegend, #105812, clone: 2B8, lot: B217855)
NK (LSBio, LS-C62548, clone: PK136, lot: 44694)
CD45 (BD Pharmigen, #563891, clone: 30-F11, lot: 4276708)
CD11b (BD Pharmigen, #557657, clone: M1/70, lot: 4275513)
GR1 (BD Pharmigen, #560603, clone: 1A8, lot: 4290881)
TER119 (BD Pharmigen, #557915, clone: TER-119, lot: 7100737)
CD3e (eBioscience, #12-0031-82, clone: 145-2C11, lot: 4308509)

Antibody used for dot blot:
anti-5hmC Rb antibody, Active Motif, #39769, Lot: 25922004-14
anti-5mC Ms antibody, Active Motif, #61479, Lot: 23264180-11
anti- rabbit HRP-conjugated secondary antibody, 111-035-003, Jackson ImmunoResearch Laboratories Inc., lot #162297
anti-mouse HRP-conjugated secondary antibody (115-035-146, Jackson ImmunoResearch Laboratories Inc., lot #154319

Validation

Antibodies used in this study were validated for this application by the manufactures. The antibodies were also previously validated and published in multiple studies. Additionally, for each experiment we had relevant controls to ensure correct interpretation of results.
Lineage biotin panel kit, BD Biosciences, #559971, containing:
B220-biotin, BD Biosciences, clone: RA3-6B2
Ter119-biotin, BD Biosciences, clone: TER-119
Mac1-biotin, BD Biosciences, clone: M1/70
CD3-biotin, BD Biosciences, clone: 145-2C11
Gr1-Biotin, BD Biosciences, clone: RB6-8C5

Validation, references:
Used in several publications (see https://www.citeab.com/kits/10160766-559971-bd-pharmingen-biotin-mouse-lineage-panel) amongst them:
Thambyrajah, R., Maqueda, M., et al. 2024 Nat Commun. 15:4673

Sca1-PB, BioLegend UK Ltd, #122520, clone: E13-161.7
Validation:
Used in several publications (see https://www.biolegend.com/en-gb/products/pacific-blue-anti-mouse-ly-6a-e-sca-1-antibody-3901 ) amongst them:
Herrejon Chavez F, et al. 2023. Nat Commun. 14:2290

c-kit AF780 (APC-Cy7), Thermo Fisher Scientific, #47-1171-82, clone: 2138
Validation:
Used in several publications (see https://www.thermofisher.com/antibody/product/CD117-c-Kit-Antibody-clone-ACK2-Monoclonal/47-1172-82?gclid=Cj0KCQiAwOe8BhCCARIsAGKeD54z_rzvl-2VH9ysxpN2DEUj4LLHMpejjip0d_eRqqhc0g-FPs2kCRwaAis9EALw_wcB&ef_id=Cj0KCQiAwOe8BhCCARIsAGKeD54z_rzvl-2VH9ysxpN2DEUj4LLHMpejjip0d_eRqqhc0g-FPs2kCRwaAis9EALw_wcB:G:s&s_kwcid=AL!3652!3!278870232429!!!g!!!1454324556!63404918784&cid=bid_pca_frg_r01_co_cp1359_pjt0000_bid00000_0se_gaw_dy_pur_con&gad_source=1 ) amongst them:
Kazuhito Naka, et al. 2020. Nat Commun. 11:4681.

CD48-BV605, BioLegend UK Ltd, #103441, clone: HM48-1
Validation:
Used in several publications (see https://www.biolegend.com/en-gb/products/apc-anti-mouse-cd48-antibody-3622 )
amongst them:
Ahrends T, et al. 2021. Cell. 184:5715

CD150-BV650, BioLegend UK Ltd, #115932, clone: TC15-12F12.2
Validation:
Used in several publications (see https://www.biolegend.com/en-us/search-results/pe-anti-mouse-cd150-slam-antibody-1369?GroupID=BLG10572&gad_source=1&gclid=Cj0KCQiAwOe8BhCCARIsAGKeD57ooSjsYZvP0TzD7ECmnfOmEKwWC2HbAy84VC3B66AGYxBHWHGWq1YaArM3EALw_wcB )
amongst them:
Li CC, et al. 2022. Nat Commun. 13:346.

Cd135 PE, BD Biosciences, #553842, clone: A2F10.1
Validation:
Used in several publications (see https://www.bdbiosciences.com/en-gb/products/reagents/flow-cytometry-reagents/research-reagents/single-color-antibodies-ruo/pe-rat-anti-mouse-cd135.553842?tab=citations_references )
amongst them:
Xiao, M., Kondo, S., et al. 2023. Nat Commun. 14:8372

Cd34 FITC, Thermo Fisher Scientific, #11-0431-82, clone: RAM34
Validation:
Used in several publications (see https://www.biocompare.com/9776-Antibodies/11185354-CD34-Monoclonal-Antibody-RAM34-FITC-eBioscience-8482/?pda=9776%7C11185354_0_0%7C1529%7C6%7CCD34 )
amongst them:
Harbour JC, et al. 2023. Cell reports, 28;42(2):112105.

Cd16/32 PerCP PeCy5.5, BioLegend UK Ltd, #101324, clone:93
Validation:
Used in several publications (see https://www.biolegend.com/en-gb/products/percp-cyanine5-5-anti-mouse-cd16-32-antibody-6165?GroupID=BLG9237 )
amongst them:
McAlpine CS, et al. 2021. Nature. 595:701

CD45.1 APC, BioLegend UK Ltd, #110714, clone: A20
Validation:
Used in several publications (see https://www.biolegend.com/en-gb/products/apc-anti-mouse-cd45-1-antibody-2319?GroupID=BLG1933 )
amongst them:
Oh J, et al. 2023. Nat Commun. 14:3278

CD45.2 Pe-Cy7, Thermo Fisher Scientific, #25-0454-82, clone: 104
Validation:
Used in several publications (see https://www.thermofisher.com/antibody/product/CD45-2-Antibody-clone-104-Monoclonal/25-0454-82 )
amongst them:
Stefanie Scherer, et al. 2023. Nat Immunol. 24(3):501-515

CD45.1 BV605, BioLegend UK Ltd, #110738, clone: A20
Validation:
Used in several publications (see https://www.biolegend.com/en-gb/products/brilliant-violet-605-anti-mouse-cd45-1-antibody-7850 )
amongst them:
Scherer S, et al. 2023. Nat Immunol. 24:501
CD45.2 FITC, BioLegend UK Ltd, #109806, clone: 104
Validation:
Used in several publications (see https://www.biolegend.com/en-gb/products/fitc-anti-mouse-cd45-2-antibody-6 )
amongst them:
Rundberg Nilsson A, et al. 2023. iScience. 26:106341

CD4 Pe-Cy5, BioLegend UK Ltd, #100514, clone: RM4-5
Validation:
Used in several publications (see https://www.biolegend.com/en-gb/products/pe-cyanine5-anti-mouse-cd4-antibody-483 )
amongst them:
Yu X, et al. 2020. Nat Commun. 11:1110

Cd8 Pe-Cy5, BioLegend UK Ltd, #100710, clone: 53-6.7
Validation:
Used in several publications (see https://www.biolegend.com/en-gb/products/pe-cyanine5-anti-mouse-cd8a-antibody-156 )
amongst them:
Campisi L, et al. 2022. Nature. 606:945

B220 Pe-Cy5, BioLegend UK Ltd, #103210, clone: RA3-6B2
Validation:
Used in several publications (see https://www.biolegend.com/en-gb/products/pe-cyanine5-anti-mouse-human-cd45r-b220-

antibody-448?GroupID=GROUP658 )
amongst them:
Rundberg Nilsson A, et al. 2023. iScience. 26:106341

Gr1 AF700, BioLegend UK Ltd, #108422, clone: RB6-8C5
Validation:
Used in several publications (see https://www.biolegend.com/en-gb/products/alexa-fluor-700-anti-mouse-ly-6g-ly-6c-gr-1-antibody-3390 )
amongst them:
Fite BZ, et al. 2021. Sci Rep. 11:927.

Mac1 AF700, BioLegend UK Ltd, #101222, clone: M1/70
Validation:
Used in several publications (see https://www.biolegend.com/en-gb/products/alexa-fluor-700-anti-mouse-human-cd11b-antibody-3388 )
amongst them:
Griffin GK, et al. 2023. Nature. 618:834

B220 AF700, BioLegend UK Ltd, #103232, clone: RA3-6B2
Validation:
Used in several publications (see https://www.biolegend.com/en-gb/products/alexa-fluor-700-anti-mouse-human-cd45r-b220-antibody-3408 )
amongst them:
Hao J, et al. 2022. Cell Rep. 41:111804

7 AAD, BioLegend UK Ltd, #420404
Validation:
Used in several publications (see https://www.biolegend.com/en-gb/products/7-aad-viability-staining-solution-1649?GroupID=BLG13283 )
amongst them:
George M, et al. 2023. Cells. 12: 1186

streptavidin BV510, BioLegend UK Ltd, #405233
Validation:
Used in several publications (see https://www.biolegend.com/en-gb/products/brilliant-violet-510-streptavidin-8140 )
amongst them:
Yankova E, et al. 2021. Nature. 593:597

CD5, BioLegend UK Ltd, #100610, clone: 53-7.3
Validation:
Used in several publications (see https://www.biolegend.com/en-gb/products/pe-cyanine5-anti-mouse-cd5-antibody-161 )
amongst them:
Campisi L, et al. 2022. Nature. 606:945

CD11b, BioLegend UK Ltd, #101210, clone: M1/70
Validation:
Used in several publications (see https://www.biolegend.com/en-gb/products/pe-cyanine5-anti-mouse-human-cd11b-antibody-350 )
amongst them:
Zhong W, et al. 2023. Nat Commun. 14:491

TER-119, BioLegend UK Ltd, #116210, clone: TER-119
Validation:
Used in several publications (see https://www.biolegend.com/en-gb/products/pe-cyanine5-anti-mouse-ter-119-erythroid-cells-antibody-1868 )
amongst them:
Rundberg Nilsson A, et al. 2023. iScience. 26:106341

GR-1, BioLegend UK Ltd, #108410, clone: RB6-8C5
Validation:
Used in several publications (see https://www.biolegend.com/en-gb/products/pe-cyanine5-anti-mouse-ly-6g-ly-6c-gr-1-antibody-461 )
amongst them:
Nita A, et al. 2021. Cell Reports. 34(5):108688

CD48, BioLegend UK Ltd, #103411, clone: HM-48-1
Validation:
Used in several publications (see https://www.biolegend.com/en-gb/products/apc-anti-mouse-cd48-antibody-3622 )
amongst them:
Gonçalves R, et al. 2023. iScience. 26:105972

CD150, BioLegend UK Ltd, #115913, clone: TC15-12F12.2
Validation:
Used in several publications (see https://www.biolegend.com/en-gb/products/pe-cyanine7-anti-mouse-cd150-slam-antibody-3056 )
amongst them:
Muto T, et al. 2022. Cell Stem Cell. 29:298.

CD34, BD Pharmigen, #553733, clone: RAM34
Validation:
Used in several publications (see https://www.citeab.com/antibodies/2413422-553733-bd-pharmingen-fitc-rat-anti-mouse-cd34 )
amongst them:
Cockburn, K., Annusver, K., et al. 2022. Nature cell Biology. (24)1692–1700

IL7Ra, BioLegend UK Ltd, #135008, clone: A7R34
Validation:
Used in several publications (see https://www.biolegend.com/en-gb/products/fitc-anti-mouse-cd127-il-7ralpha-antibody-6189?
GroupID=BLG7953 )
amongst them:
Dong J, et al. 2013. PLoS One. 8:e56378.

IL7Ra, BioLegend UK Ltd, #121103, clone: SB/199
Validation:
Used in several publications (see https://www.biolegend.com/en-gb/products/biotin-anti-mouse-cd127-il-7ralpha-antibody-3048?
GroupID=BLG4697 )
amongst them:
Gozdecka M, et al. 2018. Nat Genet. 50:883

Streptavidin, BioLegend UK Ltd, #405206
Validation:
Used in several publications (see https://www.biolegend.com/en-gb/products/pe-cyanine7-streptavidin-1477?GroupID=GROUP23 )
amongst them:
Takano T, et al. 2023. Nat Commun. 14:1451

CD16/32, BD Pharmigen, #553145, clone: 2.4G2
Validation:
Used in several publications (see https://www.citeab.com/antibodies/2408169-553145-bd-pharmingen-pe-rat-anti-mouse-cd16-
cd32 )
amongst them:
Omatsu, Y., Aiba, S., et al. 2022. Nature Commun. 13: 2654

c-KIT, BioLegend UK Ltd, #105812, clone: 2B8
Validation:
Used in several publications (see https://www.biolegend.com/en-gb/products/apc-anti-mouse-cd117-c-kit-antibody-72 )
amongst them:
Wang X, et al. 2023. Nature. 618:808

NK (LSBio, LS-C62548, clone: PK136, lot: 44694)
Validation:
Antibody used in several studies including:
Dovey O, et al., 2017, 2017 Oct 26;130(17):1911-1922

CD45, BD Pharmigen, #563891, clone: 30-F11
Validation:
Used in several publications (see  https://www.citeab.com/antibodies/2407961-563891-bd-horizon-bv510-rat-anti-mouse-cd45 )
amongst them:
Peruzzotti-Jametti, L., Willis, C. M., et al. 2024 Nature. 628: 195–203

CD11b, BD Pharmigen, #557657, clone: M1/70
Validation:
Used in several publications (see  https://www.citeab.com/antibodies/2406880-557657-bd-pharmingen-apc-cy-7-rat-anti-cd11b )
amongst them:
Natarajan, N., Florentin, J., et al. 2024. Nature Commun. 15:7337

GR1, BD Pharmigen, #560603, clone: 1A8
Validation:
Used in several publications (see https://www.citeab.com/antibodies/2408945-560603-bd-horizon-v450-rat-anti-mouse-ly-6g )
amongst them:
Mudalagiriyappa, S., Sharma, J., et al. 2022. Cell Reports.  41:111543

TER119, BD Pharmigen, #557915, clone: TER-119
Validation:
Used in several publications (see https://www.citeab.com/antibodies/2409130-557915-bd-pharmingen-fitc-rat-anti-mouse-ter-119-er )
amongst them:
Gerdes, P., Lim, S. M., et al. 2022. Nature Commun. 13: 7470

CD3e, eBioscience, #12-0031-82, clone: 145-2C11
Validation:
Used in several publications (see  https://www.thermofisher.com/antibody/product/CD3e-Antibody-clone-145-2C11-Monoclonal/16-0031-82?gclid=CjwKCAiA2JG9BhAuEiwAH_zf3iDpTVfCXP3DIkOFJhHsH3kDPRZpkiku-q4as4OxtJymEJG47ld-jxoCecUQAvD_BwE&ef_id=CjwKCAiA2JG9BhAuEiwAH_zf3iDpTVfCXP3DIkOFJhHsH3kDPRZpkiku-q4as4OxtJymEJG47ld-jxoCecUQAvD_BwE:G:s&s_kwcid=AL!3652!3!278870232429!!!g!!!1454324556!63404918784&cid=bid_pca_frg_r01_co_cp1359_pjt0000_bid00000_0se_gaw_dy_pur_con&gad_source=1 )
amongst them:
Vera-Marie E Dunlock, et al. 2022. Cell Reports. 13:111006

anti-5mC Ms antibody, Active Motif, #61479
Validation:
The antibody has been validated for use in dot blot application (please see https://www.activemotif.com/catalog/details/61479/5-methylcytosine-5-mc-antibody-mab-clone-a1).

anti-5hmC Rb antibody, Active Motif, #39769
Validation:
The antibody has been validated for use in dot blot application (see https://www.activemotif.com/catalog/details/39769 and used in several studies including:
Huang et al. 2019. Cancer Cell Apr 15;35(4):677-691.e10.

anti- rabbit HRP-conjugated secondary antibody, 111-035-003, Jackson ImmunoResearch Laboratories Inc.,
Validation:
Antibody used in sever publications, please see: https://www.biocompare.com/9776-Antibodies/5606382-Peroxidase-AffiniPure-Goat-Anti-Rabbit-IgG-H-L/

anti-mouse HRP-conjugated secondary antibody (115-035-146, Jackson ImmunoResearch Laboratories Inc.,
Validation:
Antibody used in sever publications, please see: https://www.jacksonimmuno.com/catalog/products/115-035-146/Goat-Mouse-IgG-HL-Horseradish-Peroxidase.

# Eukaryotic cell lines

Policy information about cell lines and Sex and Gender in Research

| Cell line source(s) | 293-FT (Invitrogen, R70007) |
|---|---|
| Authentication | Purchase from the provider (Invitrogen) as an authenticated cell line. We did not performed any further authentication and used early passages (p<10) for lentiviral vector production. |
| Mycoplasma contamination | Cell line was tested negative for Mycoplasma |
| Commonly misidentified lines (See ICLAC register) | No commonly misidentified cells were used |

# Palaeontology and Archaeology

| Specimen provenance | N/A |
|---|---|
| Specimen deposition | *Indicate where the specimens have been deposited to permit free access by other researchers.* |
| Dating methods | *If new dates are provided, describe how they were obtained (e.g. collection, storage, sample pretreatment and measurement), where they were obtained (i.e. lab name), the calibration program and the protocol for quality assurance OR state that no new dates are provided.* |

☐ Tick this box to confirm that the raw and calibrated dates are available in the paper or in Supplementary Information.

| Ethics oversight | *Identify the organization(s) that approved or provided guidance on the study protocol, OR state that no ethical approval or guidance was required and explain why not.* |
|---|---|

Note that full information on the approval of the study protocol must also be provided in the manuscript.

# Animals and other research organisms

Policy information about studies involving animals; ARRIVE guidelines recommended for reporting animal research, and Sex and Gender in Research

| Laboratory animals | New Dnmt3aR882H/+ mouse model was constructed by flanking native mouse exon 23 of Dnmt3a with loxP sites and introducing human exon 23 containing DNMT3A-R882 mutation. PGK-Puro cassette flanked with Rox sites was inserted after the human exon 23. Mx1-Cre mouse model was reported previously63. Cas9-expressing mice were reported previously22. Mice were analyzed at different age time points, which are specified in the corresponding figure legends/result section. Dnmt3aR882H/+ between 6-12 week of age as well as one year post pIpC were used for characterization of bone marrow compartment by FACS, this information is provided in the figure legend. For transplantation experiment Dnmt3aR882H/+, aged matched competitor and WT were used at age between 6-12 post pIpC. For isolation of cells for screens and for screen validation Dnmt3aR882H/+, Cas9 and WT,Cas9 mice were collected 6-8 weeks post pIpC.<br>B6.SJL-Ptprca Pepcb/BoyJ mice (CD45.1, Jackson Laboratory #002014). Wild type mice: C57BL/6J (CD45.2, Jackson Laboratory #000664). CD45.1/CD45.2 recipient mice, generated by crossing CD45.1 and CD45.2 mice, and were used in this study were between 8 to 18 weeks of age. CD45.1 mice used as competitors for transplantation were age/sex matched to the test cells (e.g., CD45.2-Dnmt3aR882H/+). Mice were housed in specific pathogen-free conditions. All cages were on a 12:12-h light:dark cycle (lights on, 07:30) in a temperature-controlled and humidity-controlled room. Room temperature was maintained at 72 ± 2 °F (22.2 ± 1.1 °C), and room humidity was maintained at 30–70%. |
|---|---|
| Wild animals | No wild animals were used in this study. |
| Reporting on sex | Equal numbers of males and females were used in the study, except for transplant experiments which employed female recipients. |
| Field-collected samples | No field-collected samples were used in the study. |
| Ethics oversight | The in vivo experiments were performed under project license PPL 80/2564 and PP3797858 issued by the United Kingdom Home Office, in accordance with the Animal Scientific Procedures Act 1986. Murine ethical compliance was approved by the University of Cambridge Animal Welfare and Ethical Review Body |

Note that full information on the approval of the study protocol must also be provided in the manuscript.

# Clinical data

Policy information about clinical studies
All manuscripts should comply with the ICMJE guidelines for publication of clinical research and a completed CONSORT checklist must be included with all submissions.

| Clinical trial registration | *Provide the trial registration number from ClinicalTrials.gov or an equivalent agency.* |
|---|---|
| Study protocol | *Note where the full trial protocol can be accessed OR if not available, explain why.* |
| Data collection | *Describe the settings and locales of data collection, noting the time periods of recruitment and data collection.* |
| Outcomes | *Describe how you pre-defined primary and secondary outcome measures and how you assessed these measures.* |

# Dual use research of concern

Policy information about dual use research of concern

## Hazards

Could the accidental, deliberate or reckless misuse of agents or technologies generated in the work, or the application of information presented in the manuscript, pose a threat to:

| No | Yes | |
|---|---|---|
| ☒ | ☐ | Public health |
| ☒ | ☐ | National security |
| ☒ | ☐ | Crops and/or livestock |
| ☒ | ☐ | Ecosystems |
| ☒ | ☐ | Any other significant area |

1500

## Experiments of concern

Does the work involve any of these experiments of concern:

No | Yes
☒ ☐ Demonstrate how to render a vaccine ineffective
☒ ☐ Confer resistance to therapeutically useful antibiotics or antiviral agents
☒ ☐ Enhance the virulence of a pathogen or render a nonpathogen virulent
☒ ☐ Increase transmissibility of a pathogen
☒ ☐ Alter the host range of a pathogen
☒ ☐ Enable evasion of diagnostic/detection modalities
☒ ☐ Enable the weaponization of a biological agent or toxin
☒ ☐ Any other potentially harmful combination of experiments and agents

## Plants

Seed stocks | N/A

Novel plant genotypes | *Describe the methods by which all novel plant genotypes were produced. This includes those generated by transgenic approaches, gene editing, chemical/radiation-based mutagenesis and hybridization. For transgenic lines, describe the transformation method, the number of independent lines analyzed and the generation upon which experiments were performed. For gene-edited lines, describe the editor used, the endogenous sequence targeted for editing, the targeting guide RNA sequence (if applicable) and how the editor was applied.*

Authentication | *Describe any authentication procedures for each seed stock used or novel genotype generated. Describe any experiments used to assess the effect of a mutation and, where applicable, how potential secondary effects (e.g. second site T-DNA insertions, mosiacism, off-target gene editing) were examined.*

## ChIP-seq

### Data deposition

☐ Confirm that both raw and final processed data have been deposited in a public database such as GEO.

☐ Confirm that you have deposited or provided access to graph files (e.g. BED files) for the called peaks.

Data access links
*May remain private before publication.* | *For "Initial submission" or "Revised version" documents, provide reviewer access links. For your "Final submission" document, provide a link to the deposited data.*

Files in database submission | *Provide a list of all files available in the database submission.*

Genome browser session
(e.g. UCSC) | *Provide a link to an anonymized genome browser session for "Initial submission" and "Revised version" documents only, to enable peer review. Write "no longer applicable" for "Final submission" documents.*

### Methodology

Replicates | *Describe the experimental replicates, specifying number, type and replicate agreement.*

Sequencing depth | *Describe the sequencing depth for each experiment, providing the total number of reads, uniquely mapped reads, length of reads and whether they were paired- or single-end.*

Antibodies | *Describe the antibodies used for the ChIP-seq experiments; as applicable, provide supplier name, catalog number, clone name, and lot number.*

Peak calling parameters | *Specify the command line program and parameters used for read mapping and peak calling, including the ChIP, control and index files used.*

Data quality | *Describe the methods used to ensure data quality in full detail, including how many peaks are at FDR 5% and above 5-fold enrichment.*

Software | *Describe the software used to collect and analyze the ChIP-seq data. For custom code that has been deposited into a community repository, provide accession details.*

# Flow Cytometry

## Plots

Confirm that:

☒ The axis labels state the marker and fluorochrome used (e.g. CD4-FITC).

☒ The axis scales are clearly visible. Include numbers along axes only for bottom left plot of group (a 'group' is an analysis of identical markers).

☒ All plots are contour plots with outliers or pseudocolor plots.

☒ A numerical value for number of cells or percentage (with statistics) is provided.

## Methodology

| | |
|---|---|
| Sample preparation | This information is provided in the supplementary method section. |
| Instrument | Flow cytometry analysis was performed using the LSRFortessa instrument (BD). |
| Software | Flow cytometry analysis was performed using a LSRFortessa instrument and resulting data were subsequently analysed using FlowJo (version 10.8.2). |
| Cell population abundance | The purity of the relevant cells was over 98% in the post-sort fraction as assessed by flow cytometry. |
| Gating strategy | Gating strategy is provided in Extended Data Fig. 6e-f |

☒ Tick this box to confirm that a figure exemplifying the gating strategy is provided in the Supplementary Information.

# Magnetic resonance imaging

## Experimental design

| | |
|---|---|
| Design type | *Indicate task or resting state; event-related or block design.* |
| Design specifications | *Specify the number of blocks, trials or experimental units per session and/or subject, and specify the length of each trial or block (if trials are blocked) and interval between trials.* |
| Behavioral performance measures | *State number and/or type of variables recorded (e.g. correct button press, response time) and what statistics were used to establish that the subjects were performing the task as expected (e.g. mean, range, and/or standard deviation across subjects).* |

## Acquisition

| | |
|---|---|
| Imaging type(s) | *Specify: functional, structural, diffusion, perfusion.* |
| Field strength | *Specify in Tesla* |
| Sequence & imaging parameters | *Specify the pulse sequence type (gradient echo, spin echo, etc.), imaging type (EPI, spiral, etc.), field of view, matrix size, slice thickness, orientation and TE/TR/flip angle.* |
| Area of acquisition | *State whether a whole brain scan was used OR define the area of acquisition, describing how the region was determined.* |

Diffusion MRI    ☐ Used    ☒ Not used

## Preprocessing

| | |
|---|---|
| Preprocessing software | *Provide detail on software version and revision number and on specific parameters (model/functions, brain extraction, segmentation, smoothing kernel size, etc.).* |
| Normalization | *If data were normalized/standardized, describe the approach(es): specify linear or non-linear and define image types used for transformation OR indicate that data were not normalized and explain rationale for lack of normalization.* |
| Normalization template | *Describe the template used for normalization/transformation, specifying subject space or group standardized space (e.g. original Talairach, MNI305, ICBM152) OR indicate that the data were not normalized.* |
| Noise and artifact removal | *Describe your procedure(s) for artifact and structured noise removal, specifying motion parameters, tissue signals and physiological signals (heart rate, respiration).* |
| Volume censoring | *Define your software and/or method and criteria for volume censoring, and state the extent of such censoring.* |

## Statistical modeling & inference

Model type and settings
> *Specify type (mass univariate, multivariate, RSA, predictive, etc.) and describe essential details of the model at the first and second levels (e.g. fixed, random or mixed effects; drift or auto-correlation).*

Effect(s) tested
> *Define precise effect in terms of the task or stimulus conditions instead of psychological concepts and indicate whether ANOVA or factorial designs were used.*

Specify type of analysis: ☐ Whole brain ☐ ROI-based ☐ Both

Statistic type for inference
(See Eklund et al. 2016)
> *Specify voxel-wise or cluster-wise and report all relevant parameters for cluster-wise methods.*

Correction
> *Describe the type of correction and how it is obtained for multiple comparisons (e.g. FWE, FDR, permutation or Monte Carlo).*

## Models & analysis

| n/a | Involved in the study |
| --- | --- |
| ☐ | ☐ Functional and/or effective connectivity |
| ☐ | ☐ Graph analysis |
| ☐ | ☐ Multivariate modeling or predictive analysis |

Functional and/or effective connectivity
> *Report the measures of dependence used and the model details (e.g. Pearson correlation, partial correlation, mutual information).*

Graph analysis
> *Report the dependent variable and connectivity measure, specifying weighted graph or binarized graph, subject- or group-level, and the global and/or node summaries used (e.g. clustering coefficient, efficiency, etc.).*

Multivariate modeling and predictive analysis
> *Specify independent variables, features extraction and dimension reduction, model, training and evaluation metrics.*

