## [Peer Review File · Nature]

Mitochondrial metabolism sustains DNMT3A-R882-mutant clonal haematopoiesis

Corresponding Author: Professor George Vassiliou

Version 1:

Reviewer comments:

Referee #1

(Remarks to the Author)

This manuscript by Gozdecka et. al and colleagues describes some intriguing concepts observed in clonal hematopoiesis (CH). Clonal hematopoiesis continues to be a topic of broad interest with significant implications for human health within and outside of hematologic implications. The authors aim at the development of a therapeutic approach that can prevent the progression from CH to AML. The group has created a new mouse model to study the role of mutant DNMT3A in CH and identified mitochondrial metabolic vulnerabilities in DNMT3A mutant cells. While certainly intriguing, I do have some concerns.

The biggest concern is that the main novel findings are not validated on human primary samples. Also, I am not fully convinced by the results of the CRISPR dropout screen on primary HSPC ex vivo, which is the major source of the authors' claims.

All in all, there is insufficient data supporting the therapeutic potential of CPIT-2 and metformin in preventing the malignant transformation of CH clones, as no longitudinal experiments were performed. In addition, no validation on human primary samples was performed. In its current form, the study lacks mechanistic insight that can explain the mitochondrial metabolism dependency of DNMT3A-mutant cells. Given the variable experimental settings, it is difficult to compare the results obtained in vitro and in vivo.

Specific comments:

Major:

1. Figure 1: As the authors aim to identify the drugs that can prevent the transformation of CH clones to AML, they should provide more longitudinal data.
2. Not all the R882H mice progress to malignancy such as MPN/AML. Are there differences in OXPHOS, metabolic capacity, and respiration of the HSPC from the animals that progress to the malignancy and those that do not?
3. Figure 1: were there any other cancer-associated mutations at the terminal time point especially in the animals that died?
4. Figure 2: CRISPR dropout screen is a powerful method, however, I have 2 major concerns about the experimental design. First, the rationale for the usage of liquid culture with the cytokines is not well justified, as in Figure 1 the authors show that DNMT3A clones do not acquire a colony-forming advantage until the 3rd replating. Does the current strategy reveal the genes required for the survival of DNMT3A-mutant cells

Also, there is no proper control – I recommend performing a CRISPR screen using matched WT HSPCs if feasible.

5. Additionally, it would be intriguing to see the transplantation capacities of the DNMT3A-mutant and WT cells with the specific knockout of metabolism-involved genes, such as SLC25a1.

6. Figure 3: I am puzzled by the CRISPR screen 'validation' provided by the authors using the data from DepMap portal. First, it is not clear why the authors looked only at 5 of 201 identified genes. Second, some dependency scores are pretty weak – for PDHB and SLC25A1 (2 out of 3 cell lines have a score higher than 0.5, which is considered as the absence of

dependency). Notably, the OCI-AML2 cell line has a different mutation in DNMT3A that is not described and discussed.

7. Figure 3: the selected approach to study OCR in WT and mutant cells is confusing. In the primary transplantation cohort, WT and mutant cells were transplanted in parallel. In the current experimental setting, when the WT and mutant cells are isolated from the same animal, the observed differences in respiration can be the result of the interaction between mutant and WT cells. Can the authors comment on this?

8. Figure 4: Can the authors comment on the mechanism of +/- LT-HSCs expansion upon CTPI2 treatment?

9. Figure 5: one of the characteristics of DNMT3A-mutant cells is the ability to repopulate (also shown in Fig1). I suggest providing at least the data for the serial replating capacity of mutant and WT cells upon treatment with the suggested inhibitors, which can at least partially elaborate on the relevance of these drugs for the prevention of leukemic transformation ("the development of therapeutic approaches to avert AML development").

Minor:

1. How was the 1:2 ratio for CD45.2 and CD45.1 cells for the transplantation? Why do authors decide not to go with 1:1, which is more common?
2. For the validation of CRISPR screen results, I recommend measuring the protein levels to validate successful gene knockout.

Referee #2

(Remarks to the Author)

Gozdecka et al., by means of a genome wide CRISPR screen, identify mitochondrial metabolism as vulnerability in DNMT3A-R882 mutant HSPCs. In vivo administration of metformin suppressed post-transplant expansion of dnmt3a-r882h HSCs. Consistent with this finding in murine models, retrospective analysis of the UKBB showed that prior use of metformin was associated with a decreased prevalence of DNMT3A-R882H CH.

Studying the association between metformin and CH is challenging due to a host of potential confounders. The authors have performed a series of analyses to address potential confounding particularly in regards to diabetes and risk factors for diabetes. Additional work attempting to demonstrate a dose-response relationship between CH and metformin use could also further strengthen the support for a casual relationship between CH and metformin.

My main criticism is that while the authors have shown here compelling evidence that DNMT3A-R882H CH is inhibited by metformin, whether metformin might have a role in DNMT3A-R882H-AML is not shown. Overall though I think the manuscript is well-written and well-executed with compelling data from both experimental work and the UKBB supporting that metformin inhibits DNMT3A-R882H CH.

A few specific comments are below:

1. Have the authors examined whether metformin is associated with decreased AML risk among DNMT3A-R882H CH mutation carriers? Similarly have the studied the association between metformin and risk of AML? In the absence of experimental data that metformin inhibits AML or evidence that metformin use decreases the risk of AML in the UKBB, the authors should temper statements regarding AML. In the abstract, I would recommend removing the statement "Our data propose that modulation of mitochondrial metabolism warrants urgent investigation as a therapeutic strategy for prevention of DNMT3A-R882-mutant AML."
2. DNMT3A-R882-mutant CH overall is low risk with the vast majority of individuals never progressing to AML. While metformin has been used widely there are significant side effects including the common side effect of GI discomfort that while low-risk does impact quality of life, B12 deficiency (rare) and lactic acidosis (very rare). Thus, the risk-benefit for applying metformin to all individuals with DNMT3A-R882-mutant CH is not clear. If implemented clinical it would likely need to be limited to a rare high-risk population. In this vein, the concluding statement should be tempered: "Metformin has been an established treatment for Type 2 diabetes for decades and is currently taken by millions of people worldwide with an acceptable and predictable side-effect profile. This reassuring clinical experience, along with our mechanistic and functional pre-clinical data, calls for the urgent initiation of clinical trials of metformin usage specifically in individuals with DNMT3A-R882-mutant CH".
3. The series of analyses the authors have performed in figures 6 and extended data figure 8 are very helpful in addressing potential sources of confounding. Establishing a dose-response relationship between metformin use and DNMT3A-R882H CH would also be helpful. This would include studying the association between the duration of metformin use and CH, the relationship between metformin and CH VAF and for individuals with prior metformin use, the impact of the time between stopping metformin and blood draw on CH-metformin associations.

Referee #3

(Remarks to the Author)

This manuscript explores the potential of timely interventions in preventing the malignant progression of clonal hematopoiesis (CH) into myeloid neoplasms (MNs), focusing on DNMT3A-R882 mutations, a common precursor in acute myeloid leukemia (AML). They use a conditional Dnmt3aR882H mouse model to study these mutations' role in clonal expansion and identifies metabolic vulnerabilities that could be targeted therapeutically. They performed a genome-wide CRISPR screen on primary HSPCs that identified many genetic vulnerabilities related to mitochondrial metabolism and

OXPHOS. Notably, SLC25A1 (mitochondrial citrate-malate carrier) was the top hit. Metabolic analysis confirmed the Dnmt3aR882H HSPCs' heavier reliance on OXPHOS compared to normal HSCs. The paper demonstrates treatments targeting specific vulnerabilities of mitochondrial citrate carrier or complex I by metformin reversed the growth advantage of mutated cells over wild type. Importantly analysis of 412,234 UK Biobank participants revealed that individuals taking metformin had a markedly lower prevalence of DNMT3A5 R882-mutant CH (OR = 0.49[95% CI: 0.32-0.74], P = 0.00081). Overall, the paper is timely and interesting. However, there are key mechanistic insights they are missing. Strength of the paper is in the CRISPR screen hits and UK biobank linking metformin to DNMT3A5 R882 mutation.

Weakness that needs to be addressed:

It is interesting that other potential pathways in citrate-malate shuttle were not observed in the screen including ACLY, which in the cytosol would convert citrate into the cytosol for acetyl-CoA for lipogenesis or histone acetylation and oxaloacetate for aspartate and asparagine. They should do metabolomics to see levels of these metabolites. It suggests that it is not likely decrease in metabolites linked to growth that is the likely mechanism.

A major complication in humans with SLC25A1 loss is D, L-2hydroxyglutariaciduria. This is due to accumulation of D and L forms of 2HG. (PMID: 37503155 and PMID: 24687295 and PMID: 27856334). These metabolites are known to cause TET inhibition. Could it be that SLC25A1 loss accumulates L or D-2HG and then causes TET inhibition in DNMT3A5 R882 mutation to restore DNA methylation? Also, complex I could increase L-2HG by increasing the NADH/NAD ratio.

(1) They should measure both L and D-2HG (will require a special column coupled with MS) in cells with SLC25A1 loss and complex I inhibitors.

(2) Does DNA methylation increase in SLC25A1 null cells? L or D-2HG inhibition would restore DNA methylation in DNMT mutant cells by inhibiting TETs.

(3) If these metabolites accumulate then they should add cell permeable L and D-2HG to examine DNA methylation in DNMT mutant cells without SLC25A1 loss. L and D-2HG are made from aKG. Aminooxyacetate (AOA) is known to inhibit the generation of aKG from glutamate would lower L and D-2HG levels in SLC25A1 cells. They should see what AOA does in SLC25A1 loss cells.

Referee #4

(Remarks to the Author)

In this manuscript Gozdecka et al report on a clinically important finding that expansion of Dnmt3a-mutant HSCs seen in clonal hematopoiesis can be dampened by the commonly used drug metformin. Dnmt3a is the most commonly mutated gene in CH and some mutations in this gene portend a higher risk of progression to AML and poor outcomes from AML. The authors performed a genome wide CRISPR screen to identify genetic vulnerabilities in Dnmt3a-mutant HSCs. Depleted genes were enriched in genes involved in metabolism, and overexpression of several of these genes have also been linked to poor outcomes in AML. Dnmt3a mutant HSCs have increased reliance on mitochondrial respiration. By investigating druggable candidates among these list, the authors determined that inhibited Slc25a1 or Ndufb11 of the electron transport chain limits Dnmt3a mutant but not WT HSC expansion. Metformin treatment of mice transplanted with Dnmt3a mutant marrow resulted in a dampening of Dnmt3a mutant HSC expansion in the BM without affected total frequency of cells or WT cells. Analysis of human data in the UK Biobank revealed a significant reduction in prevalence of Dnmt3a R882H-mutant CH among people taking metformin compared to those not taking metformin. Overall this is an excellent study with translationally important and impactful findings. There are a few concerns that should be addressed:

Specific concerns:

1) The authors find in the mouse model that metformin does not affect peripheral blood counts of Dnmt3a mutant cells. Only the expansion of BM HSCs is depleted in the mouse model. This is not consistent with the human data that indicates prevalence of Dnmt3a mutant CH based on peripheral blood sequencing was lower in people taking metformin. Why are the peripheral blood counts not affected in mice? What happens if the metformin treatment is carried out for longer than 7 weeks?

2) Secondary transplants of marrow from metformin treated chimeric mice would strengthen the data to ascertain a functional, not just phenotypic depletion.

3) The authors state that PB analysis showed no differences in Dnmt3a mutant PB cell numbers (Figure 5g) – was this also true if you consider myeloid, B and T cell subsets separately?

4) Dnmt3a mutations can be classified by protein stability (Huang Cancer Discovery 2022). If the authors classify all UK biobank participants with Dnmt3a mutation into high and low functionality mutations, is the OR of CH with low functionality mutations lower in metformin treated individuals?

5) Can the authors account for the amount of time the UK Biobank participants have been taking metformin and determine a time to effect?

6) Figure 4e p-values shown are reflecting what comparison and what statistical test? Same for Figure 5i.

7) Key references are missing from the introduction which are mentioned but not appropriately cited: Challen Nat Gen 2011; Hormaechea Cell Stem Cell 2021; San Miguel Cancer Discovery 2022; Jeong Cell Reports 2018.

Version 3:

Reviewer comments:

Referee #1

(Remarks to the Author)

The authors made significant improvements and performed all feasible experiments mentioned by me and other reviewers.

Validation of the murine data on human samples in the UK Biobank and ex vivo is quite impressive. However, the ex vivo experiment was performed using only one R882 CH sample. On the other hand, I do appreciate the fact that CH samples are extremely rare. All in all, integrating this data with comprehensive murine data forms a complete picture, thus I am satisfied with the conclusions and claims.

I would recommend a minor text edit:

- "The effect of metformin was recapitulated using human DNMT3AR882 CH cells" - to avoid the confusion of the readers about the number of samples I would recommend changing "DNMT3AR882 CH cells" with "DNMT3AR882 CH sample".

Referee #2

(Remarks to the Author)

My comments from the previous version have been addressed. They have moderated statements regarding a linkage between metformin use and protection against AML and I believe the discussion regarding this is appropriate. The MR analyses now included are interesting but have some important caveats that I believe should be addressed. That being said, even though the MR analysis and their retrospective data does not provide definitive casual evidence of a linkage between metformin use and DNMT3A suppression, taken together with their functional data I do find the evidence to be strong.

Specifically in regards to the mendelian randomization analysis there are a few major issues:

- 1) The genetic instruments used by Zheng et al., were chosen based on an association with "both expression levels of corresponding in the cis-acting regions (500 kb window from the centinal variant) and HbA1c levels in the UKBB". The weighted GRS for metformin targets were "proxied by their association with HbA1c" (i.e the beta coefficients are the association with HbA1c level not protein or RNA expression of the gene product). Thus, the weighted GRS estimates for metformin targets are strongly associated with HgA1c and risk of DM. A limitation of the human data in the UKBB is that it is difficult to tease out the impact of metformin use from diabetes or glycemia itself on causing the association with DNMT3A. The MR data here does not help with this issue since the assumption of pleiotropy is violated. Indeed the GRS estimates for the metformin targets were also associated with BMI in the Zheng article which is another source of pleiotropy. A few analyses that could help suggest (but not prove) this is not completely driven by pleiotropy would be to look at the association between GRS for metformin targets among those without DM and among those of normal weight.
- 2) Given the above limitations, statements regarding the strength of the MR data should be tempered with appropriate discussion of the limitations of the analysis in the discussion
- 3) The authors state they are performing two sample MR analysis but since the estimates for the genetic instruments come from the association with HbA1c levels in the UKBB, this appears to be a one-sample MR analysis
- 4) The authors should compare results using different MR tests and perform sensitivity analyses studying the impact of individual SNPs on DNMT3A prevalence (such as Supplementary Figure 5 in the Zheng et al article)

On a separate note, given a link possible link between metformin and adiposity brought up by the Zheng et al article, the association between metformin use and DM should be adjusted for both BMI and waist-to-hip ratio

Referee #3

(Remarks to the Author)

The authors have adequately responded.

Referee #4

(Remarks to the Author)

The authors have responded well to the reviews. This is a comprehensive study that used a CRISPR screen to identify therapeutic vulnerability in Dnmt3a mutant clonal hematopoiesis, validate this in an animal model, and then validate that use of metformin to inhibit mitochondrial metabolism correlates with less CH in UK biobank data. The data are innovative, highly translationally important, and performed with high quality methodology and rigor.

My only remaining question is regarding the UK biobank data: was there any reduction in those with large chip among those on metformin compared to those not on metformin? (separate out the proportion of small CHIP, large CHIP)—one might predict that among those on metformin there is a relative decrease in those with large CHIP (>10%)

Version 4:

Reviewer comments:

Referee #2

(Remarks to the Author)

Thank you for the detailed response and the interesting discussion. I agree with the decision to remove the MR analysis given the limitations and the strength of other functional data supporting the association.

Referee #4

(Remarks to the Author)

Again, this is a very rigorous, well performed, and important study which should be published.

In the last round of review, I asked whether there was any reduction in those with large chip among those on metformin compared to those not on metformin. The reason behind my question was to know whether the Dnmt3a-suppressive effects of metformin were as effective for people with large CHIP as for those with small CHIP.

The data presented in the rebuttal argue against this hypothesis – in fact the data suggest that the beneficial effects of metformin are only present if applied while CHIP is small. If the goal is to prevent progression to leukemia, then these data suggest that CHIP must be detected early and that metformin should be applied before the clone reaches a large size.

This is valuable data that should be mentioned in the study.
